# Learning to Abstain From Uninformative Data

**Yikai Zhang** *
*Morgan Stanley*

*Yikai.Zhang@morganstanley.com*

**Songzhu Zheng** *
*Morgan Stanley*

*Songzhu.Zheng@morganstanley.com*

**Mina Dalirrooyfard** *
*Morgan Stanley*

*Mina.Dalirrooyfard@morganstanley.com*

**Pengxiang Wu**
*Snap Inc.*

*pxiangwu@gmail.com*

**Anderson Schneider**
*Morgan Stanley*

*Anderson.Schneider@morganstanley.com*

**Anant Raj**
*Morgan Stanley*

*Anant.Raj@morganstanley.com*

**Yuriy Nevmyvaka**
*Morgan Stanley*

*Yuriy.Nevmyvaka@morganstanley.com*

**Chao Chen**
*Stony Brook University*

*chao.chen.1@stonybrook.edu*

## Abstract

Learning and decision-making in domains with naturally high noise-to-signal ratios – such as Finance or Healthcare – is often challenging, while the stakes are very high. In this paper, we study the problem of learning and acting under a general noisy generative process. In this problem, the data distribution has a significant proportion of uninformative samples with high noise in the label, while part of the data contains useful information represented by low label noise. This dichotomy is present during both training and inference, which requires the proper handling of uninformative data during both training and testing. We propose a novel approach to learning under these conditions via a loss inspired by the selective learning theory. By minimizing this loss, the model is guaranteed to make a near-optimal decision by distinguishing informative data from uninformative data and making predictions. We build upon the strength of our theoretical guarantees by describing an iterative algorithm, which jointly optimizes both a predictor and a selector, and evaluates its empirical performance in a variety of settings.

## 1 Introduction

Despite the success of machine learning (ML) in computer vision (Krizhevsky et al., 2009; He et al., 2016a; Huang et al., 2017) and natural language processing (Vaswani et al., 2017; Devlin et al., 2018), the power of ML is yet to make similarly-weighty impact in other areas such as Finance or Public Health. One major challenge is the inherently high noise-to-signal ratio in certain domains. In financial statistical arbitrage, for instance, the spread between two assets is usually modeled using Orstein-Uhlembeck processes (Øksendal, 2003; Avellaneda & Lee, 2010). Spreads behave almost randomly near zero and are naturally unpredictable.

---

*: Equal contribution.

They become predictable in certain rare pockets/scenarios: for example, when the spread exceeds a certain threshold, it will move towards zero with a high probability, making arbitrage profits possible. In cancer research, due to limited resources, only a small number of the most popular gene mutations are routinely tested for differential diagnosis and prognosis. However, because of the long tail distribution of mutation frequencies across genes, these popular gene mutations can only capture a small proportion of the relevant list of driver mutations of a patient (Reddy et al., 2017). For a significant number of patients, the tested gene mutations may not be in the relevant list of driver mutations and their relationship w.r.t. the outcome may appear completely random. Identifying these patients automatically will justify additional gene mutation testing.

These high noise-to-signal ratio datasets pose new challenges for learning. New methods are required to deal with the large fraction of uninformative/high-noise data in both the training and testing stages. The source of uninformative data can be either due to the random nature of the data-generating process or due to the fact that the real causing factors are missing from the data. Direct application of standard supervised learning methods to such datasets is both challenging and unwarranted. Deep neural networks are even more affected by the presence of noise, due to their strong memorization power (Zhang et al., 2017): they are likely to overfit the noise and make overly confident predictions where weak/no real structure exists.

In this paper, we propose a novel method for learning on datasets where a significant portion of content has high noise. Instead of forcing the classifier to make predictions for every sample, we learn to first decide whether a datapoint is informative or not. Our idea is inspired by the classic selective prediction problem (Chow, 1957), in which one learns to select a subset of the data and only predict on that subset. However, the goal of selective prediction is very different from ours. A selective prediction method pursues a balance between coverage (i.e. proportion of the data selected) and conditional accuracy on the selected data, and does not explicitly model the underlying generative process. In particular, the aforementioned balance needs to be specified by a human expert, as opposed to being derived directly from the data. In our problem, we assume that uninformative data is an integral part of the underlying generative process and needs to be accounted for. By definition, no learning method, no matter how powerful, can successfully make predictions on *uninformative* data. Our goal is therefore to identify these uninformative/high noise samples and, consequently, to train a classifier that is less influenced by the noisy data.

Our method learns a *selector*, $g$, to approximate the optimal indicator function of informative data, $g^*$. We assume that $g^*$ exists as a part of the data generation process, but it is never revealed to us, even during training. Instead of direct supervision, we therefore must rely on mistakes made by the predictor in order to train the selector. To achieve this goal, we propose a novel *selector loss* enforcing that (1) the selected data best fits the predictor, and (2) in the portion of the data where we abstain from forecasting, the predictor's performance is similar to random chance. The resulting loss function is quite different from the loss in classic selective prediction, which penalizes all unselected data equally.

We theoretically analyze our method under a general noisy data generation process, imposing an additional structure on the standard data-dependent label noise model (Massart & Nédélec, 2006; Hanneke, 2009). We distinguish informative versus uninformative data via a gap in the label noise ratio. A major contribution of this paper is the derivation of theoretical guarantees for the selector loss using empirical risk minimization (ERM). A minimax-optimal sample complexity bound for approximating the optimal selector is provided. We show that optimizing the selector loss can recover nearly all the informative data in a PAC fashion (Valiant, 1984). This guarantee holds even in a challenging setting where the uninformative data has purely random labels and dominates the training set. By leveraging the estimated selector, one can further pick out the informative subset of the training samples. We prove that the classifier generated through risk minimization conditional on the informative subset exhibits a superior upper bound on risk compared to a conventional classifier trained using the complete training dataset.

The theoretical results generalize the method to a more realistic setting where the sample size is limited. Furthermore, the initial predictor is not sufficiently close to the ground truth. Our method yields an iterative algorithm, in which both the predictor and the selector are progressively optimized. The selector is improved by optimizing our novel selector loss. Meanwhile, the predictor is strengthened by optimizing the empirical risk: re-weighted based on the output from the selector, where uninformative samples identified by the

selector are down-weighed. Experiments on both synthetic and real-world datasets demonstrate the merit of our method compared to existing baselines.

## 2 Related work

**Learning with untrusted data** aims to recover the ground truth model from a partially corrupted dataset. Different noise models have been studied, including random label noise (Bylander, 1994; Natarajan et al., 2013), Massart Noise (Massart & Nédélec, 2006; Awasthi et al., 2015; Hanneke, 2009; Hanneke & Yang, 2015; Yan & Zhang, 2017; Diakonikolas et al., 2019; 2020; Zhang & Li, 2021) and adversarial noise (Kearns & Li, 1993; Kearns et al., 1994; Kalai et al., 2008; Klivans et al., 2009; Awasthi et al., 2017). Our noise model is similar to General Massart Noise (Massart & Nédélec, 2006; Hanneke, 2009; Diakonikolas et al., 2019), where the label noise is *data-dependent*, and the label can be generated via a purely random coin flipping. The major distinction in our noisy generative model is the existence of some uninformative data with high noise in label compared to informative data with low noise in label. We characterize such uninformative/informative data structure via a non-vanishing label noise ratio gap. While there exists a long history of literature studying training classifiers with noisy label (Thulasidasan et al., 2019), we are the first to investigate learning a model robust against label noise at inference stage. We study the case where uninformative samples are an integral part of the generative process and thus will appear during inference stage as well, where they must be discarded if detected. We view this as a realistic setup in industries like Finance and Healthcare.

**Selective learning** is an active research area (Chow, 1957; 1970; El-Yaniv et al., 2010; Kalai et al., 2012; Nan & Saligrama, 2017; Ni et al., 2019; Acar et al., 2020; Gangrade et al., 2021a) that expands on the classic selective prediction problem. It focuses on how to select a subset of data for different learning tasks, and has also been further generalized to other problems, e.g., learning to defer human expert (Madras et al., 2018; Mozannar & Sontag, 2020). We can approximately classify existing methods into 4 categories: Monte Carlo sampling based methods (Gal & Ghahramani, 2016; Kendall & Gal, 2017; Pearce et al., 2020), margin based methods (Fumera & Roli, 2002; Bartlett & Wegkamp, 2008; Grandvalet et al., 2008; Wegkamp et al., 2011; Zhang et al., 2018), confidence based methods (Wiener & El-Yaniv, 2011; Geifman & El-Yaniv, 2017; Jiang et al., 2018) and customized selective loss (Cortes et al., 2016; Geifman & El-Yaniv, 2019; Liu et al., 2019; Gangrade et al., 2021c). Notably, several works propose customized losses, and incorporate them into neural networks. In (Geifman & El-Yaniv, 2019), the network maintains an extra output neuron to indicate rejection of datapoints. Liu et al. (2019) introduces the Gambler loss where a cost term is associated with each output neuron and a doubling-rate-like loss function is used to balance rejections and predictions. Thulasidasan et al. (2019) also applies an extra output neuron for identifying noise label to improve the robustness in learning. Huang et al. (2020) adopts a progressive label smoothing method which prevents DNN from overfitting and improves selective risk when applied to selective classification task. Cortes et al. (2016) performs data selection with an extra model and introduces a selective loss that helps to maximize the coverage ratio, thus trading off a small fraction of data for better precision. Sharing a similar spirit with (Kalai et al., 2012), (Gangrade et al., 2021c) applies a one-sided prediction method to model a high confidence region for each individual class, and maximizes coverage while maintaining a low risk level.

Existing works on selective prediction are all motivated by the trade off between accuracy and coverage - i.e. one wants to make confident predictions to achieve higher precision while maintaining a reasonable recall. To the best of our knowledge, our paper is the first to investigate the case where some (or even the majority) of the data is uninformative, and thus must be discarded at test time. Unlike the selective prediction, our framework considers a latent never-revealed ground truth indicator function about whether a data point should be selected or not. Our method is guaranteed to identify those uninformative samples.

## 3 Problem formulation

In this section, we describe the framework for the inherently noisy data generation process that we study.

**Definition 1** (Noisy Generative Process)**.** *Let $\alpha \in (0,1)$ be a problem-dependent constant and $\Omega_{\mathcal{D}} \subseteq \mathbb{R}^d$ be the support of $\mathcal{D}_\alpha$ below. We define* Noisy Generative Process *by the following notation $\boldsymbol{x} \sim \mathcal{D}_\alpha$ where*

$$\mathcal{D}_\alpha \equiv \begin{cases} \boldsymbol{x} \sim \mathcal{D}_U & \text{with prob. } 1 - \alpha \quad \textbf{\textit{(Uninformative)}} \\ \boldsymbol{x} \sim \mathcal{D}_I & \text{with prob. } \alpha \qquad \textbf{\textit{(Informative)}}. \end{cases} \tag{1}$$

*Given $\mathcal{X} \subseteq \mathbb{R}^d$, let the ground truth labeling function $f^* : \mathcal{X} \to \{0,1\}$ be in hypothesis class $\mathcal{F}$. Suppose $\{\Omega_U, \Omega_I\}$ is a partition of $\Omega_\mathcal{D}$. Let $\lambda(\boldsymbol{x}) \in (\bar{\lambda}, \frac{1}{2}]$ with $\bar{\lambda} > 0$, the latent informative/uninformative status $z \in \{1,0\}$ has posterior distribution:*

$$\mathbb{P}[z = 1 | \boldsymbol{x}] \equiv \begin{cases} \frac{1}{2} - \lambda(\boldsymbol{x}), & \text{if } \boldsymbol{x} \in \Omega_U \\ \frac{1}{2} + \lambda(\boldsymbol{x}), & \text{if } \boldsymbol{x} \in \Omega_I. \end{cases} \tag{2}$$

*The observed data $(\boldsymbol{x}, y)$ is generated according to:*

$$\begin{aligned} &\boldsymbol{x} \sim \mathcal{D}_\alpha; \\ &z \sim \mathbb{P}[z | \boldsymbol{x}]; \\ &\begin{cases} y \sim Bernoulli(0.5), & \text{if } z = 0 \\ y = f^*(\boldsymbol{x}), & \text{if } z = 1. \end{cases} \end{aligned} \tag{3}$$

Since $\lambda(\boldsymbol{x}) > 0$, $\boldsymbol{x}$ from $\Omega_U$ has a lower probability to be observed with true label compared to $\Omega_I$, thus it can be viewed as uninformative data in a relative sense. On the contrary, $x$ from $\Omega_I$ can be viewed as informative data. Our Noisy Generative Process follows standard data-dependent label noise, e.g., Massart Noise (Massart & Nédélec, 2006) and Benign Label Noise (Hanneke, 2009; Hanneke & Yang, 2015; Diakonikolas et al., 2019) with label noise ratio $\frac{1}{4} - \frac{(2g^*(\boldsymbol{x})-1)\lambda(\boldsymbol{x})}{2}$. Indeed, one can always choose $\lambda(\boldsymbol{x}) \in [0, \frac{1}{2}]$ and $\alpha \in [0,1]$ to replicate General Massart noise. Compared to classical label noise models, the assumption $\lambda(\boldsymbol{x}) > \bar{\lambda}$ introduces a label noise ratio gap, which distinguishes the informative and uninformative data. In Equation 3, the $Bernoulli(0.5)$ label noise serves as a proxy for "white noise" in label corruption. When $\lambda(\boldsymbol{x}) = \frac{1}{2}$ and $\boldsymbol{x} \in \Omega_U$, $Bernoulli(0.5)$ random label noise can be viewed as the strongest known non-adversarial label noise, of both theoretical and practical interest (Diakonikolas et al., 2019). Such $Bernoulli(0.5)$ random label noise could happen when hard-to-classify examples are shown to human annotators (Klebanov & Beigman, 2010), or when fluctuations in financial markets closely resemble a random walk (Tsay, 2005).

A typical setting that is studied in this work is the case that both values of $1 - \alpha$ and $\lambda(x)$ are non-vanishing, i.e., there is a significant fraction of uninformative data (large $1 - \alpha$) and the label noise ratio gap is distinguishable between informative and uninformative data (large $\bar{\lambda}$).

The next definition describes a recoverable condition of the optimal function for the latent informative/uninformative status $z$.

**Definition 2** ($\mathcal{G}$-realizable). *Given support $\Omega_\mathcal{D}$ and $\lambda(\boldsymbol{x}) \in (\bar{\lambda}, \frac{1}{2}]$, let the posterior distribution of $z$ be defined in Equation (2). We say $\Omega_\mathcal{D}$ is $\mathcal{G}$-realizable if there exists $g^* \in \mathcal{G} : \mathcal{X} \to \{0,1\}$ satisfying $g^*(\boldsymbol{x}) = \mathbb{1}\{\mathbb{P}[z = 1 | \boldsymbol{x}] > \frac{1}{2}\}$.*

Ideally, one would want to select all informative samples where signal dominates noise. This can be done via recovering $g^*(\cdot)$, which we view as the *ground truth selector* we wish to recover. The $\mathcal{G}$-realizable condition is analogous to the realizability condition (Massart & Nédélec, 2006; Hanneke & Yang, 2015) in the classical label noise problem. The major difference and challenge in recovering $g^*(\cdot)$ compared to learning a classifier, is that there is no direct observation on the informative/non-informative status $z$. The major contribution of this work is proposing a natural selector risk which recovers $g^*(\cdot)$ without observing the latent variable $z$.

Having introduced the data generation process, we now describe the learning task:

**Assumption 1.** *Data $S_n = \{\boldsymbol{x}_i, y_i\}_{i=1}^n$ is i.i.d generated according to the* Noisy Generative Process *(Definition 1), with $f^* \in \mathcal{F}$ and support $\Omega_\mathcal{D}$ satisfies $\mathcal{G}$-realizable condition.*

Given the above assumption, we are interested in the following learning task:

**Problem 1** (Abstain from Uninformative Data). *Under Assumption 1 with i.i.d observations from $\mathcal{D}_\alpha$, we aim to learn a selector $\widehat{g} \in \mathcal{G}$ that is close to $g^*(\boldsymbol{x})$ and predictor $\widehat{f}$ with low selective risk.*

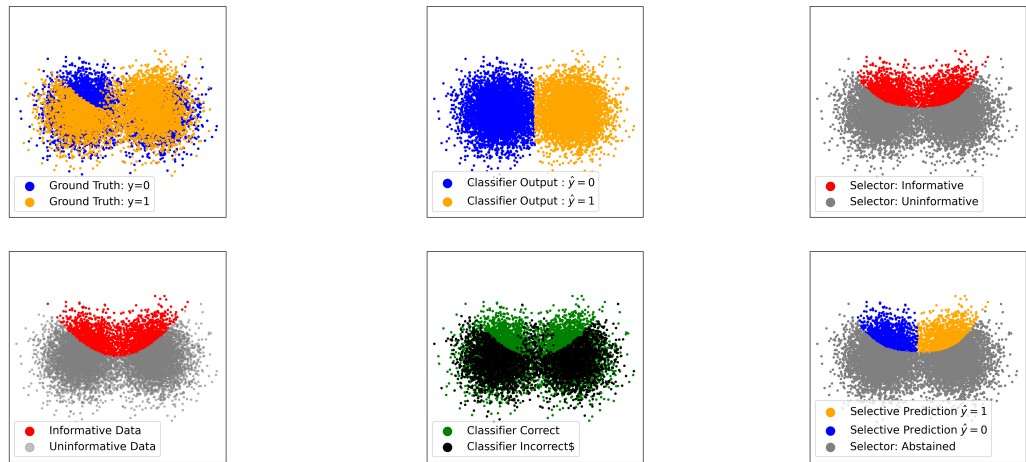

(a) Inform/Uninformative Data    (b) Correct/Incorrect classification    (c) Selective classification

Figure 1: Illustration of the learning strategy when $x$ is distributed according to a Gaussian mixture. We replace the 0-1 loss with hinge loss and train SVM models for $f$ and $g$. (a) upper panel shows the original dataset and bottom panel shows the region of informative (easy) and uninformative (hard) data. (b) shows that the classifier has high accuracy in the informative region, but low accuracy in the uninformative region. In (c), the selector trained with $\widehat{f}$ successfully recovers informative support thus resulting in low selective risk, and we abstain from making a prediction elsewhere.

## 4  Our method

In this section, we present our approach to learning and abstaining in the presence of uninformative data (Problem 1). The main challenge is that the latent informative/uninformative status of a datapoint is unknown. Our main idea is to introduce a novel yet natural *selector loss* function that trains a selector based on the performance of the best predictor (Section 4.1). In Section 5, we present our main theoretical result. We show that, given any reasonably good classifier, by finding a selector minimizing the proposed selector loss, we can solve Problem 1 with minimax-optimal sample complexity. Inspired by the theoretical results, in Section 6, we propose a practical algorithm that iteratively optimizes the predictor and the selector.

### 4.1  Selector loss

In an idealized setting, when access to latent informative/uninformative variables $\{(\boldsymbol{x}_i, z_i)\}_{i=1}^n$ is available, recovering $g^*$ shares a similar spirit with learning a classifier under label noise. It suffices to minimize the following classical classification risk :

$$Non\text{-}Realizable\ Risk(g; S_n) \equiv \sum_{i=1}^n \mathbb{1}\{g(\boldsymbol{x}_i) \neq z_i\} \tag{4}$$

However, in practice $z$ is never revealed. To learn a selector without direct supervision, we have to leverage the performance of a predictor $f$. We propose to replace $z$ in the Equation 4 with a *pseudo-informative label* $\mathbb{1}\{f(\boldsymbol{x}) = y\}$, which has randomness coming from $z$ and noisy label $y$.

**Definition 3** (Selector Loss). *Let scalar $\beta > 0$ be some weight coefficient, given $f \in \mathcal{F}$ and its selector $g \in \mathcal{G}$, we define the following empirical version of weighted 0-1 type risk w.r.t $g(\cdot)$ as selector risk:*

$$R_{S_n}(g; f, \beta) \equiv \sum_{i=1}^n \left\{ \beta \mathbb{1}\{f(\boldsymbol{x}_i) \neq y_i\} \mathbb{1}\{g(\boldsymbol{x}_i) > 0\} + \mathbb{1}\{f(\boldsymbol{x}_i) = y_i\} \mathbb{1}\{g(\boldsymbol{x}_i) \leq 0\} \right\} \tag{5}$$

*where $S_n = \{(\boldsymbol{x}_i, y_i)\}_{i=1}^n$ defined in Assumption 1.*

The selector loss is also a natural metric to evaluate the quality of the selector. This loss penalizes when (1) the predictor makes a correct prediction on a datapoint that the selector considers uninformative and

abstains from, or (2) the predictor makes an incorrect prediction when the selector considers informative. Intuitively, the loss will drive the selector to partition the domain into informative and uninformative regions. Within the informative region, the predictor is supposed to fit the data well, and should be more accurate. Meanwhile, within the uninformative region, the label is random and the predictor is prone to make mistakes.

Note that there are two types of errors penalized in the selector loss: an incorrect prediction on a selected datapoint, $(f(\boldsymbol{x}) \neq y) \wedge (g(\boldsymbol{x}) > 0)$, and a correct prediction on an unselected datapoint, $(f(\boldsymbol{x}) = y) \wedge (g(\boldsymbol{x}) \leq 0)$. Since the label noise is non-adversarial, $y$ tends to have higher probability of coincidence with $f^*(\boldsymbol{x})$, introducing imbalance on the pseudo-informative label. We thus use $\beta$ to weigh these two types of errors in the loss. An analysis can be found in section A.1 on the choice of $\beta$. Our theoretical analysis suggests that, for a wide range of $\beta$, the accuracy of the selector is guaranteed. In particular, the range of $\beta$ depends on the value of label noise ratio gap $\lambda$, empirically, many values work - see Appendix - and we choose $\beta = 3$ throughout. Our experiments show stability with regard to this choice.

**Learning a selector with the novel loss.** To learn a selector, one can follow standard procedures e.g., empirical risk minimization (ERM), to get a predictor $\widehat{f}$ with reasonable quality. The selector can be estimated by minimizing the selector loss $\widehat{g} = \arg\min_{g \in \mathcal{G}} R_{S_n}(g, \widehat{f}, \beta)$, conditioned on the estimated predictor $\widehat{f}$.

In Figure 1, we show an example of using the ERM strategy using SVM with 0-1 loss replaced by hinge loss. In this case, the losses are all convex and the empirical minimizers $\widehat{f}$ and $\widehat{g}$ can be computed exactly.

In practice, however, empirical minimization is not always possible, as optimization for general complex models (e.g., DNNs), resulting in non-convex losses, remains an open problem. We therefore propose a practical algorithm in the spirit of our theoretical results: it jointly learns $f$ and $g$ by minimizing the selector loss and a reweighed classification risk iteratively (see Section 6).

## 5   Theoretical results

In this section, we present our theoretical results. The main one can be summarized in the following (informal) statement.

**Main Result (Informal)**   For any reasonably good predictor $\widehat{f}$, with sufficient data, the selector $\widehat{g}$ estimated using $\widehat{g} = \arg\min_{g \in \mathcal{G}} R_{S_n}(g, \widehat{f}, \beta)$ is sufficiently close to the targets $g^*$ with high probability, where $S_n = \{(\boldsymbol{x}_i, y_i)\}_{i=1}^n$ defined in Assumption 1. Furthermore, training predictor $f$ only using informative data selected by $\widehat{g}$ improves selective risk.

**Remark 1.** *The toolkit we use in the proof is a Bernstein-type inequality for fast generalization rate under margin condition (Massart & Nédélec, 2006; Van Erven et al., 2015; Li & Liu, 2021). We also provide an information theoretic lower bound construction in section A.2 to show our selector risk bound is minimax-optimal. Our construction of the lower bound is motivated from (Ehrenfeucht et al., 1989; Blumer et al., 1989) and Le Cam's method (Yu, 1997). Due to space constraints, detailed proofs of our theorems are provided in the Appendix. To avoid lengthy technical definitions, we only present a light version of our results in the format of finite hypothesis class. The extension to VC-class using Local Rademacher Average tools (Bartlett et al., 2005) is shown in section B.*

**Theorem 1** (Minimax-Opitmal Selector Risk Bound)**.** *Let $S_n = \{(\boldsymbol{x}_i, y_i)\}_{i=1}^n$ be i.i.d sample from Data Generative Process described in Definition 1 under Assumption 1, with $f^*(\cdot) \in \mathcal{F}$ and $g^*(\cdot) \in \mathcal{G}$, $|\mathcal{F}| < \infty$ $|\mathcal{G}| < \infty$. Given $\bar{\lambda}$, let $\beta \in \left[\frac{3-2\bar{\lambda}}{1+2\bar{\lambda}} + \bar{\lambda}, \min(\frac{3+2\bar{\lambda}}{1-2\bar{\lambda}} - \frac{\bar{\lambda}}{1-4\bar{\lambda}^2}, 10)\right]$. For any $\widehat{f}(\cdot) \in \mathcal{F}$, let $\widehat{g} = \underset{g \in \mathcal{G}}{\arg\min}\, R_{S_n}(g; \widehat{f}, \beta)$.*

*Then for any $\varepsilon > 0$, $\delta > 0$ such that the following holds: For $n \geq max\{\frac{32\beta^2 \log(\frac{|\mathcal{G}|}{\delta})}{\bar{\lambda}\varepsilon}, \frac{24\beta \log(\frac{|\mathcal{F}|}{\delta})}{\varepsilon}\}$, and for $\widehat{f}$ that satisfies one of the following condition:*

- *For any $\widehat{f}(\cdot) \in \mathcal{F}$ such that $\mathbb{E}_{\boldsymbol{x}}[\widehat{f}(\boldsymbol{x}) \neq f^*(\boldsymbol{x})] \leq \frac{\varepsilon}{8\beta}$ with prob at least $1 - \delta$,*

- *For any $\widehat{f}(\cdot) \in \mathcal{F}$ such that $\mathbb{E}_{\boldsymbol{x}}[\widehat{f}(\boldsymbol{x}) \neq f^*(\boldsymbol{x}) | \boldsymbol{x} \in \Omega_I] \leq \frac{\varepsilon}{8\beta\alpha}$ with prob at least $1 - \delta$,*

*The following holds with probability at least $1 - 2\delta$:*

$$R(\widehat{g}; f^*, \beta) - R(g^*; f^*, \beta) \leq \varepsilon$$

**Remark 2.** *The assumption that $\mathbb{E}_{\boldsymbol{x}}[\widehat{f}(\boldsymbol{x}) \neq f^*(\boldsymbol{x})] \leq \varepsilon$ could be achieved via an ERM on classification loss $\sum_{i=1}^{n} \mathbb{1}\{f(\boldsymbol{x}_i) \neq y_i\}$ under some margin condtions (Bousquet, 2004; Massart & Nédélec, 2006; Bartlett et al., 2005). In practice, one can also apply some methods beyond ERM to obtain $\widehat{f}$ (Namkoong & Duchi, 2017). In particular, in the case $\lambda = \frac{1}{2}$, the data in support $\Omega_U$ is un-learnable as $y$ are purely random. While approximating $f^*$ on the full support is not possible in general, one can control the conditional risk $\mathbb{E}_{\boldsymbol{x}}[\widehat{f}(\boldsymbol{x}) \neq f^*(\boldsymbol{x})|\boldsymbol{x} \in \Omega_I]$ via a standard ERM schema (see proof in appendix Section A.7). We stress that Theorem 1 holds for any classifier that is close to $f^*$, even the case where $\widehat{f}$ and $\widehat{g}$ are trained on the same dataset.*

**Corollary 1** (Recovering $g^*$). *Given conditions in Theorem 1, if we choose $\beta = 3$, we have:*

$$\mathbb{E}_{\boldsymbol{x}}[\mathbb{1}\{\widehat{g}(\boldsymbol{x}) \neq g^*(\boldsymbol{x})\}] \leq \frac{4\varepsilon(1 + 2\bar{\lambda})}{\bar{\lambda}} \tag{6}$$

Corollary 1 suggests that by minimizing the empirical version of the loss from Definitions 3, one can recover $g^*$ in a PAC fashion. The theoretical guarantee holds even under a very challenging case where $\alpha > 0.5$ and $\bar{\lambda} = \frac{1}{2}$, .e.g, the majority of the data has purely random labels. The analysis of the selector loss (Theorem 1) relies on the quality of the classifier $\widehat{f}$. But since we know that $\widehat{g}$ is able to abstain from uninformative data, we can retrain $\widehat{f}$ beyond standard ERM, with upweighted informative data, therefore improving the accuracy of $\widehat{f}$. Such circular logic naturally leads to a alternating risk minimization schema. We formulate such schema in equations (8) and (9). Next theorem studies the risk bounds of classifiers by minimizing the empirical risk conditional on the samples selected by $\widehat{g}$. The risk bound improves the conditional risk compared to the standard empirical risk minimization method up to some problem-dependent constant.

**Theorem 2** (Joint Risk Bounds ). *Let $S_n\{(\boldsymbol{x}_i, y_i)\}_{i=1}^{n}$ be i.i.d samples from the Data Generative Process described in Definition 1 under Assumption 1, with $f^*(\cdot) \in \mathcal{F}$ and $g^*(\cdot) \in \mathcal{G}$, $|\mathcal{F}| < \infty, |\mathcal{G}| < \infty$ and $\lambda(x) < \frac{1}{2} - h, h > 0$. For any $\epsilon > 0$ and $0 < \delta < 1$, if $n \geq 64 \max\left\{\frac{\log(\frac{|\mathcal{G}|}{\delta})}{\bar{\lambda}^2\varepsilon}, \frac{\log(\frac{|\mathcal{F}|}{\delta})}{h^2\varepsilon}\right\}$, and:*

$$\text{ERM Classifier: } \widehat{f} = \underset{f \in \mathcal{F}}{\arg\min} \frac{1}{n} \sum_{i=1}^{n} \mathbb{1}\{f(\boldsymbol{x}_i) \neq y_i\} \tag{7}$$

$$\text{ERM Selector: } \widehat{g} = \underset{g \in \mathcal{G}}{\arg\min} R_{S_n}(g; \widehat{f}, 3) \tag{8}$$

$$\text{Subset-ERM Classifier: } \widetilde{f} = \underset{f \in \mathcal{F}}{\arg\min} \frac{1}{n} \sum_{i=1}^{n} \mathbb{1}\{\widehat{g}(\boldsymbol{x}_i) > 0\}\mathbb{1}\{f(\boldsymbol{x}_i) \neq y_i\} \tag{9}$$

*the following inequalities hold with probability at least $1 - 3\delta$:*

$$\mathbb{E}_{\boldsymbol{x}}[\widehat{f}(\boldsymbol{x}) \neq f^*(\boldsymbol{x})] \leq \varepsilon \tag{10}$$

$$\mathbb{E}_{\boldsymbol{x}}[\mathbb{1}\{\widehat{g}(\boldsymbol{x}) \neq g^*(\boldsymbol{x})\}] \leq 4\varepsilon \tag{11}$$

$$\mathbb{E}_{\boldsymbol{x}}[\widetilde{f}(\boldsymbol{x}) \neq f^*(\boldsymbol{x})|g^*(\boldsymbol{x}) = 1] \leq \frac{h^2\varepsilon}{\alpha} \tag{12}$$

Theorem 2 implies the convergence of the alternating minimization procedure between equation (8) and (9) by setting $\widehat{f} := \widetilde{f}$, due to the fact that equation (12) satisfies the pre-requisite condition on $\widehat{f}$ in Theorem 1. Another take away of Theorem 2 is an improved selective risk of classifier resulting from training only with samples picked out by the selector. To see this, we note that Massart Noise condition is satisfied with margin $\frac{h}{2}$ under the assumption that $\lambda(\boldsymbol{x}) < \frac{1}{2} - h$. The risk bound under standard margin condition in equation (10) follows from a standard result (see Section 5.2 in (Bousquet, 2004)). Indeed, equation (10) implies the following risk bounds conditional on $\Omega_I$:

$$\mathbb{E}_{\boldsymbol{x}}[\widetilde{f}(\boldsymbol{x}) \neq f^*(\boldsymbol{x})|g^*(\boldsymbol{x}) = 1] \leq \frac{\varepsilon}{\alpha}. \tag{13}$$

Which is improved by a problem-dependent constant factor, of order $h^2$, compared to equation (12). While risk bound in equation (10) is minimax optimal in general due to (Massart & Nédélec, 2006), the problem-dependent structure introduced in Definition 1 allows tighter risk bounds on sub-regions. The refined *selective* risk could be achieved by a weighted-ERM, with binary weights $\{0,1\}$ given by selectors.

**Remark 3.** *The analysis presented in (Cortes et al., 2016) suggests that the generalization bound exhibits a rate of $O(1/\sqrt{n})$. This risk bound covers scenarios where there is no presumption of a ground truth model, nor any gap on the label noise ratio to distinguish informative and uninformative data. In comparison, the risk bounds established in Theorem 2 indicate a faster, mini-max optimal rate of $O(1/n)$ by capitalizing on the structural characteristics of the underlying data generation process. This contrast in assumptions stems from the motivation behind selective learning. While in the works of (Cortes et al., 2016; Geifman & El-Yaniv, 2019), the selective loss is devised with a focus on coverage ratio, i.e., optimizing for higher precision (selective loss) by trading coverage ratio, our approach is tailored to distinguish data that is inherently unlearnable and unpredictable. This discrepancy leads to an alternative theoretical outcome. The analysis in (Cortes et al., 2016) primarily centers on selective risk, whereas our theoretical analysis places emphasis on the selector's capability to differentiate between informative and uninformative data without adjusting the rejection cost imposed by a human.*

---

**Algorithm 1** `Iterative Soft Abstain` (**ISA**)

---
1: **Input:** Dataset $S_n = \{(\boldsymbol{x}_1, y_1), ..., (\boldsymbol{x}_n, y_n))\}$, weight parameter:$\beta$, random initial classifier $\hat{f}^0$ and selector $\hat{g}^0$, number of iterations $T$
2: **for** $t \leftarrow 1, \cdots, T$ **do**
3:     Optimize loss to update predictor $\hat{f}^t : \frac{1}{n} \sum_{i=1}^{n} \hat{g}^t(\boldsymbol{x}_i)\{y_i \log(\hat{f}^t(\boldsymbol{x}_i)) + (1 - y_i) \log(1 - \hat{f}^t(\boldsymbol{x}_i))\}$
4:     Approximate the 'pseudo-informative label' : $z_i^t = \mathbb{1}\{\mathbb{1}\{\hat{f}^t(\boldsymbol{x}_i) > \frac{1}{2}\} = y_i\}$
5:     Optimize loss to update selector $g^t : \sum_{i=1}^{n} \{z_i^t \log(\hat{g}^t(\boldsymbol{x}_i)) + \beta(1 - z_i^t) \log(1 - \hat{g}^t(\boldsymbol{x}_i))\}$
6: **end for**
7: **Output:** $\hat{f}^T, \hat{g}^T$

---

## 6   A practical algorithm

Motivated by our theoretical analysis, we propose a practical algorithm that shares a spirit similar to the selector loss. From a computational standpoint, we replace the binary loss by cross-entropy loss instead and require that both $f$ and $g$ have continuous-valued output, ranging between 0 and 1 instead of binary output. The label $y$ also needs to be processed so that the values are in the $\{0,1\}$ range.

Following alternating steps given in equation (8) and (9), the practical algorithm trains both predictor and selector in an iterative manner. To accelerate the training, instead of applying an alternating minimization schema, we relax the requirement for minimization oracles by stochastic gradient updates. During the joint optimization process, the predictor is counting on the selector to upweight informative data. By putting more effort on the informative data, we wish to improve the performance of the predictor on informative data, as in equation 12. Algorithm 1 shows the logic above. A pictorial example of Algorithm 1's performance can be found in Figure 7 in the Appendix.

## 7   Experiments

In this section, we test the efficacy of our practical algorithm (Algorithm 1) on both publicly-available and semi-synthetic datasets. The code for reproducing the results could be found in `https://github.com/morganstanley/MSML/tree/main/paper/Learn_to_Abstain`. The empirical study aims to answer the following questions:

$Q_1$ : *How does Algorithm 1 compare to baselines on semi-synthetic datasets in recovering ground truth selector $g^*$?*

The results are presented in Figure 2, 3 and 4, 5. We conducted semi-synthetic experiments to evaluate the ability of each baseline in recovering the $g^*$ given different values for the noise ratio gap $\bar{\lambda}$ and sample size constraints. One can see our proposed method outperforms all listed baselines in this particular task.

$Q_2$ : *Does our algorithm outperform ERM on selected data points ?*

One important motivation behind selective learning is to find a model that outperforms the ERM on the informative part of the data. This is a challenging task as ERM is minimax optimal in general setting(Massart & Nédélec, 2006; Bartlett & Mendelson, 2006). We show in Table 1 that our proposed algorithm does exploit additional problem structure and achieves lower risk on selected datapoints.

$Q_3$ : *How does Algorithm 1 work on real world datasets compared to selective learning baselines?*

On real world datasets, Algorithm 1 consistently shows competitive/superior performance against other baselines in the low coverage regime, e.g., the proportion of data chosen by the selector being below 20%. These empirical results suggest that that our method is good at picking out strongly informative data. This observation supports theorem 2.

**Baselines.** We compare our method to recently proposed selective learning algorithms. (1) **SelectiveNet** (Geifman & El-Yaniv, 2019), which integrates an extra neuron as a data selector in the output layer and also introduces a loss term to control the coverage ratio; (2) **DeepGambler** (Liu et al., 2019), which also maintains an extra neuron for abstention and uses a doubling-rate-like loss term (i.e., gambler loss) to train the model. (3) **Adaptive** (Huang et al., 2020) uses the moving average confidence of classifier as the soft-label for the training and an extra output neuron is used to indicate abstention; (4) **Oneside** (Gangrade et al., 2021b) formulates selective classification problems as multiple one-sided prediction problems where each data class's coverage is maximized under error constrains. The implemented classifier is trained to optimize a relaxed min-max loss. The selection then is performed according to the classifier's confidence; (5) We also create a heuristic baseline that selects data using the model prediction confidence, which we refer to as **Confidence**. The intuition behind this heuristic baseline is that informative data should have higher confidence compared to uninformative data.

**Our Algorithm: ISA and ISA-V2.** Several specific implementation details need to be mentioned. First of all, we empirically get better performance if we average $z^t$ (in Algorithm 1) over past 10 epochs instead of using $z^t$ from the most recent epoch. Second, we use a rolling window average of past 10 epochs of $g$ for sample weights instead of only using current epoch's output. Furthermore, inspired by existing literature, we empirically get better performance in real-world datasets if we allow both the classifier model and the selector model to use the same backbone network to share data representations. We add an extra output neuron to the classifier model to achieve this goal. Finally, we use focal loss Lin et al. (2017) in order to deal with imbalance of informative versus uninformative data. The selector focal loss has the following format:

$$L(z_i, \hat{g}_i) = \sum_{i=1}^{n} \left\{ -\left[ \mathbb{1}\{z_i\}(1 - \hat{g}_i) \log \hat{g}_i + \beta \mathbb{1}\{1 - z_i\} \hat{g}_i \log (1 - \hat{g}_i) \right] \right\} \tag{14}$$

where $z_i = \mathbb{1}\{\hat{f}_i = y_i\}$ is the classifier $\hat{f}$'s correctness on ith input. We use $\hat{g}_i$ as the abbreviation for the ith entry in $\hat{g}_i(\boldsymbol{x}_i)$.

We will denote as **ISA** the modified version of Algorithm 1 that adopts the rolling average and focal loss, and **ISA-V2** as the version that adopts all 4 relaxations. Notice that both ISA and ISA-V2 can be easily applied in multi-class scenarios in practice.

**Experiment Details.** We use a lightweight CNN for MNIST+Fashion. Its architecture is given in Table 5 in section D.1. We use ResNet18 (He et al., 2016b) for SVHN. For all of our synthetic experiments, we use Adam optimizer with learning rate 1e-3 and weight decay rate 1e-4. We use batch size 256 and train 60 epochs for MNIST+Fashion and 120 epochs for SVHN. The learning rate is reduced by 0.5 at epochs 15, 35 and 55 for MNIST+Fashion and is reduced by 0.5 at epochs 40, 60 and 80 for SVHN. We repeat each experiment 3 times using seeds 80, 81 and 82.

Regarding hyper-parameters for each baseline, we mainly follow the setting given by the original paper. Except for DeepGambler in MNIST+Fashion, which the original paper does not contemplate. We set the pay-off $o = 1.5$ instead of 2.6, which gives better performance for DeepGambler. For Oneside, since we are not maximizing the coverage (the optimal coverage is fixed in our problem), we fix the Lagrangian multiplier $\mu$ to be 0.5 and use $t$ as the score to compute average precision. We made our best effort to implement this algorithm, given that the authors have not released the original code. We list all the hyper-parameters in Table 6 in section D.1.

We conduct all our experiments using Pytorch 3.10 (Paszke et al., 2019). We execute our program on Red Hat Enterprise Linux Server 7.9 (Maipo) and use NVIDIA V100 GPU with cuda version 12.1.

## 7.1 Experiments Using Semi-Synthetic Data for $Q_1$

**Dataset Construction**: We explicitly control the support of informative/uninformative data. For MNIST+Fashion-MNIST dataset, images from MNIST are defined to be uninformative, while images from Fashion-MNIST are set to be informative. For SVHN(Netzer et al., 2011) dataset, class 5-9 are set to be uninformative and class 0-4 are set to be informative. Datasets are constructed with different values of informative data fraction $\alpha$ and label noise ratio gap $\bar{\lambda}$ driving the noisy generative process. We inject label noise accordingly to Definition 1 by setting $\lambda(\boldsymbol{x}) = \bar{\lambda}$. Note that when we vary $\alpha$, we only constrain the sample size in the training set, while in the testing set we always use the complete testing set for performance evaluation.

We present two sets of experiments. In the first experiment, we set $\bar{\lambda} = 0.5$, which gives realizable informative data and completely randomly shuffled uninformative data. We fix the number of uninformative data points and increase the number of informative ones as a proxy for different value of $\alpha$ in the Definition 1. Specifically, we run our experiments with $\frac{\sharp\text{Informative Data}}{\sharp\text{Uninformative Data}} = \frac{\alpha}{1+\alpha} \in \{1.0, 0.75, 0.5, 0.25\}$. Furthermore, we conduct our experiments in both a complete dataset setting and a sparse dataset setting to test each algorithm's sample efficiency. In a complete dataset setting, we use 100% of all informative data and uninformative data, whie in a sparse dataset setting, we use only 25% of samples.

In the second set of experiments, we inject 10%, 20% and 30% proportion of uniform label noise into the informative part and also inject 80%, 70% and 60% uniform label noise into the uninformative part.

**Evaluation Metrics.** Every baseline will generate a score besides their classification output to decide if the input should be abstained. For SLNet, DeepGambler, Adaptive and ISA-V2, the score is given by the extra neuron. For Confidence and Oneside, such score is its maximum probabilistic output. For ISA, the score is given by the selector model. In the semi-synthetic experiment, we can calculate the **average precision (AP)** using the selection score and ground-truth informative versus uninformative binary label to test each baseline's ability in recovering $g^*$. In our study, we especially focus on the case where $\alpha$ is small and informative data is the minority part of the dataset. In this situation, there is severe class imbalance in the latent informative/uninformative labeling. AP is preferred over other metrics, like F1 or AUC, in this scenario (Saito & Rehmsmeier, 2015). In Table 2, Table 3 and Table 4, we use transformation $-\log(1-x)$ to make the performance difference visually distinguishable.

**Results and Discussion.** There are 3 empirical observations we can get from Figure 2, 3, 4, where we manifest the performance difference though the transformation $-log(1-\text{Average Precision})$ for differentiation purpose. More detailed results can be found in appendix section D.2. Firstly, the proposed ISA method outperforms all baselines under most of the scenarios according to the average precision criterion, which supports the superiority of the proposed method in recovery $g^*$. Secondly, we observe that our method's performance is more stable and suffers less deterioration as the data size becomes more limited or the label noise becomes stronger. Finally, in many cases, Confidence, which simply uses the confidence of a model trained with vanilla cross entropy loss, is a competitive baseline. Our proposed method outperforms Confidence in most scenarios.

Figure 2: Average Precision (AP) ↑ v.s Different $\alpha$. Numerical results in Table 7.

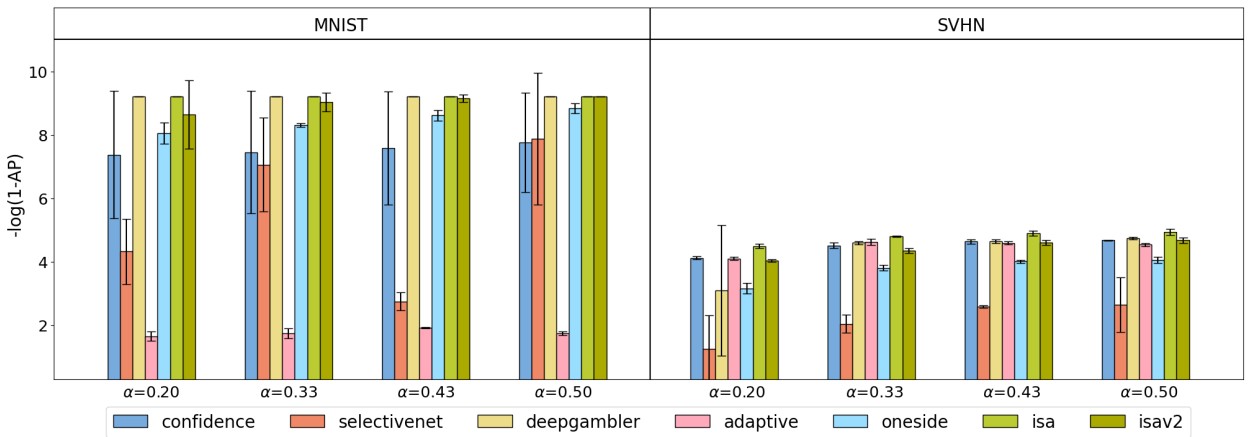

Figure 3: Average Precision (AP) ↑ v.s Different $\alpha$ - 25% Samples Size. Numerical results in Table 8.

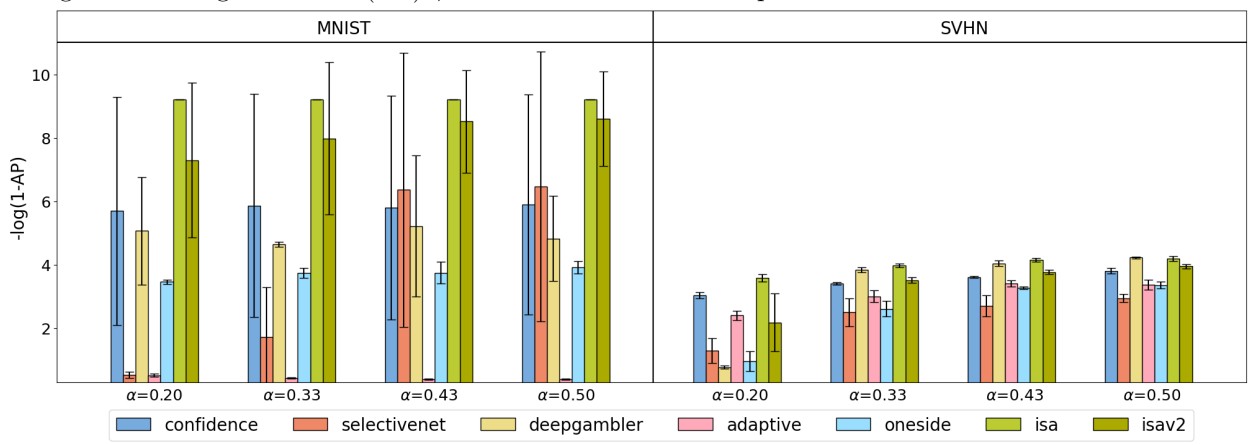

Figure 4: Average Precision (AP) ↑ v.s. Different $\lambda$ . Numerical results in Table 9.

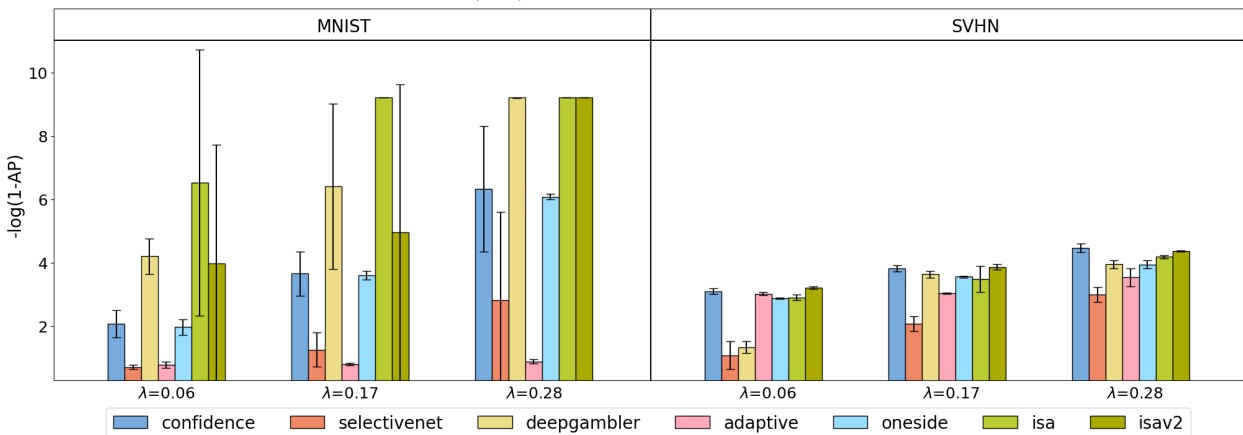

## 7.2 Experiments Using Semi-Synthetic Data for $Q_2$

**Evaluation Metrics**: The construction of informative/uninformative data follows Section 7.1. In this section, we want to measure the classification performance of the proposed method conditioning on selected data points, by looking at the **selective risk (SR)** metric. Its formal definition is given by $SR_\tau = \frac{1}{n} \sum_i^n \mathbb{I}\{\arg\max \hat{f}(\boldsymbol{x}_i) \neq y_i | \mathbb{1}\{\hat{g}(\boldsymbol{x}_i) > q\}\}$, where the quantity $q$ is used to control the coverage. The confidence method could be viewed as a proxy for ERM method with following adjustments: 1) replacing cross-entropy as surrogate loss for binary classification loss 2) replacing selector $\mathbb{1}\{\hat{g}(\boldsymbol{x}) \geq q\}$ by using estimated margin based selection rule $\mathbb{1}\{|\hat{f}_{ERM}(\boldsymbol{x}) - 1/2| \geq q\}$. Coverage is defined as the proportion of selected data to the whole dataset. In this experiment, we always calculate $SR$ at the ground-truth coverage $\alpha = 0.5$. It is the proportion of informative data that we mixed into the testing set. Ideally, we hope that the selector can select all but only informative data, which should give the lowest risk given the coverage level.

Table 1: SR ↓. ERM v.s ISA

| Dataset | Method | 100% Sample | | | 25% Sample | | |
|---|---|---|---|---|---|---|---|
| | | $\tau_I = 0.3$ $\tau_U = 0.6$ | $\tau_I = 0.2$ $\tau_U = 0.7$ | $\tau_I = 0.1$ $\tau_U = 0.8$ | $\tau_I = 0.3$ $\tau_U = 0.6$ | $\tau_I = 0.2$ $\tau_U = 0.7$ | $\tau_I = 0.1$ $\tau_U = 0.8$ |
| MNIST+Fashion | Confidence | $0.387 \pm 0.001$ | $0.293 \pm 0.005$ | $0.188 \pm 0.005$ | $0.469 \pm 0.002$ | $0.362 \pm 0.002$ | $0.235 \pm 0.002$ |
| | ISA | $\mathbf{0.376 \pm 0.004}$ | $\mathbf{0.280 \pm 0.004}$ | $\mathbf{0.186 \pm 0.003}$ | $\mathbf{0.446 \pm 0.002}$ | $\mathbf{0.338 \pm 0.005}$ | $\mathbf{0.233 \pm 0.004}$ |
| | ISA-V2 | $0.378 \pm 0.008$ | $0.281 \pm 0.001$ | $0.192 \pm 0.001$ | $0.487 \pm 0.022$ | $0.355 \pm 0.008$ | $0.254 \pm 0.007$ |
| SVHN | Confidence | $0.409 \pm 0.006$ | $0.324 \pm 0.006$ | $0.227 \pm 0.007$ | $\mathbf{0.526 \pm 0.008}$ | $0.416 \pm 0.012$ | $0.295 \pm 0.007$ |
| | ISA | $\mathbf{0.391 \pm 0.003}$ | $\mathbf{0.310 \pm 0.018}$ | $\mathbf{0.217 \pm 0.003}$ | $0.554 \pm 0.011$ | $\mathbf{0.365 \pm 0.016}$ | $\mathbf{0.256 \pm 0.010}$ |
| | ISA-V2 | $0.455 \pm 0.006$ | $0.357 \pm 0.010$ | $0.243 \pm 0.003$ | $\mathbf{0.516 \pm 0.007}$ | $0.430 \pm 0.014$ | $0.307 \pm 0.027$ |

**Results and discussion.** In Table 1, we present the SR for Confidence (ERM) and the proposed ISA. Following the setting in section 7.1, we inject different levels of uniform label noise into uninformative data and informative data separately, denoted by $\tau_I$ and $\tau_U$, corresponding to label noise ratio for informative and uninformative data respectively. Parameter $\bar{\lambda}$ in Definition 1 can be calculated accordingly: $\bar{\lambda} = \frac{0.9 - \tau_I}{1.8} = \frac{\tau_U}{1.8}$.

There are two main empirical observations from Table 1. First, ISA consistently maintains smaller SR than Confidence (ERM) across different scenarios. This result is consistent with Theorem 2, suggesting that upweighting informative data leads to improved risk on informative data. Second, we observe that both methods exhibit performance deterioration as the noise gap becomes smaller, which is suggested by the sample complexity in our theorem. Our theorem suggests that the sample complexity increases as the label noise ratio gap $\bar{\lambda}$ decreases. When $\bar{\lambda}$ vanishes, informative date becomes less distinguishable from the uninformative data, and thus learning $g$ becomes more challenging.

## 7.3 Experiments Using Real-world Data for $Q_3$

**Dataset.** In this section, we report our empirical study on 3 publicly-available datasets: (1) Oxford realized volatility (Volatility) dataset (Heber et al., 2009), (2) breast ultrasound images (BUS) (Al-Dhabyani et al., 2020), and (3) lending club dataset (LC) (Lending Club, 2007). The experiments aim to demonstrate the potential of the proposed algorithm in real-world application and its advantages in selecting useful information out of noisy dataset. The data description and respective train-test splits are presented in Table 11 in section D.1.

**Experiment Details.** We use the same network architectures, software and hardward as we do in section 7.1-7.2. For Volatility and LC, we use the same Adam optimizer with learning rate (1e-3), weight decay rate (1e-4) and batch size 256. For BUS, due to its limited sample size, we use smaller batch size (16) and reduce learning rate (1e-4) accordingly. For all three datasets, we train each algorithm for 50 epochs and reduce the learning rate by half at epochs 15 and 35. Each experiment is repeated 6 times with random seeds 77, 78, 79, 80, 81 and 82.

Specifically, for DeepGambler the default hyper-parameter $o = 2.2$ gives unreasonably poor performance on LC and BUS. We find that setting $o = 1.5$ gives the best performance and hence we use this value for DeepGambler in these two datasets. For other baselines as well as our method, we use consistent hyper-parameters across all experiments. They are listed in Table 4 in section D.1.

**Evaluation Metric.** Unlike synthetic experiments, in the real-world datasets, the ground-truth binary labels that distinguish whether data is informative/uninformative are not available. Hence metrics like precision/recall are not applicable. Instead, we report the selective risk of each algorithm given different coverage levels. Specifically, we pick testing data points that have top% coverage selective confidence given by each selector, and calculate the testing selective risk at different coverage levels accordingly.

**Results and Discussion.** From Table 2, we can see that the proposed method gains competitive performance against other baselines at low coverage levels. This suggests that our method is especially good at picking out strongly informative data. Such low risk regime can be captured by our selector loss, leading to lower selective risk, which is consistent with the conclusion in Theorem 2.

Table 2: Real-World Experiments: Selective Risk v.s Coverage

| Dataset | Coverage | Confidence | SLNet | DeepGambler | Adaptive | Oneside | ISA | ISA-V2 |
|---|---|---|---|---|---|---|---|---|
| Volatility | 0.001 | **0.000 ± 0.000** | 0.200 ± 0.000 | 0.267 ± 0.121 | **0.000 ± 0.000** | 0.050 ± 0.055 | **0.000 ± 0.000** | **0.000 ± 0.000** |
| | 0.100 | 0.097 ± 0.008 | 0.122 ± 0.003 | 0.328 ± 0.015 | 0.168 ± 0.008 | 0.113 ± 0.006 | 0.091 ± 0.008 | **0.083 ± 0.004** |
| | 0.200 | **0.133 ± 0.004** | 0.198 ± 0.002 | 0.326 ± 0.009 | 0.225 ± 0.012 | 0.159 ± 0.003 | **0.136 ± 0.002** | **0.127 ± 0.008** |
| | 0.500 | **0.218 ± 0.004** | 0.316 ± 0.003 | 0.327 ± 0.005 | 0.290 ± 0.004 | 0.239 ± 0.003 | 0.227 ± 0.002 | **0.225 ± 0.004** |
| | 0.900 | **0.309 ± 0.001** | 0.332 ± 0.003 | 0.327 ± 0.001 | 0.321 ± 0.001 | 0.314 ± 0.001 | **0.312 ± 0.002** | **0.310 ± 0.001** |
| | 0.999 | **0.327 ± 0.003** | 0.333 ± 0.003 | 0.327 ± 0.001 | 0.329 ± 0.001 | 0.332 ± 0.002 | 0.328 ± 0.001 | **0.324 ± 0.002** |
| BUS | 0.001 | **0.026 ± 0.032** | **0.167 ± 0.408** | **0.000 ± 0.000** | **0.167 ± 0.408** | 0.052 ± 0.036 | **0.000 ± 0.000** | **0.000 ± 0.000** |
| | 0.100 | **0.026 ± 0.032** | 0.396 ± 0.156 | 0.156 ± 0.147 | 0.083 ± 0.076 | 0.052 ± 0.036 | **0.020 ± 0.031** | **0.010 ± 0.026** |
| | 0.200 | **0.026 ± 0.032** | 0.422 ± 0.134 | 0.151 ± 0.057 | 0.089 ± 0.094 | 0.052 ± 0.036 | **0.021 ± 0.026** | **0.005 ± 0.013** |
| | 0.500 | 0.038 ± 0.029 | 0.432 ± 0.062 | 0.184 ± 0.046 | 0.135 ± 0.140 | 0.052 ± 0.036 | 0.058 ± 0.019 | **0.017 ± 0.016** |
| | 0.900 | **0.085 ± 0.019** | 0.393 ± 0.039 | 0.133 ± 0.035 | 0.200 ± 0.109 | **0.093 ± 0.023** | **0.098 ± 0.018** | **0.095 ± 0.022** |
| | 0.999 | **0.109 ± 0.016** | 0.382 ± 0.040 | **0.121 ± 0.032** | 0.114 ± 0.026 | 0.219 ± 0.107 | **0.109 ± 0.013** | **0.118 ± 0.024** |
| LC | 0.001 | 0.829 ± 0.052 | 0.410 ± 0.142 | 0.362 ± 0.058 | 0.357 ± 0.072 | 0.106 ± 0.036 | **0.061 ± 0.012** | 0.219 ± 0.068 |
| | 0.100 | 0.204 ± 0.005 | 0.370 ± 0.102 | 0.419 ± 0.031 | 0.284 ± 0.023 | 0.207 ± 0.036 | **0.151 ± 0.007** | **0.151 ± 0.021** |
| | 0.200 | **0.216 ± 0.008** | 0.393 ± 0.077 | 0.419 ± 0.023 | 0.265 ± 0.014 | **0.228 ± 0.038** | **0.214 ± 0.006** | **0.212 ± 0.019** |
| | 0.500 | 0.312 ± 0.006 | 0.421 ± 0.030 | 0.414 ± 0.011 | **0.241 ± 0.007** | 0.305 ± 0.030 | 0.333 ± 0.005 | 0.362 ± 0.011 |
| | 0.900 | 0.385 ± 0.004 | 0.399 ± 0.009 | 0.418 ± 0.004 | **0.229 ± 0.004** | 0.373 ± 0.021 | 0.389 ± 0.006 | 0.427 ± 0.009 |
| | 0.999 | 0.398 ± 0.004 | 0.399 ± 0.010 | 0.421 ± 0.004 | **0.231 ± 0.003** | 0.386 ± 0.019 | 0.395 ± 0.007 | 0.427 ± 0.008 |

## 7.4 Ablation Study

In this section we present ablation studies about 1) the sensitivity of the algorithm's performance to the choice of hyper-parameters and 2) aforementioned practical implementations. We first present the results on MNIST+Fashion given different hyper-parameter combinations. In section 7.1 we set $\beta = 3.0$, $\Delta T = 1$ and pre-train epochs to be 10. In this section, we vary each of them one-by-one while fix the rest of them. The results are presented in Figure 6 in Appendix. We can see that the proposed algorithm's performance is robust against the choice of hyper-parameters. We next present an ablation study on the aforementioned practical implementations in Table 3. Recall that we have the following relaxations: (1) moving average estimation of sample weight (MAW); (2) moving average of soft-label for informative data; (3) classifier and selector sharing the same backbone and the use of focal loss instead of cross-entropy. We systematically incorporate each module one at a time, evaluating their marginal contributions to the performance on SVHN in the high label noise setting ($\tau_I = 0.3$ and $\tau_U = 0.6$). Results shown in Table 3. For Algorithm 1, the best run can get $AP = 0.87$ but the vanilla algorithm is not stable and end up getting large variance in its performance. In comparison, the relaxed version get much more stable performance. The ablation study suggests that each relaxation strategy helps stabilizing the algorithm and make it robust against variance from stochastic approximation.

Table 3: Ablation Study on Implementation Relaxation.

| Method | AP | SR |
|---|---|---|
| Algorithm 1 | 0.778 ± 0.131 | 0.576 ± 0.034 |
| + MA-Weight | 0.763 ± 0.094 | 0.573 ± 0.019 |
| + MA-Soft (ISA) | 0.850 ± 0.020 | 0.554 ± 0.011 |
| + Focal Loss (ISA-V2) | 0.906 ± 0.000 | 0.516 ± 0.007 |

## 8 Conclusion and Future Work

In this work, we take the first step towards principled learning in domains where a lot of data is naturally uninformative/highly noisy and should be discarded in both learning and inference stage. We propose a general noisy generative process that formally describes such setting. Supported by theoretical guarantees, a novel loss is designed for the training of the selector model. Based on this loss, we design a practical algorithm that jointly learns the predictor and selector. Empirical analysis demonstrates the effectiveness of our method. There are several directions for future work:

- In the current framework, the fundamental principle of proposed method is distinguishing the uninformative data points and to discard them during the training process. While this approach may enhance the statistical efficiency and risk associated with informative data, it may under-utilize the potential of data from a representation learning perspective - for instance, while the uninformative labels are useless by definition, the discarded samples themselves may still be helpful for representation learning. This can be achieved through methods similar to those presented in studies like (Sohn et al., 2020; Li et al., 2021).

  using methods similar to (Sohn et al., 2020; Li et al., 2021).

- The Noisy Generative Process can be potentially generalized to solve different problems, such as active learning (Cohn et al., 1994) and out of distribution generalization (Arjovsky et al., 2019).

- Bridging the gap between Algorithm 1 and Theorem 2 presents two main challenges. Firstly, analyzing the disparity when minimizing cross-entropy as a surrogate for binary classification. We are optimistic that the $\mathcal{H}$-consistency bound framework introduced in (Awasthi et al., 2022) could be utilized for this purpose. Secondly, examining the use of Stochastic Gradient Descent (SGD) instead of the Empirical Risk Minimization (ERM) oracle in Theorem 2. In this regard, we believe that tools for analyzing SGD in bi-level optimization problems (Chen et al., 2021) could be applied.

We look forward to these extensions.

## 9 Acknowledgement

We thank anonymous reviewers for their constructive feedback, improving the quality of this work. This effort was partially supported by the Intelligence Advanced Research Projects Agency (IARPA) and Army Research Office (ARO) under Contract No. W911NF20C0038. Any opinions, findings, and conclusions in this paper are those of the authors only and do not necessarily reflect the views of our sponsors.

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

## A   Theoretical Results Details

In this appendix section, we present the missing proofs as well as additional theory results.

## A.1 Preliminaries

We describe the risk for selector loss on $(\boldsymbol{x}, y) \sim \mathcal{X} \times \mathcal{Y} \subset \mathbb{R}^d \times \{+1, -1\}$.

$$R(g; f, \beta) := \mathbb{E}_{\boldsymbol{x},y}\big[\beta \mathbb{1}\{g(\boldsymbol{x}) = 1\}\mathbb{1}\{f(\boldsymbol{x}) \neq y\} + \mathbb{1}\{g(\boldsymbol{x}) \neq 1\}\mathbb{1}\{f(\boldsymbol{x}) = y\}\big] \tag{15}$$

The choice of $\beta$ should ensure that given Bayes optimal classifier $f^*(\cdot)$, $g^*(\cdot)$ is the minimizer for the selector risk $R(g; f^*, \beta)$. We have that given $f^*$, the risk gap between any selector $g$ and $g^*$ is $R(g; f^*, \beta) - R(g^*; f^*, \beta)$ could be written as:

$$R(g; f^*, \beta) - R(g^*; f^*, \beta)$$

$$= \mathbb{E}_{\boldsymbol{x},y}\bigg[\beta \mathbb{1}\{g(\boldsymbol{x}) = 1\}\mathbb{1}\{g^*(\boldsymbol{x}) = 1\}\mathbb{1}\{f^*(\boldsymbol{x}) \neq y\} + \beta \mathbb{1}\{g(\boldsymbol{x}) = 1\}\mathbb{1}\{g^*(\boldsymbol{x}) \neq 1\}\mathbb{1}\{f^*(\boldsymbol{x}) \neq y\}$$

$$+ \mathbb{1}\{g(\boldsymbol{x}) \neq 1\}\mathbb{1}\{g^*(\boldsymbol{x}) = 1\}\mathbb{1}\{f^*(\boldsymbol{x}) = y\} + \mathbb{1}\{g(\boldsymbol{x}) \neq 1\}\mathbb{1}\{g^*(\boldsymbol{x}) \neq 1\}\mathbb{1}\{f^*(\boldsymbol{x}) = y\}\bigg]$$

$$- \mathbb{E}_{\boldsymbol{x},y}\bigg[\beta \mathbb{1}\{g(\boldsymbol{x}) = 1\}\mathbb{1}\{g^*(\boldsymbol{x}) = 1\}\mathbb{1}\{f^*(\boldsymbol{x}) \neq y\} + \beta \mathbb{1}\{g(\boldsymbol{x}) \neq 1\}\mathbb{1}\{g^*(\boldsymbol{x}) = 1\}\mathbb{1}\{f^*(\boldsymbol{x}) \neq y\} \tag{16}$$

$$+ \mathbb{1}\{g(\boldsymbol{x}) \neq 1\}\mathbb{1}\{g^*(\boldsymbol{x}) \neq 1\}\mathbb{1}\{f^*(\boldsymbol{x}) = y\} + \mathbb{1}\{g(\boldsymbol{x}) = 1\}\mathbb{1}\{g^*(\boldsymbol{x}) \neq 1\}\mathbb{1}\{f^*(\boldsymbol{x}) = y\}\bigg]$$

$$= \mathbb{E}_{\boldsymbol{x}}\bigg[\bigg\{\beta\Big(\frac{1}{4} + \frac{\lambda(\boldsymbol{x})}{2}\Big) - \frac{3}{4} + \frac{\lambda(\boldsymbol{x})}{2}\bigg\}\mathbb{1}\{g(\boldsymbol{x}) = 1\}\mathbb{1}\{g^*(\boldsymbol{x}) \neq 1\}\bigg]$$

$$+ \mathbb{E}_{\boldsymbol{x}}\bigg[\bigg\{\frac{3}{4} + \frac{\lambda(\boldsymbol{x})}{2} - \beta\Big(\frac{1}{4} - \frac{\lambda(\boldsymbol{x})}{2}\Big)\bigg\}\mathbb{1}\{g(\boldsymbol{x}) \neq 1\}\mathbb{1}\{g^*(\boldsymbol{x}) = 1\}\bigg]$$

Since $\lambda(\boldsymbol{x})$ is data dependent, to ensure that $R(g, f^*, \beta) \geq R(g^*; f^*, \beta)$ for all $g \in \mathcal{G}$, it suffices to pick $\beta\big(\frac{1}{4} + \frac{\lambda(\boldsymbol{x})}{2}\big) - \frac{3}{4} + \frac{\lambda(\boldsymbol{x})}{2} \geq 0$ and $\frac{3}{4} + \frac{\lambda(\boldsymbol{x})}{2} - \beta\big(\frac{1}{4} - \frac{\lambda(\boldsymbol{x})}{2}\big) \geq 0$. So we need $\beta \geq \sup_{\boldsymbol{x}} \frac{3 - 2\lambda(\boldsymbol{x})}{1 + 2\lambda(\boldsymbol{x})}$ and $\beta \leq \inf_{\boldsymbol{x}} \frac{3 + 2\lambda(\boldsymbol{x})}{1 - 2\lambda(\boldsymbol{x})}$ which implies that it suffices to pick $\beta \in \big[\frac{3 - 2\bar{\lambda}}{1 + 2\bar{\lambda}}, \frac{3 + 2\bar{\lambda}}{1 - 2\bar{\lambda}}\big]$.

Assuming $\lambda(\boldsymbol{x}) \geq \bar{\lambda}$, we pick $\beta$ within certain margin of the above interval: by picking $\big[\frac{3 - 2\bar{\lambda}}{1 + 2\bar{\lambda}} + \bar{\lambda}, \frac{3 + 2\bar{\lambda}}{1 - 2\bar{\lambda}} - \frac{\bar{\lambda}}{1 - 4\bar{\lambda}^2}\big]$ we have

$$R(g; f^*, \beta) - R(g^*; f^*, \beta) \geq \frac{\bar{\lambda}}{4(1 + 2\bar{\lambda})}\mathbb{E}_{\boldsymbol{x}}[\mathbb{1}\{g(\boldsymbol{x}) \neq g^*(\boldsymbol{x})\}] \tag{17}$$

Note that if $\bar{\lambda} = \frac{1}{2}$, the above interval for $\beta$ is $[2, \infty]$.

## A.2 Proof of Information Theoretic Lower Bound

In this section we quantify the hardness of recovering $g^*$, from an information theoretic perspective. We define a generative process that conforms with the noisy generative process defined in 1, and show a sample lower bound for finding a selector given samples generated from this process.

Let $\mathcal{X} : \{\tau \cdot \boldsymbol{e} | \boldsymbol{e} \in \{\boldsymbol{e}^1, ..., \boldsymbol{e}^d\}, |\tau| \leq 1\}$ where $e^j$ represents the $j$-th cannonical basis. So $\mathcal{X}$ consists of $\tau$-scaled basis vectors. Let $\mathcal{Y} = \{+1, -1\}$, samples are drawn from $\mathcal{X} \times \mathcal{Y} \subset \mathbb{R}^d \times \{+1, -1\}$. Let $\boldsymbol{w}$ be vector of ones, $\boldsymbol{w} = \mathbb{1}$. For our lower bound construction we define $f^*$ as follows for $x \in \mathcal{X}$.

$$f^*(\boldsymbol{x}) = 2\mathbb{1}\{\boldsymbol{w}^\top \boldsymbol{x} > 0\} - 1.$$

Let $\mathcal{G}$ be the hypothesis class that contains all functions $g : \mathcal{X} \to \{0, 1\}$. So $|\mathcal{G}| = 2^d$ and $\mathcal{G}$ contains $g^*(\boldsymbol{x})$. Let $\boldsymbol{\sigma} = (\sigma^1, \ldots, \sigma^d) \in \{+1, -1\}^d$ be a $d$-dimensional Rademacher vector, $\alpha \in (0, 1)$. We define $g^*_{\boldsymbol{\sigma}} \in \mathcal{G}$ as follows

$$g^*_{\boldsymbol{\sigma}}(\boldsymbol{x}) = \sum_{j=1}^{d} \mathbb{1}\{\boldsymbol{x}^\top \boldsymbol{e}^j \neq 0\}\big\{\mathbb{1}\{\|\boldsymbol{x}\| \geq 1 - \alpha\} \cdot (1 - \sigma^j)/2 + \mathbb{1}\{\|\boldsymbol{x}\| \leq \alpha\} \cdot (1 + \sigma^j)/2\big\}.$$

To clarify the definitions above, suppose for each $j = 1, \ldots, d$, $\boldsymbol{e}^j$ is the vector with the $j$th entry being one and the other entries zero. Let $\Omega^j : \{\boldsymbol{x}|\boldsymbol{x} = \tau \cdot \boldsymbol{e}^j\}$ be all the vectors in $\mathcal{X}$ with non-zero $j$th. Then for $\boldsymbol{x} \in \Omega^j$, we have that $f^*(\boldsymbol{x})$ is 1 if $\tau$ is positive and $-1$ if $\tau$ is negative. Moreover, $g^*(x)$ is $\frac{1-\sigma^j}{2}$ if $\|\tau\| \geq 1 - \alpha$, it is $\frac{1+\sigma^j}{2}$ if $\|\tau\| \leq \alpha$, and it is 0 otherwise. In other words, if $\sigma^j = -1$, the informative part of $\Omega^j$ is $\{\boldsymbol{x}|\|x\| \geq 1 - \alpha\}$ otherwise the informative part of $\Omega^j$ becomes $\{\boldsymbol{x}|\|x\| \leq \alpha\}$. Moreover, considering Definition 1, the domain $\mathcal{D}_\alpha$ is $\cup_{j=1}^d \Omega^j$. Let $S_{\sigma,n} = \{(\boldsymbol{x}_i, y_i)\}_{i=1}^n$ be generated from following process which we denote as $\mathcal{Q}$:

$$
\begin{aligned}
&\boldsymbol{\sigma} \sim Unif\{+1, -1\}^d \\
&g^*_{\boldsymbol{\sigma}}(\boldsymbol{x}) = \sum_{j=1}^d \mathbb{1}\{\boldsymbol{x}^\top \boldsymbol{e}^j \neq 0\}\big\{\mathbb{1}\{\|\boldsymbol{x}\| \geq 1 - \alpha\} \cdot (1 - \sigma^j)/2 \\
&\quad + \mathbb{1}\{\|\boldsymbol{x}\| \leq \alpha\} \cdot (1 + \sigma^j)/2\big\} \\
&\text{Generate } S = \{(\boldsymbol{x}_i, y_i)\}_{i=1}^n \text{ according to:} \\
&j \sim \begin{cases} j = 1, & \text{with prob } 1 - \frac{\varepsilon}{\bar{\lambda}} \\ j \sim Unif\{2, ..., d\} & \text{with prob } \frac{\varepsilon}{\bar{\lambda}}. \end{cases} \\
&\tau \sim Unif[-1, 1] \\
&\boldsymbol{x} = \tau \cdot e^j \\
&y = \begin{cases} f^*(\boldsymbol{x}), & \text{with prob } \frac{3}{4} + \frac{(2g^*(\boldsymbol{x})-1)\bar{\lambda}}{2} \\ -f^*(\boldsymbol{x}) & \text{with prob } \frac{1}{4} - \frac{(2g^*(\boldsymbol{x})-1)\bar{\lambda}}{2} \end{cases}
\end{aligned}
\tag{18}
$$

Process $\mathcal{Q}$ follows process 1. Let $\mathcal{A}$ be *any* (potentially randomized) algorithm that takes dataset $S_{\boldsymbol{\sigma},n}$ as input where $S_{\boldsymbol{\sigma},n}$ is generated from the process described in Equation 18. Let $\widehat{g}$ be the hypothesis ouput of algorithm $\mathcal{A}$, i.e. $\widehat{g}(\cdot) = \mathcal{A}(S_{\boldsymbol{\sigma},n})$. For a parameter $\beta$ we define the risk on algorithm $\mathcal{A}$ as

$$
R(\mathcal{A}(S_{\boldsymbol{\sigma},n}), \beta) = R(\widehat{g}(\boldsymbol{x}), f^*, \beta) = \mathbb{E}_{\boldsymbol{x},y}\big[\beta\mathbb{1}\{\widehat{g}(\boldsymbol{x}) = 1\}\mathbb{1}\{f^*(\boldsymbol{x}) \neq y\} + \mathbb{1}\{\widehat{g}(\boldsymbol{x}) \neq 1\}\mathbb{1}\{f^*(\boldsymbol{x}) = y\}\big]. \tag{19}
$$

**Theorem 3.** *Let $\beta$ and $\varepsilon$ be any real numbers where $\beta \in \left[\frac{3-2\bar{\lambda}}{1+2\bar{\lambda}} + \bar{\lambda}, \frac{3+2\bar{\lambda}}{1-2\bar{\lambda}} - \frac{\bar{\lambda}}{1-4\bar{\lambda}^2}\right]$ and $0 < \varepsilon \leq \bar{\lambda}$. For any algorithm $\mathcal{A}$ that takes $n$ samples $S_n$ from the noisy generative process as in Assumption 1 for some integer $n$ and outputs a selector $R(S_n)$ we have the following: If the risk gap is bounded by $\frac{\varepsilon}{8(1+2\bar{\lambda})}$, i.e.*

$$
\mathbb{E}_{S_n}[R(\mathcal{A}(S_n), f^*, \beta) - R(g^*, f^*, \beta)] \leq \frac{\varepsilon}{8(1 + 2\bar{\lambda})}
$$

*then $\mathcal{A}$ requires $\frac{\log(|\mathcal{G}|)}{\bar{\lambda}\varepsilon}$ many samples, i.e. $n \geq \frac{\log(|\mathcal{G}|)}{\bar{\lambda}\varepsilon}$.*

*Proof.* The lower bound construction is the process defined in Equation 18. Let $\widehat{g} = \mathcal{A}(S_n)$ be the output of $\mathcal{A}$. From equation (17) the risk gap $R(\widehat{g}, f^*, \beta) - R(g^*, f^*, \beta)$ averaged over $\boldsymbol{\sigma}$ and $S_{\boldsymbol{\sigma},n}$ can be written as follows.

$$
\begin{aligned}
&\mathbb{E}_{\boldsymbol{\sigma}}\mathbb{E}_{S_{\boldsymbol{\sigma},n}}[R(\widehat{g}, f^*, \beta) - R(g^*_{\boldsymbol{\sigma}}, f^*, \beta)] \\
&\geq \frac{\bar{\lambda}}{4(1 + 2\bar{\lambda})}\mathbb{E}_{\boldsymbol{\sigma}}\mathbb{E}_{S_{\boldsymbol{\sigma},n}}\left[\mathbb{E}_{\boldsymbol{x}}[\mathbb{1}\{\widehat{g}(\boldsymbol{x}) \neq g^*_{\boldsymbol{\sigma}}(\boldsymbol{x})\}]\Big|\boldsymbol{\sigma}\right] \\
&\geq \frac{\bar{\lambda}}{4(1 + 2\bar{\lambda})}\mathbb{E}_{\boldsymbol{\sigma}}\mathbb{E}_{S_{\boldsymbol{\sigma},n}}\left\{\sum_{j=2}^d \mathbb{P}_{\boldsymbol{x}}\big[\boldsymbol{x} \in \Omega^j\big]\mathbb{P}_{\boldsymbol{x}}\big[\widehat{g}(\boldsymbol{x}) \neq g^*_{\boldsymbol{\sigma}}(\boldsymbol{x})\big|\boldsymbol{x} \in \Omega^j\big]\Big|\boldsymbol{\sigma}\right\} \\
&\geq \frac{\varepsilon}{4(1 + 2\bar{\lambda})d}\sum_{j=2}^d \mathbb{E}_{\boldsymbol{\sigma}}\left\{\mathbb{E}_{S_{\boldsymbol{\sigma},n}}\left[\mathbb{P}_{\boldsymbol{x}}\big[\widehat{g}(\boldsymbol{x}) \neq g^*_{\boldsymbol{\sigma}}(\boldsymbol{x})\big|\boldsymbol{x} \in \Omega^j\big]\right]\Big|\boldsymbol{\sigma}\right\}
\end{aligned}
\tag{20}
$$

In the last inequality we use the fact that $\mathbb{P}_{\boldsymbol{x}}[\boldsymbol{x} \in \boldsymbol{\Omega}^j] = \epsilon/(\bar{\lambda}(d-1)) \geq \epsilon/(\bar{\lambda}d)$ for $j \geq 2$.

Let $\boldsymbol{\sigma}^{/j}$ be a Rademacher vector conditional on coordinates $\{1, ..., j-1, j+1, ...d\}$. Let $\boldsymbol{\sigma}^{\{-j\}}$ be a vector equal to $\sigma$ except at the $j$th entry. We drop the $n$ subscript from $S_{\boldsymbol{\sigma},n}$ for simplicity. Above equation becomes:

$$
\begin{aligned}
&\mathbb{E}_{\boldsymbol{\sigma}}\mathbb{E}_{S_{\boldsymbol{\sigma}}}[R(\widehat{g}, f^*, \beta) - R(g_{\boldsymbol{\sigma}}^*, f^*, \beta)] \\
&\geq \frac{\varepsilon}{8(1+2\bar{\lambda})d}\sum_{j=2}^{d}\mathbb{E}_{\boldsymbol{\sigma}^{/j}}\Bigg\{\mathbb{E}_{S_{\boldsymbol{\sigma}}}\Big[\mathbb{P}_{\boldsymbol{x}}\big[\widehat{g}(\boldsymbol{x}) \neq g_{\boldsymbol{\sigma}}^*(\boldsymbol{x})\big|\boldsymbol{x} \in \Omega^j\big]\Big] \\
&\quad + \mathbb{E}_{S_{\boldsymbol{\sigma}^{\{-j\}}}}\Big[\mathbb{P}_{\boldsymbol{x}}\big[\widehat{g}(\boldsymbol{x}) \neq g_{\boldsymbol{\sigma}}^*(\boldsymbol{x})\big|\boldsymbol{x} \in \Omega^j\big]\Big]\Big|\boldsymbol{\sigma}^{/j}\Bigg\} \\
&= \frac{1}{2^{d-1}}\sum_{\boldsymbol{\sigma}^{/j}\in\{+1,-1\}^{d-1}}\frac{\varepsilon}{8(1+2\bar{\lambda})d}\sum_{j=2}^{d}\Bigg\{\mathbb{P}_{S_{\boldsymbol{\sigma}},\boldsymbol{x}}\big[\widehat{g}(\boldsymbol{x}) \neq g_{\boldsymbol{\sigma}}^*(\boldsymbol{x})\big|\boldsymbol{x} \in \Omega^j\big] \\
&\quad + \mathbb{P}_{S_{\boldsymbol{\sigma}^{\{-j\}}},\boldsymbol{x}}\big[\widehat{g}(\boldsymbol{x}) \neq g_{\boldsymbol{\sigma}}^*(\boldsymbol{x})\big|\boldsymbol{x} \in \Omega^j\big]\Bigg\}
\end{aligned}
\tag{21}
$$

We make our notation more specific. Let $\mathcal{A}(S_{\boldsymbol{\sigma}}) = \widehat{g}_{\boldsymbol{\sigma}}$ and $\mathcal{A}(S_{\boldsymbol{\sigma}^{\{-j\}}}) = \widehat{g}_{\boldsymbol{\sigma}^{-j}}$. For simplicity, by $g_{\boldsymbol{\sigma}^{-j}}^*$ we mean $g_{\boldsymbol{\sigma}^{\{-j\}}}^*$.

Note that for all $\boldsymbol{x} \in \Omega^j$, $g_{\boldsymbol{\sigma}^{-j}}^*(\boldsymbol{x}) \neq g_{\boldsymbol{\sigma}}^*(\boldsymbol{x})$ could happen only when $\alpha \geq \|\boldsymbol{x}\|$ or $\|\boldsymbol{x}\| \geq 1-\alpha$. So equation 21 becomes

$$
\begin{aligned}
&\mathbb{E}_{\boldsymbol{\sigma}}\mathbb{E}_{S_{\boldsymbol{\sigma}}}[R(\widehat{g}, f^*, \beta) - R(g_{\boldsymbol{\sigma}}^*, f^*, \beta)] \\
&\geq \frac{1}{2^{d-1}}\sum_{\boldsymbol{\sigma}^{/j}\in\{+1,-1\}^{d-1}}\frac{\varepsilon}{8(1+2\bar{\lambda})d}\sum_{j=2}^{d}\Bigg\{\mathbb{P}_{S_{\boldsymbol{\sigma}},\boldsymbol{x}}\big[\widehat{g}_{\boldsymbol{\sigma}}(\boldsymbol{x}) \neq g_{\boldsymbol{\sigma}}^*(\boldsymbol{x})\big|\boldsymbol{x} \in \Omega^j\big] \\
&\quad + \mathbb{P}_{S_{\boldsymbol{\sigma}^{\{-j\}}},\boldsymbol{x}}\big[\widehat{g}_{\boldsymbol{\sigma}^{-j}}(\boldsymbol{x}) \neq g_{\boldsymbol{\sigma}^{-j}}^*(\boldsymbol{x})\big|\boldsymbol{x} \in \Omega^j\big]\Bigg\} \\
&= \frac{1}{2^{d-1}}\sum_{\boldsymbol{\sigma}^{/j}\in\{+1,-1\}^{d-1}}\frac{\alpha\varepsilon}{8(1+2\bar{\lambda})d}\sum_{j=2}^{d}\Bigg\{\mathbb{P}_{S_{\boldsymbol{\sigma}},\boldsymbol{x}}\big[\widehat{g}_{\boldsymbol{\sigma}}(\boldsymbol{x}) \neq g_{\boldsymbol{\sigma}}^*(\boldsymbol{x})\big|\boldsymbol{x} \in \Omega^j, \alpha \geq \|\boldsymbol{x}\|, or, \|\boldsymbol{x}\| \geq 1-\alpha\big] \\
&\quad + \mathbb{P}_{S_{\boldsymbol{\sigma}^{\{-j\}}},\boldsymbol{x}}\big[\widehat{g}_{\boldsymbol{\sigma}^{-j}}(\boldsymbol{x}) \neq g_{\boldsymbol{\sigma}^{-j}}^*(\boldsymbol{x})\big|\boldsymbol{x} \in \Omega^j, \alpha \geq \|\boldsymbol{x}\|, or, \|\boldsymbol{x}\| \geq 1-\alpha\big]\Bigg\} \\
&= \frac{1}{2^{d-1}}\sum_{\boldsymbol{\sigma}^{/j}\in\{+1,-1\}^{d-1}}\frac{\alpha\varepsilon}{8(1+2\bar{\lambda})d}\sum_{j=2}^{d}\Bigg\{\mathbb{P}_{S_{\boldsymbol{\sigma}},\boldsymbol{x}}\big[\widehat{g}_{\boldsymbol{\sigma}}(\boldsymbol{x}) \neq g_{\boldsymbol{\sigma}}^*(\boldsymbol{x})\big|\boldsymbol{x} \in \Omega^j, \alpha \geq \|\boldsymbol{x}\|, or, \|\boldsymbol{x}\| \geq 1-\alpha\big] \\
&\quad + \mathbb{P}_{S_{\boldsymbol{\sigma}^{\{-j\}}},\boldsymbol{x}}\big[\widehat{g}_{\boldsymbol{\sigma}^{-j}}(\boldsymbol{x}) \neq -g_{\boldsymbol{\sigma}}^*(\boldsymbol{x})\big|\boldsymbol{x} \in \Omega^j, \alpha \geq \|\boldsymbol{x}\|, or, \|\boldsymbol{x}\| \geq 1-\alpha\big]\Bigg\}
\end{aligned}
\tag{22}
$$

Next we make Equation 22 independent of $x$.

$$\mathbb{E}_{\boldsymbol{\sigma}}\mathbb{E}_{S_{\boldsymbol{\sigma}}}[R(\widehat{g}, f^*, \beta) - R(g^*_{\boldsymbol{\sigma}}, f^*, \beta)]$$

$$\geq \frac{1}{2^{d-1}} \sum_{\boldsymbol{\sigma}/j \in \{+1,-1\}^{d-1}} \frac{\alpha\varepsilon}{8(1+2\bar{\lambda})d} \sum_{j=2}^{d} \Bigg\{ \mathbb{P}_{S_{\boldsymbol{\sigma}},\boldsymbol{x}}\big[\widehat{g}_{\boldsymbol{\sigma}}(\boldsymbol{x}) \neq g^*_{\boldsymbol{\sigma}}(\boldsymbol{x}) \big| \boldsymbol{x} \in \Omega^j, \alpha \geq \|\boldsymbol{x}\|, or, \|\boldsymbol{x}\| \geq 1-\alpha\big]$$

$$+ \mathbb{P}_{S_{\boldsymbol{\sigma}\{-j\}},\boldsymbol{x}}\big[\widehat{g}_{\boldsymbol{\sigma}^{-j}}(\boldsymbol{x}) \neq -g^*_{\boldsymbol{\sigma}}(\boldsymbol{x}) \big| \boldsymbol{x} \in \Omega^j, \alpha \geq \|\boldsymbol{x}\|, or, \|\boldsymbol{x}\| \geq 1-\alpha\big] \Bigg\}$$

$$= \frac{1}{2^{d-1}} \sum_{\boldsymbol{\sigma}/j \in \{+1,-1\}^{d-1}} \frac{\alpha\varepsilon}{8(1+2\bar{\lambda})d} \sum_{j=2}^{d} \Bigg\{ \mathbb{E}_{S_{\boldsymbol{\sigma}},\boldsymbol{x}}\big[\mathbb{1}\{\widehat{g}_{\boldsymbol{\sigma}}(\boldsymbol{x}) \neq g^*_{\boldsymbol{\sigma}}(\boldsymbol{x})\} \big| \boldsymbol{x} \in \Omega^j, \alpha \geq \|\boldsymbol{x}\|, or, \|\boldsymbol{x}\| \geq 1-\alpha\big]$$

$$+ \mathbb{E}_{S_{\boldsymbol{\sigma}\{-j\}},\boldsymbol{x}}\big[\mathbb{1}\{\widehat{g}_{\boldsymbol{\sigma}^{-j}}(\boldsymbol{x}) \neq -g^*_{\boldsymbol{\sigma}}(\boldsymbol{x})\} \big| \boldsymbol{x} \in \Omega^j, \alpha \geq \|\boldsymbol{x}\|, or, \|\boldsymbol{x}\| \geq 1-\alpha\big] \Bigg\}$$

$$= \frac{1}{2^{d-1}} \sum_{\boldsymbol{\sigma}/j \in \{+1,-1\}^{d-1}} \frac{\alpha\varepsilon}{8(1+2\bar{\lambda})d} \sum_{j=2}^{d} \Bigg\{ \mathbb{E}_{S_{\boldsymbol{\sigma}},\boldsymbol{x}}\big[\mathbb{1}\{\widehat{g}_{\boldsymbol{\sigma}} \neq g^*_{\boldsymbol{\sigma}}\} \big| \boldsymbol{x} \in \Omega^j, \alpha \geq \|\boldsymbol{x}\|, or, \|\boldsymbol{x}\| \geq 1-\alpha\big] \qquad (23)$$

$$+ \mathbb{E}_{S_{\boldsymbol{\sigma}\{-j\}},\boldsymbol{x}}\big[\mathbb{1}\{\widehat{g}_{\boldsymbol{\sigma}^{-j}} \neq -g^*_{\boldsymbol{\sigma}}\} \big| \boldsymbol{x} \in \Omega^j, \alpha \geq \|\boldsymbol{x}\|, or, \|\boldsymbol{x}\| \geq 1-\alpha\big] \Bigg\}$$

$$\overset{*}{=} \frac{1}{2^{d-1}} \sum_{\boldsymbol{\sigma} \in \{+1,-1\}^{d-1}} \frac{\alpha\varepsilon}{8(1+2\bar{\lambda})d} \sum_{j=2}^{d} \Bigg\{ \mathbb{P}_{S_{\boldsymbol{\sigma}}}\big[\widehat{g}_{\boldsymbol{\sigma}} \neq g^*_{\boldsymbol{\sigma}}\big] + \mathbb{P}_{S_{\boldsymbol{\sigma}\{-j\}}}\big[\widehat{g}_{\boldsymbol{\sigma}^{-j}} \neq -g^*_{\boldsymbol{\sigma}}\big] \Bigg\}$$

$$= \frac{1}{2^{d-1}} \sum_{\boldsymbol{\sigma} \in \{+1,-1\}^{d-1}} \frac{\alpha\varepsilon}{8(1+2\bar{\lambda})d} \sum_{j=2}^{d} \Bigg\{ 1 - \mathbb{P}_{S_{\boldsymbol{\sigma}}}\big[\widehat{g}_{\boldsymbol{\sigma}} \neq -g^*_{\boldsymbol{\sigma}}\big] + \mathbb{P}_{S_{\boldsymbol{\sigma}\{-j\}}}\big[\widehat{g}_{\boldsymbol{\sigma}^{-j}} \neq -g^*_{\boldsymbol{\sigma}}\big] \Bigg\}$$

$$\geq \frac{1}{2^{d-1}} \sum_{\boldsymbol{\sigma}/j \in \{+1,-1\}^{d-1}} \frac{\alpha\varepsilon}{8(1+2\bar{\lambda})d} \sum_{j=2}^{d} \Bigg\{ 1 - \|Q_{\sigma}^{(n)} - Q_{\sigma\{-j\}}^{(n)}\|_{TV} \Bigg\}$$

where $Q_{\sigma}^{(n)}$ and $Q_{\sigma\{-j\}}^{(n)}$ are the product distribution of $n$ samples for $S_{\sigma}$ and $S_{\sigma^{-j}}$ respectively. The last step of inequality follows from the Le Cam's method. In the Equation $*$ we use the fact that for all $\boldsymbol{x}$ s.t., $\|x\| \leq \alpha$ or $\|x\| \geq 1 - \alpha$, $\mathbb{1}\{\widehat{g}_{\boldsymbol{\sigma}^j}(\boldsymbol{x}) \neq g^*_{\boldsymbol{\sigma}^j}(\boldsymbol{x})\} = \mathbb{1}\{\widehat{g}_{\boldsymbol{\sigma}^j} \neq g^*_{\boldsymbol{\sigma}^j}\}$ is free of $\boldsymbol{x}$.

Let $Q_{\sigma}$ be the distribution of $S_{\sigma}$ and $Q_{\sigma\{-j\}}$ be distribution of $S_{\sigma^{-j}}$. The total variation distance can be bounded using the Hellinger distance, which is denoted as $\mathcal{H}(\cdot, \cdot)$. Below we bound the TV distance using Hellinger distance.

$$\|Q_{\sigma}^{(n)} - Q_{\sigma\{-j\}}^{(n)}\|_{TV}$$

$$\leq \mathcal{H}(Q_{\sigma}^{(n)}, Q_{\sigma\{-j\}}^{(n)})\sqrt{1 - \frac{\mathcal{H}^2(Q_{\sigma}^{(n)}, Q_{\sigma\{-j\}}^{(n)})}{4}}$$

$$\leq \sqrt{n}\mathcal{H}(Q_{\sigma}, Q_{\sigma\{-j\}})\sqrt{1 - \frac{\mathcal{H}^2(Q_{\sigma}^{(n)}, Q_{\sigma\{-j\}}^{(n)})}{4}} \qquad (24)$$

$$\overset{\mathcal{H}^2(Q_{\sigma}^{(n)}, Q_{\sigma\{-j\}}^{(n)}) \leq n\mathcal{H}^2(Q_{\sigma}, Q_{\sigma\{-j\}})}{\leq} \sqrt{n}\mathcal{H}(Q_{\sigma}, Q_{\sigma\{-j\}})$$

Now we bound the Hellinger distance.

$$
\begin{aligned}
&\mathcal{H}^2(Q_\sigma, Q_{\sigma\{-j\}}) \\
&= \int_{\boldsymbol{x},y} \left(\sqrt{Q_\sigma(\boldsymbol{x},y)} - \sqrt{Q_{\sigma\{-j\}}(\boldsymbol{x},y)}\right)^2 d\boldsymbol{x}dy \\
&= \int_{\boldsymbol{x}\in\Omega^j, \|\boldsymbol{x}\|\leq\alpha} \int_{y=f^*(\boldsymbol{x})} \left(\sqrt{Q_\sigma(\boldsymbol{x},y)} - \sqrt{Q_{\sigma\{-j\}}(\boldsymbol{x},y)}\right)^2 d\boldsymbol{x}dy \\
&\quad + \int_{\boldsymbol{x}\in\Omega^j, \|\boldsymbol{x}\|\geq 1-\alpha} \int_{y=f^*(\boldsymbol{x})} \left(\sqrt{Q_\sigma(\boldsymbol{x},y)} - \sqrt{Q_{\sigma\{-j\}}(\boldsymbol{x},y)}\right)^2 d\boldsymbol{x}dy \\
&\quad + \int_{\boldsymbol{x}\in\Omega^j, \|\boldsymbol{x}\|\leq\alpha} \int_{y\neq f^*(\boldsymbol{x})} \left(\sqrt{Q_\sigma(\boldsymbol{x},y)} - \sqrt{Q_{\sigma\{-j\}}(\boldsymbol{x},y)}\right)^2 d\boldsymbol{x}dy \\
&\quad + \int_{\boldsymbol{x}\in\Omega^j, \|\boldsymbol{x}\|\geq 1-\alpha} \int_{y\neq f^*(\boldsymbol{x})} \left(\sqrt{Q_\sigma(\boldsymbol{x},y)} - \sqrt{Q_{\sigma\{-j\}}(\boldsymbol{x},y)}\right)^2 d\boldsymbol{x}dy \\
&= \frac{\alpha\varepsilon}{d\bar{\lambda}}\left\{\left(\sqrt{\frac{3}{4}+\frac{\bar{\lambda}}{2}} - \sqrt{\frac{3}{4}-\frac{\bar{\lambda}}{2}}\right)^2 + \left(\sqrt{\frac{1}{4}+\frac{\bar{\lambda}}{2}} - \sqrt{\frac{1}{4}-\frac{\bar{\lambda}}{2}}\right)^2\right\} \\
&\leq \frac{3\alpha\varepsilon\bar{\lambda}}{d}
\end{aligned}
\tag{25}
$$

Thus we can bound the total variation distance as:

$$
\|Q^{(n)}_\sigma - Q^{(n)}_{\sigma\{-j\}}\|_{TV} \leq \sqrt{\frac{3n\alpha\varepsilon\bar{\lambda}}{d}}
\tag{26}
$$

Note that from inequality 23, inequality 24 and inequality 26 we have

$$
\begin{aligned}
&\mathbb{E}_{\boldsymbol{\sigma}}\mathbb{E}_{S_\sigma}[R(\mathcal{A}(S_{\boldsymbol{\sigma}}),\beta) - R(g^*,f^*,\beta)] \\
&= \mathbb{E}_{\boldsymbol{\sigma}}\mathbb{E}_{S_\sigma}[R(\widehat{g},f^*,\beta) - R(g^*,f^*,\beta)] \\
&\geq \frac{d-1}{d}\frac{\alpha\varepsilon}{8(1+2\bar{\lambda})}\left(1 - \sqrt{\frac{3n\alpha\varepsilon\bar{\lambda}}{d}}\right)
\end{aligned}
\tag{27}
$$

Above implies

$$
\sup_{\boldsymbol{\sigma}\in\{+1,-1\}^d} \mathbb{E}_{S_\sigma}[R(\mathcal{A}(S_{\boldsymbol{\sigma}}),f^*,\beta) - R(g^*,f^*,\beta)] \geq \mathbb{E}_{\boldsymbol{\sigma}}\mathbb{E}_{S_\sigma}[R(\mathcal{A}(S_{\boldsymbol{\sigma}}),\beta) - R(g^*,f^*,\beta)]
$$

$$
\geq \frac{d-1}{d}\frac{\alpha\varepsilon}{8(1+2\bar{\lambda})}\left(1 - \sqrt{\frac{3n\alpha\varepsilon\bar{\lambda}}{d}}\right).
$$

Since $|\mathcal{G}| = 2^d$, any algorithm $\mathcal{A}$ needs at least $n = \Omega\left(\frac{\log|\mathcal{G}|}{\bar{\lambda}\varepsilon\alpha}\right)$ number of samples so that there is a hope to achieve

$$
\sup_{\boldsymbol{\sigma}} \mathbb{E}_{S_\sigma}[R(\mathcal{A}(S_{\boldsymbol{\sigma}}),\beta)] - R(g^*,f^*,\beta) \leq \frac{\alpha\varepsilon}{32(1+2\bar{\lambda})}.
$$

Replacing $\alpha\varepsilon$ with $\varepsilon$ finishes the proof.

**Remark 4.** *From the second inequality in Equation 20, it can be observed that the construction of information theoretic lower bound for risk function $R(g,\beta)$ can also be applied to the construction of an $\Omega(\log(|\mathcal{G}|/(\bar{\lambda}\varepsilon)))$ sample complexity lower bound for $\mathbb{E}_{\boldsymbol{x}}[g(\boldsymbol{x}) \neq g^*(\boldsymbol{x})]$. Thus our Corollary 1 also achieves minimax-optimal rate for recovering $g^*$ for family of Noise Generative Process.*

$\square$

## A.3 Proof of Sample Complexity Upper Bound

Here we prove Theorem 1 in which we bound the risk gap $R(g; f^*, \beta) - R(g^*; f^*, \beta)$. Recall that the empirical version of the selector loss is

$$R_{S_n}(g; f, \beta) = \frac{1}{n} \sum_{i=1}^{n} \left\{ \beta \mathbb{1}\{g(\boldsymbol{x}_i) = 1\} \mathbb{1}\{f(\boldsymbol{x}_i) \neq y_i\} + \mathbb{1}\{g(\boldsymbol{x}_i) = 1\} \mathbb{1}\{f(\boldsymbol{x}_i) = y_i\} \right\}.$$

Our high level approach is as follows. We first analyze the gap between $R_{S_n}(\widehat{g}; f^*, \beta)$ and $R_{S_n}(g^*; f^*, \beta)$ and provide an upper bound for it. Then we use this upper bound to get an upper bound for the gap between $R(\widehat{g}; f^*, \beta)$ and $R(g^*; f^*, \beta)$ using concentration properties and Bernstein inequality.

**CASE I: $\widehat{f}(\cdot) \in \mathcal{F}$ and $\mathbb{E}_{\boldsymbol{x}}[\widehat{f}(\boldsymbol{x}) \neq f^*(\boldsymbol{x})] \leq \frac{\varepsilon}{8\beta}$ with probability at least $1 - \delta$.** To upper bound $R_{S_n}(\widehat{g}; f^*, \beta) - R_{S_n}(g^*; f^*, \beta)$, we use $R_{S_n}(\widehat{g}; \widehat{f}, \beta)$ and $R_{S_n}(g^*; \widehat{f}, \beta)$ as a middle step. Since $\widehat{g}$ is the empirical risk minimizer, we have

$$R_{S_n}(\widehat{g}; \widehat{f}, \beta) \leq R_{S_n}(g^*; \widehat{f}, \beta). \tag{28}$$

Next we leverage on the fact that $\widehat{f}$ is consistent with $f$ to establish an inequality in following fashion:

$$R_{S_n}(\widehat{g}; f^*, \beta) \leq R_{S_n}(g^*; f^*, \beta) + const \cdot \varepsilon$$

Note we have that

$$
\begin{aligned}
&R_{S_n}(\widehat{g}; \widehat{f}, \beta) \\
&= \frac{1}{n} \sum_{i=1}^{n} \mathbb{1}\left\{ \widehat{f}(\boldsymbol{x}_i) \neq f^*(\boldsymbol{x}_i) \right\} \left\{ \beta \mathbb{1}\{\widehat{g}(\boldsymbol{x}_i) = 1\} \mathbb{1}\{\widehat{f}(\boldsymbol{x}_i) \neq y_i\} + \mathbb{1}\{\widehat{g}(\boldsymbol{x}_i) = 1\} \mathbb{1}\{\widehat{f}(\boldsymbol{x}_i) = y_i\} \right\} \\
&\quad + \frac{1}{n} \sum_{i=1}^{n} \mathbb{1}\left\{ \widehat{f}(\boldsymbol{x}_i) = f^*(\boldsymbol{x}_i) \right\} \left\{ \beta \mathbb{1}\{\widehat{g}(\boldsymbol{x}_i) = 1\} \mathbb{1}\{\widehat{f}(\boldsymbol{x}_i) \neq y_i\} + \mathbb{1}\{\widehat{g}(\boldsymbol{x}_i) = 1\} \mathbb{1}\{\widehat{f}(\boldsymbol{x}_i) = y_i\} \right\}
\end{aligned}
\tag{29}
$$

Recall that

$$R_{S_n}(\widehat{g}; f^*, \beta) = \frac{1}{n} \sum_{i=1}^{n} \left[ \beta \mathbb{1}\{\widehat{g}(\boldsymbol{x}_i) = 1\} \mathbb{1}\{f^*(\boldsymbol{x}_i) \neq y_i\} + \mathbb{1}\{\widehat{g}(\boldsymbol{x}_i) = 1\} \mathbb{1}\{f^*(\boldsymbol{x}_i) = y_i\} \right].$$

So

$$
\begin{aligned}
&R_{S_n}(\widehat{g}; \widehat{f}, \beta) \\
&= R_{S_n}(\widehat{g}; f^*, \beta) \\
&\quad - \frac{1}{n} \sum_{i=1}^{n} \mathbb{1}\left\{ \widehat{f}(\boldsymbol{x}_i) \neq f^*(\boldsymbol{x}_i) \right\} \left\{ \beta \mathbb{1}\{\widehat{g}(\boldsymbol{x}_i) = 1\} \mathbb{1}\{f^*(\boldsymbol{x}_i) \neq y_i\} + \mathbb{1}\{\widehat{g}(\boldsymbol{x}_i) = 1\} \mathbb{1}\{f^*(\boldsymbol{x}_i) = y_i\} \right\} \\
&\quad + \frac{1}{n} \sum_{i=1}^{n} \mathbb{1}\left\{ \widehat{f}(\boldsymbol{x}_i) \neq f^*(\boldsymbol{x}_i) \right\} \left\{ \beta \mathbb{1}\{\widehat{g}(\boldsymbol{x}_i) = 1\} \mathbb{1}\{\widehat{f}(\boldsymbol{x}_i) \neq y_i\} + \mathbb{1}\{\widehat{g}(\boldsymbol{x}_i) = 1\} \mathbb{1}\{\widehat{f}(\boldsymbol{x}_i) = y_i\} \right\} \\
&\geq R_{S_n}(\widehat{g}; f^*, \beta) - \frac{\beta - 1}{n} \sum_{i=1}^{n} \mathbb{1}\left\{ \widehat{f}(\boldsymbol{x}_i) \neq f^*(\boldsymbol{x}_i) \right\}
\end{aligned}
\tag{30}
$$

Recall that in the theorem assumptions we have $\mathbb{E}_{\boldsymbol{x}}[\widehat{f}(\boldsymbol{x}) \neq f^*(\boldsymbol{x})] \leq \frac{\varepsilon}{8\beta}$ with probability at least $1 - \delta$. By Lemma 2, if $n \geq \frac{24\beta^2 \log(|\mathcal{F}|/\delta)}{\varepsilon}$ we have with probability at least $1 - \delta$, $\frac{1}{n} \sum_{i=1}^{n} \mathbb{1}\{\widehat{f}(\boldsymbol{x}_i) \neq f^*(\boldsymbol{x}_i)\} \leq \frac{\varepsilon}{4\beta}$, so we have

$$R_{S_n}(\widehat{g}; \widehat{f}, \beta) \geq R_{S_n}(\widehat{g}; f^*, \beta) - \varepsilon/4 \tag{31}$$

With a similar approach we get that

$$R_{S_n}(g^*; \widehat{f}, \beta) \leq R_{S_n}(g^*; f^*, \beta) + \varepsilon/4$$

Thus using (28) the following inequality holds with probability at least $1 - \delta$

$$R_{S_n}(\widehat{g}; f^*, \beta) \le R_{S_n}(g^*; f^*, \beta) + \varepsilon/2. \tag{32}$$

To get a bound for $R(\widehat{g}; f^*, \beta) - R(g^*; f^*, \beta)$, we first define $\ell(g; f, \boldsymbol{x}, y) = \beta \mathbb{1}\{g(\boldsymbol{x}) = 1\} \mathbb{1}\{f(\boldsymbol{x}) \ne y\} + \mathbb{1}\{g(\boldsymbol{x}) = 1\} \mathbb{1}\{f(\boldsymbol{x}) = y\}$. Note that at this point we think of $\beta$ as fixed and so we have not included it in the arguments of $\ell(\cdot)$ for simplicity.

Observe that $R_{S_n}(g, f^*, \beta) = \frac{1}{n} \sum_{i=1}^n \ell(g; f^*, \boldsymbol{x}_i, y)$ and $R(g, f^*, \beta) = \mathbb{E}_{\boldsymbol{x},y}\big[\ell(g; f^*, \boldsymbol{x}, y)\big]$ for any $g$. First we have the following simple inequality directly taken from (32).

$$
\begin{aligned}
R(\widehat{g}, f^*, \beta) &- R(g^*, f^*, \beta) \\
&= \mathbb{E}_{\boldsymbol{x},y} \ell(\widehat{g}; f^*, \boldsymbol{x}, y) - \mathbb{E}_{\boldsymbol{x},y} \ell(g^*; f^*, \boldsymbol{x}, y) \\
&\le R_{S_n}(g^*; f^*, \beta) - R_{S_n}(\widehat{g}; f^*, \beta) - (\mathbb{E}_{\boldsymbol{x},y} \ell(g^*; f^*, \boldsymbol{x}, y) - \mathbb{E}_{\boldsymbol{x},y} \ell(\widehat{g}; f^*, \boldsymbol{x}, y)) + \varepsilon/2
\end{aligned}
\tag{33}
$$

By defining $\Delta\ell(g^*, g, \boldsymbol{x}, y) = \ell(g^*; f^*, \boldsymbol{x}, y) - \ell(g; f^*, \boldsymbol{x}, y)$ for any $g$, we can express the above inequality as follows:

$$
\begin{aligned}
\mathbb{E}_{\boldsymbol{x},y} \ell(\widehat{g}; f^*, \boldsymbol{x}, y) &- \mathbb{E}_{\boldsymbol{x},y} \ell(g^*; f^*, \boldsymbol{x}, y) \\
&\le \frac{1}{n} \sum_{i=1}^n \Delta\ell(g^*, \widehat{g}, \boldsymbol{x}_i, y) - \mathbb{E}_{\boldsymbol{x},y} \Delta\ell(g^*; \widehat{g}, \boldsymbol{x}, y) + \varepsilon/2
\end{aligned}
\tag{34}
$$

To bound $\frac{1}{n} \sum_{i=1}^n \Delta\ell(g^*, \widehat{g}, \boldsymbol{x}_i, y) - \mathbb{E}_{\boldsymbol{x},y} \Delta\ell(g^*; \widehat{g}, \boldsymbol{x}, y)$ with high probability over all $S_n$, we need to find a bound on $\frac{1}{n} \sum_{i=1}^n \Delta\ell(g^*, g, \boldsymbol{x}_i, y) - \mathbb{E}_{\boldsymbol{x},y} \Delta\ell(g^*; g, \boldsymbol{x}, y)$ that is true for all $g$ simultaneously with high probability. We have :

$$
\begin{aligned}
\mathbb{P}_{S_n} &\left[ \exists g \in \mathcal{G}, \left\{ \frac{1}{n} \sum_{i=1}^n \Delta\ell(g^*, g; \boldsymbol{x}_i, y_i) - \mathbb{E}_{\boldsymbol{x},y}[\Delta\ell(g^*, g; \boldsymbol{x}, y)] \right. \right. \\
&\qquad \left. \left. \ge \frac{n}{\beta} \log(|\mathcal{G}|/\delta) + \sqrt{\frac{2\boldsymbol{Var}(\Delta(g^*, g; \boldsymbol{x}, y)) \log(|\mathcal{G}|/\delta)}{n}} \right\} \right] \\
&\le \sum_{g \in \mathcal{G}} \mathbb{P}_{S_n} \left[ \frac{1}{n} \sum_{i=1}^n \Delta\ell(g^*, g; \boldsymbol{x}_i, y_i) - \mathbb{E}_{\boldsymbol{x},y}[\Delta\ell(g^*, g; \boldsymbol{x}, y)] \right. \\
&\qquad \left. \ge \frac{n}{\beta} \log(|\mathcal{G}|/\delta) + \sqrt{\frac{2\boldsymbol{Var}(\Delta(g^*, g; \boldsymbol{x}, y)) \log(|\mathcal{G}|/\delta)}{n}} \right]
\end{aligned}
\tag{35}
$$

Now to bound $\frac{1}{n} \sum_{i=1}^n \Delta\ell(g^*, g, \boldsymbol{x}_i, y) - \mathbb{E}_{\boldsymbol{x},y} \Delta\ell(g^*; g, \boldsymbol{x}, y)$ we use Bernstein inequality. For that we need to bound $\boldsymbol{Var}_{\boldsymbol{x},y}[\Delta\ell(g^*, g, \boldsymbol{x}, y)]$. First we expand $\Delta\ell(g^*, g, \boldsymbol{x}, y)$.

$$
\begin{aligned}
\Delta\ell(g^*, g, \boldsymbol{x}, y) &= \ell(g^*; f^*, \boldsymbol{x}, y) - \ell(g; f^*, \boldsymbol{x}, y) \\
&= \beta \mathbb{1}\{g^*(\boldsymbol{x}) = 1\} \mathbb{1}\{f^*(\boldsymbol{x}) \ne y\} + \mathbb{1}\{g^*(\boldsymbol{x}) = 1\} \mathbb{1}\{f^*(\boldsymbol{x}) = y\} \\
&\quad - \beta \mathbb{1}\{g(\boldsymbol{x}) = 1\} \mathbb{1}\{f^*(\boldsymbol{x}) \ne y\} - \mathbb{1}\{g(\boldsymbol{x}) = 1\} \mathbb{1}\{f^*(\boldsymbol{x}) = y\}
\end{aligned}
$$

So we have

$$
\begin{aligned}
\Delta^2 &\ell(g^*, g, \boldsymbol{x}, y) \\
&= \beta^2 \left[ \mathbb{1}\{g^*(\boldsymbol{x}) = 1\} + \mathbb{1}\{g(\boldsymbol{x}) = 1\} - 2\mathbb{1}\{g(\boldsymbol{x}) = 1\} \mathbb{1}\{g^*(\boldsymbol{x}) = 1\} \right] \mathbb{1}\{f^*(\boldsymbol{x}) \ne y\} \\
&\quad + \left[ \mathbb{1}\{g^*(\boldsymbol{x}) = 1\} + \mathbb{1}\{g(\boldsymbol{x}) = 1\} - 2\mathbb{1}\{g(\boldsymbol{x}) = 1\} \mathbb{1}\{g^*(\boldsymbol{x}) = 1\} \right] \mathbb{1}\{f^*(\boldsymbol{x}) = y\} \\
&= \left( \beta^2 \mathbb{1}\{f^*(\boldsymbol{x}) \ne y\} + \mathbb{1}\{f^*(\boldsymbol{x}) = y\} \right) \left[ \mathbb{1}\{g(\boldsymbol{x}) = 1\} \mathbb{1}\{g^*(\boldsymbol{x}) \ne 1\} + \mathbb{1}\{g(\boldsymbol{x}) \ne 1\} \mathbb{1}\{g^*(\boldsymbol{x}) = 1\} \right] \\
&\le \beta^2 \mathbb{1}\{g^*(\boldsymbol{x}) \ne g(\boldsymbol{x})\}
\end{aligned}
\tag{36}
$$

Hence we conclude that $\boldsymbol{Var}_{\boldsymbol{x},y}[\Delta\ell(g^*,g,\boldsymbol{x},y)] \leq \mathbb{E}_{\boldsymbol{x},y}\Delta^2\ell(g^*,g,\boldsymbol{x},y) \leq \beta^2\mathbb{E}_{\boldsymbol{x}}[\mathbb{1}\{g^*(\boldsymbol{x}) \neq g(\boldsymbol{x})\}]$. On the other hand, Equation 17 implies that $R(g;f^*,\beta) - R(g^*;f^*,\beta) \geq \frac{\bar{\lambda}}{1+2\bar{\lambda}}\mathbb{E}_{\boldsymbol{x}}[\mathbb{1}\{g^*(\boldsymbol{x}) \neq g(\boldsymbol{x})\}]$. Thus we can use the following inequality to achieve fast rate of convergence using the Bernstein Inequality:

$$\boldsymbol{Var}_{\boldsymbol{x},y}[\Delta\ell(g^*,g;\boldsymbol{x},y)] \leq \frac{\beta^2(1+2\bar{\lambda})}{\bar{\lambda}}\{R(g;f^*,\beta)) - R(g^*;f^*,\beta)\}. \tag{37}$$

We use following version of the Bernstein Inequality, with $X_1,...,X_n$ i.i.d random variable uniformly bounded by $b$:

$$\mathbb{P}\left(\frac{1}{n}\sum_{i=1}^{n}X_i - \mathbb{E}[X] < \frac{b}{n}\log(1/\delta) + \sqrt{\frac{2\boldsymbol{Var}(\boldsymbol{X})\log(1/\delta)}{n}}\right) \geq 1 - \delta$$

Using union bounds, the Bernstein Inequality implies that with probability for all $g \in \mathcal{G}$ simultaneously:

$$\mathbb{P}_{S_n}\left[\left\{\frac{1}{n}\sum_{i=1}^{n}\Delta\ell(g^*,g;\boldsymbol{x}_i,y_i) - \mathbb{E}_{\boldsymbol{x},y}[\Delta\ell(g^*,g;\boldsymbol{x},y)]\right\}\right.$$
$$\left.\leq\frac{\beta}{n}\log\left(|\mathcal{G}|/\delta\right) + \sqrt{\frac{2\boldsymbol{Var}(\Delta(g^*,g;\boldsymbol{x},y))\log\left(|\mathcal{G}|/\delta\right)}{n}}\right] \geq 1 - \delta \tag{38}$$

Thus by applying inequality 38 with $\widehat{g}$ we have:

$$\mathbb{P}_{S_n}\left[\left\{\frac{1}{n}\sum_{i=1}^{n}\Delta\ell(g^*,\widehat{g};\boldsymbol{x}_i,y_i) - \mathbb{E}_{\boldsymbol{x},y}[\Delta\ell(g^*,\widehat{g};\boldsymbol{x},y)]\right\}\right.$$
$$\left.\leq\frac{\beta}{n}\log\left(|\mathcal{G}|/\delta\right) + \sqrt{\frac{2\boldsymbol{Var}(\Delta(g^*,\widehat{g};\boldsymbol{x},y))\log\left(|\mathcal{G}|/\delta\right)}{n}}\right] \geq 1 - \delta$$

By inequality 32, we have $\frac{1}{n}\sum_{i=1}^{n}\Delta\ell(g^*,\widehat{g};\boldsymbol{x}_i,y_i) = R_{S_n}(g^*;f^*,\beta) - R_{S_n}(\widehat{g};f^*,\beta) \geq -\varepsilon/2$ holds with probability $1 - \delta$. Note $R(\widehat{g};f^*,\beta)) - R(g^*;f^*,\beta) = -\mathbb{E}_{\boldsymbol{x},y}[\Delta\ell(g^*,\widehat{g};\boldsymbol{x},y)]$. So we have

$$\mathbb{P}_{S_n}\left[\{R(\widehat{g};f^*,\beta) - R(g^*;f^*,\beta)\}\right.$$
$$\left.\leq\frac{\beta}{n}\log\left(|\mathcal{G}|/\delta\right) + \sqrt{\frac{2\beta^2(1+2\bar{\lambda})\{R(\widehat{g};f^*,\beta)) - R(g^*;f^*,\beta)\}\log\left(|\mathcal{G}|/\delta\right)}{\bar{\lambda}n}} + \varepsilon/2\right]$$
$$\geq 1 - 2\delta$$

The choice of $n \geq \frac{16\beta^2\log(\frac{|\mathcal{G}|}{\delta})}{\bar{\lambda}\varepsilon}$ ensures that with probability at least $1 - 2\delta$, $R(\widehat{g};f^*,\beta) - R(g^*;f^*,\beta) \leq \varepsilon$.

**CASE II: $\widehat{f}(\cdot) \in \mathcal{F}$ that satisfies $\mathbb{E}_{\boldsymbol{x}}[\widehat{f}(\boldsymbol{x}) \neq f^*(\boldsymbol{x})|\boldsymbol{x} \in \Omega_I] \leq \frac{\varepsilon}{8\beta\alpha}$ with probability at least $1 - \delta$.**
Note $\widehat{f}$ only approximates $f^*(\cdot)$ on the informative support $\Omega_I$: $\widehat{f}(\cdot) \in \mathcal{F}$ that satisfies $\mathbb{E}_{\boldsymbol{x}}[\widehat{f}(\boldsymbol{x}) \neq f^*(\boldsymbol{x})|\boldsymbol{x} \in \Omega_I] \leq \frac{\varepsilon}{8\beta\alpha}$ with probability at least $1 - \delta$. For simplicity of analysis, we introduce a 'pseudo' version of $f^*(\cdot)$ denoted as $\widetilde{f}^*$. Let $\widetilde{\mathcal{F}}$ be following hypothesis class:

$$\left\{\widetilde{f}\middle|\widetilde{f}(\boldsymbol{x}) = \begin{cases} f_1(\boldsymbol{x}), & \boldsymbol{x} \in \Omega_U \\ f_2(\boldsymbol{x}), & \boldsymbol{x} \in \Omega_I \end{cases}, f_1 \in \mathcal{F}, f_2 \in \mathcal{F}\right\}$$

and we let $\widetilde{f}^*(\cdot)$ be:

$$\widetilde{f}^*(\boldsymbol{x}) = \begin{cases} \widehat{f}(\boldsymbol{x}), & \boldsymbol{x} \in \Omega_U \\ f^*(\boldsymbol{x}), & \boldsymbol{x} \in \Omega_I \end{cases}$$

Clearly, $\widetilde{f}^* \in \widetilde{\mathcal{F}}$. Note such hypothesis class is only introduced in analysis and is potentially impractical. The cardinality of hypothesis class $|\widetilde{\mathcal{F}}| \leq |\mathcal{F}|^2$. The construction of $\widetilde{f}^*$ is to make $R(g, \widetilde{f}^*, \beta) - R(g^*, \widetilde{f}^*, \beta) \geq R(g, f^*, \beta) - R(g^*, f^*, \beta)$ for all $g \in \mathcal{G}$. To see this, we use the fact that $f^*$ is the Bayes optimal classifier, which implies that $\forall \boldsymbol{x}, \mathbb{E}_y[\mathbb{1}\{g^*(\boldsymbol{x}) \neq 1\}\mathbb{1}\{\widetilde{f}^*(\boldsymbol{x}) \neq y\}] \geq \mathbb{E}_y[\mathbb{1}\{g^*(\boldsymbol{x}) \neq 1\}\mathbb{1}\{f^*(\boldsymbol{x}) \neq y\}]$. Subsequently, we have $\forall \boldsymbol{x}, \mathbb{E}_y[\mathbb{1}\{g^*(\boldsymbol{x}) \neq 1\}\mathbb{1}\{\widetilde{f}^*(\boldsymbol{x}) = y\}] \leq \mathbb{E}_y[\mathbb{1}\{g^*(\boldsymbol{x}) \neq 1\}\mathbb{1}\{f^*(\boldsymbol{x}) = y\}]$. So we have the following.

$$
\begin{aligned}
&R(g(\boldsymbol{x}), \widetilde{f}^*, \beta) - R(g^*(\boldsymbol{x}), \widetilde{f}^*, \beta) \\
=&\mathbb{E}_{\boldsymbol{x},y}\big[\beta\mathbb{1}\{g(\boldsymbol{x}) = 1\}\mathbb{1}\{g^*(\boldsymbol{x}) \neq 1\}\mathbb{1}\{\widetilde{f}^*(\boldsymbol{x}) \neq y\} - \mathbb{1}\{g(\boldsymbol{x}) = 1\}\mathbb{1}\{g^*(\boldsymbol{x}) \neq 1\}\mathbb{1}\{\widetilde{f}^*(\boldsymbol{x}) = y\} \\
&+\mathbb{1}\{g(\boldsymbol{x}) \neq 1\}\mathbb{1}\{g^*(\boldsymbol{x}) = 1\}\mathbb{1}\{\widetilde{f}^*(\boldsymbol{x}) = y\} - \beta\mathbb{1}\{g(\boldsymbol{x}) \neq 1\}\mathbb{1}\{g^*(\boldsymbol{x}) = 1\}\mathbb{1}\{\widetilde{f}^*(\boldsymbol{x}) \neq y\}\big] \\
=&\mathbb{E}_{\boldsymbol{x},y}\big[\beta\mathbb{1}\{g(\boldsymbol{x}) = 1\}\mathbb{1}\{g^*(\boldsymbol{x}) \neq 1\}\mathbb{1}\{\widetilde{f}^*(\boldsymbol{x}) \neq y\} - \mathbb{1}\{g(\boldsymbol{x}) = 1\}\mathbb{1}\{g^*(\boldsymbol{x}) \neq 1\}\mathbb{1}\{\widehat{f}^*(\boldsymbol{x}) = y\} \\
&+\mathbb{1}\{g(\boldsymbol{x}) \neq 1\}\mathbb{1}\{g^*(\boldsymbol{x}) = 1\}\mathbb{1}\{f^*(\boldsymbol{x}) = y\} - \beta\mathbb{1}\{g(\boldsymbol{x}) \neq 1\}\mathbb{1}\{g^*(\boldsymbol{x}) = 1\}\mathbb{1}\{f^*(\boldsymbol{x}) \neq y\}\big] \\
\geq&\mathbb{E}_{\boldsymbol{x},y}\big[\beta\mathbb{1}\{g(\boldsymbol{x}) = 1\}\mathbb{1}\{g^*(\boldsymbol{x}) \neq 1\}\mathbb{1}\{f^*(\boldsymbol{x}) \neq y\} - \mathbb{1}\{g(\boldsymbol{x}) = 1\}\mathbb{1}\{g^*(\boldsymbol{x}) \neq 1\}\mathbb{1}\{f^*(\boldsymbol{x}) = y\} \\
&+\mathbb{1}\{g(\boldsymbol{x}) \neq 1\}\mathbb{1}\{g^*(\boldsymbol{x}) = 1\}\mathbb{1}\{f^*(\boldsymbol{x}) = y\} - \beta\mathbb{1}\{g(\boldsymbol{x}) \neq 1\}\mathbb{1}\{g^*(\boldsymbol{x}) = 1\}\mathbb{1}\{f^*(\boldsymbol{x}) \neq y\}\big] \\
=&\mathbb{E}_{\boldsymbol{x}}\left[\left\{\beta\left(\frac{1}{4} + \frac{\lambda(\boldsymbol{x})}{2}\right) - \frac{3}{4} + \frac{\lambda(\boldsymbol{x})}{2}\right\}\mathbb{1}\{g(\boldsymbol{x}) = 1\}\mathbb{1}\{g^*(\boldsymbol{x}) \neq 1\}\right] \\
&+\mathbb{E}_{\boldsymbol{x}}\left[\left\{\frac{3}{4} + \frac{\lambda(\boldsymbol{x})}{2} - \beta\left(\frac{1}{4} - \frac{\lambda(\boldsymbol{x})}{2}\right)\right\}\mathbb{1}\{g(\boldsymbol{x}) \neq 1\}\mathbb{1}\{g^*(\boldsymbol{x}) = 1\}\right] \\
=&R(g; f^*, \beta) - R(g^*; f^*, \beta)
\end{aligned}
\tag{39}
$$

Meanwhile $\widetilde{f}^*(\cdot)$ also satisfies the property that $\mathbb{E}_{\boldsymbol{x}}[\widetilde{f}^*(\boldsymbol{x}) \neq \widehat{f}(\boldsymbol{x})] \leq \frac{\varepsilon}{8\beta}$ with probability at least $1 - \delta$. Thus by Lemma 2, if $n \geq \frac{24\beta^2 \log(\frac{|\mathcal{F}|}{\delta})}{\varepsilon}$ we have with probability at least $1 - \delta$, $\frac{1}{n}\sum_{i=1}^n \mathbb{1}\{\widehat{f}(\boldsymbol{x}_i) \neq \widetilde{f}^*(\boldsymbol{x}_i)\} \leq \frac{\varepsilon}{8\beta}$.

The rest of the proof is the same as the proof in **CASE I** by replacing $f^*$ with $\widetilde{f}^*$, leveraging on the fact that $\widetilde{f}^*$ is to make $R(g(\boldsymbol{x}), \widetilde{f}^*, \beta) - R(g^*(\boldsymbol{x}), \widetilde{f}^*, \beta) \geq R(g(\boldsymbol{x}), f^*, \beta) - R(g^*(\boldsymbol{x}), f^*, \beta)$

### A.4   Proof for Corollary 6

It can be easily verified that $\beta = 3$ is in the interval $\beta \in \left[\frac{3-2\bar{\lambda}}{1+2\bar{\lambda}} + \bar{\lambda}, \min(\frac{3+2\bar{\lambda}}{1-2\bar{\lambda}} - \frac{\bar{\lambda}}{1-4\bar{\lambda}^2}, 10)\right]$. By the choice of $\beta$, from (17) we have

$$
R(\widehat{g}, f^*, \beta) - R(g^*, f^*, \beta) \geq \frac{\bar{\lambda}}{4(1 + 2\bar{\lambda})}\mathbb{E}_{\boldsymbol{x}}[\mathbb{1}\{\widehat{g}(\boldsymbol{x}) \neq g^*(\boldsymbol{x})\}],
$$

together with the conclusion in Theorem 1 that

$$
R(\widehat{g}; f^*, \beta) - R(g^*; f^*, \beta) \leq \varepsilon
$$

we can conclude that Equation 6 holds.

### A.5   Theorem 4 on improving selective risk

**Theorem 4** (Improving Predictor Selective Risk ). *Let $S_n = \{(\boldsymbol{x}_i, y_i)\}_{i=1}^n$ be i.i.d samples from the Data Generative Process described in Definition 1 under Assumption 1, with $f^*(\cdot) \in \mathcal{F}$ and $g^*(\cdot) \in \mathcal{G}$, $|\mathcal{F}| < \infty, |\mathcal{G}| < \infty$. For any $\widehat{g}(\cdot) \in \mathcal{G}$ s.t., $\mathbb{E}_{\boldsymbol{x}}[\mathbb{1}\{g^*(\boldsymbol{x}) \neq \widehat{g}(\boldsymbol{x})\}] \leq \varepsilon$, let $\widetilde{f} = \underset{f \in \mathcal{F}}{\arg\min} \frac{1}{n}\sum_{i=1}^n \mathbb{1}\{\widehat{g}(\boldsymbol{x}_i) > 0\}\mathbb{1}\{f(\boldsymbol{x}_i) \neq y_i\}$.*

*Then for any $\varepsilon > 0$, $\delta > 0$ such that the following holds: For $n \geq max\{\frac{24\log(\frac{|\mathcal{G}|}{\delta})}{\varepsilon}, \frac{12\log(\frac{|\mathcal{F}|}{\delta})}{\varepsilon}\}$, we have*

$$
\mathbb{E}_{\boldsymbol{x}}[\mathbb{1}\{\widetilde{f}(\boldsymbol{x}) \neq f^*(\boldsymbol{x})\}|g^*(\boldsymbol{x}) \neq \widehat{g}(\boldsymbol{x})] \leq \frac{\varepsilon}{\alpha}
\tag{40}
$$

*Proof.* Recall that $\widetilde{f} = \underset{f \in \mathcal{F}}{\arg\min} \frac{1}{n} \sum_{i=1}^{n} \mathbb{1}\{\widehat{g}(\boldsymbol{x}_i) > 0\} \mathbb{1}\{f(\boldsymbol{x}_i) \neq y_i\}$ with $\widehat{g}(\cdot) \in \mathcal{G}$ s.t., $\mathbb{E}_{\boldsymbol{x}}[\mathbb{1}\{g^*(\boldsymbol{x}) \neq \widehat{g}(\boldsymbol{x})\}] \leq \varepsilon$. From the minimization property we have

$$\frac{1}{n} \sum_{i=1}^{n} \mathbb{1}\{\widehat{g}(\boldsymbol{x}_i) > 0\} \mathbb{1}\{\widetilde{f}(\boldsymbol{x}_i) \neq y_i\} \leq \frac{1}{n} \sum_{i=1}^{n} \mathbb{1}\{\widehat{g}(\boldsymbol{x}_i) > 0\} \mathbb{1}\{f^*(\boldsymbol{x}_i) \neq y_i\} \tag{41}$$

Note we have for any $f$:

$$\left| \frac{1}{n} \sum_{i=1}^{n} \mathbb{1}\{\widehat{g}(\boldsymbol{x}_i) > 0\} \mathbb{1}\{f(\boldsymbol{x}_i) \neq y_i\} - \frac{1}{n} \sum_{i=1}^{n} \mathbb{1}\{g^*(\boldsymbol{x}_i) > 0\} \mathbb{1}\{f(\boldsymbol{x}_i) \neq y_i\} \right|$$

$$\leq \frac{1}{n} \sum_{i=1}^{n} \left| \mathbb{1}\{\widehat{g}(\boldsymbol{x}_i) > 0\} - \mathbb{1}\{g^*(\boldsymbol{x}_i) > 0\} \right| \mathbb{1}\{f(\boldsymbol{x}_i) \neq y_i\}$$

$$\leq \frac{1}{n} \sum_{i=1}^{n} \left| \mathbb{1}\{\widehat{g}(\boldsymbol{x}_i) > 0\} - \mathbb{1}\{g^*(\boldsymbol{x}_i) > 0\} \right|$$

$$= \frac{1}{n} \sum_{i=1}^{n} \mathbb{1}\{\widehat{g}(\boldsymbol{x}_i) \neq g^*(\boldsymbol{x}_i)\} \tag{42}$$

For $n \geq \frac{3 \log(\frac{|\mathcal{G}|}{\delta})}{\varepsilon}$, by Lemma 2 we have $\frac{1}{n} \sum_{i=1}^{n} \mathbb{1}\{\widehat{g}(\boldsymbol{x}_i) \neq g^*(\boldsymbol{x}_i)\} \leq \epsilon + \mathbb{E}_{\boldsymbol{x}}[\mathbb{1}\{g^*(\boldsymbol{x}) \neq \widehat{g}(\boldsymbol{x})\}] \leq 2\epsilon$ with probability at least $1 - \delta$. Thus by inequality 42 and inequality 41 we have both

$$\frac{1}{n} \sum_{i=1}^{n} \mathbb{1}\{g^*(\boldsymbol{x}_i) > 0\} \mathbb{1}\{\widetilde{f}(\boldsymbol{x}_i) \neq y_i\} - 2\varepsilon \leq \frac{1}{n} \sum_{i=1}^{n} \mathbb{1}\{\widehat{g}(\boldsymbol{x}_i) > 0\} \mathbb{1}\{f^*(\boldsymbol{x}_i) \neq y_i\}$$

and

$$\frac{1}{n} \sum_{i=1}^{n} \mathbb{1}\{\widehat{g}(\boldsymbol{x}_i) > 0\} \mathbb{1}\{\widetilde{f}(\boldsymbol{x}_i) \neq y_i\} \leq \frac{1}{n} \sum_{i=1}^{n} \mathbb{1}\{g^*(\boldsymbol{x}_i) > 0\} \mathbb{1}\{f^*(\boldsymbol{x}_i) \neq y_i\} + 2\varepsilon.$$

with probability at least $1 - \delta$. Summing up the above inequalities and using inequality 41 we have with probability at least $1 - \delta$:

$$\frac{1}{n} \sum_{i=1}^{n} \mathbb{1}\{g^*(\boldsymbol{x}_i) > 0\} \mathbb{1}\{\widetilde{f}(\boldsymbol{x}_i) \neq y_i\} \leq \frac{1}{n} \sum_{i=1}^{n} \mathbb{1}\{g^*(\boldsymbol{x}_i) > 0\} \mathbb{1}\{f^*(\boldsymbol{x}_i) \neq y_i\} + 4\varepsilon \tag{43}$$

Next we bound for $\mathbb{E}_{\boldsymbol{x},y}[\mathbb{1}\{g^*(\boldsymbol{x}) > 0\}\{\mathbb{1}\{\widetilde{f}(\boldsymbol{x}) \neq y\} - \mathbb{1}\{f^*(\boldsymbol{x}) \neq y\}\}]$, we first define $\Delta(f^*, f, \boldsymbol{x}_i, y_i) = \{\mathbb{1}\{f^*(\boldsymbol{x}_i) \neq y_i\} - \mathbb{1}\{f(\boldsymbol{x}_i) \neq y_i\}\}$ for any $f \in \mathcal{F}$. Due to inequality (43), we have:

$$\mathbb{E}_{\boldsymbol{x}}[\mathbb{1}\{g^*(\boldsymbol{x}) > 0\}\{\mathbb{1}\{\widetilde{f}(\boldsymbol{x}) \neq y\} - \mathbb{1}\{f^*(\boldsymbol{x}) \neq y\}\}]$$

$$\leq \frac{1}{n} \sum_{i=1}^{n} \mathbb{1}\{g^*(\boldsymbol{x}_i) > 0\} \Delta(f^*, \widetilde{f}, \boldsymbol{x}_i, y_i) - \mathbb{E}_{\boldsymbol{x},y}[\mathbb{1}\{g^*(\boldsymbol{x}) > 0\} \Delta(f^*, \widetilde{f}, \boldsymbol{x}, y)\}] + 4\varepsilon \tag{44}$$

To bound $\frac{1}{n} \sum_{i=1}^{n} \mathbb{1}\{g^*(\boldsymbol{x}_i) > 0\} \Delta(f^*, \widetilde{f}, \boldsymbol{x}_i, y_i) - \mathbb{E}_{\boldsymbol{x},y}[\mathbb{1}\{g^*(\boldsymbol{x}) > 0\} \Delta(f^*, \widetilde{f}, \boldsymbol{x}, y)\}]$ with high probability over all $S_n$, we use the Bernstein inequality to achieve the fast generalization rate (similar to the proof in Theorem 1).

By the definition of $\Delta(f^*, f, \boldsymbol{x}, y)$ we have

$$\Delta^2(f^*, f, \boldsymbol{x}, y) = \mathbb{1}\{g^*(\boldsymbol{x}) > 0\}\{\mathbb{1}\{f(\boldsymbol{x}) \neq y\} - \mathbb{1}\{f^*(\boldsymbol{x}) \neq y\}\}^2 = \mathbb{1}\{g^*(\boldsymbol{x}) > 0\} \mathbb{1}\{f(\boldsymbol{x}) \neq f^*(\boldsymbol{x})\} \tag{45}$$

Let $\omega(\boldsymbol{x}) \equiv \mathbb{P}[f^*(\boldsymbol{x}) \neq y | \boldsymbol{x}] = \frac{3}{4} + \frac{\lambda(\boldsymbol{x})(2g^*(\boldsymbol{x})-1)}{2}$. We obtain the following:

$$
\begin{aligned}
&\mathbb{E}_{\boldsymbol{x},y}[\mathbb{1}\{g^*(\boldsymbol{x}) > 0\}\{\mathbb{1}\{\widetilde{f}(\boldsymbol{x}) \neq y\} - \mathbb{1}\{f^*(\boldsymbol{x}) \neq y\}\}] \\
=&\mathbb{E}_{\boldsymbol{x}}[\mathbb{1}\{g^*(\boldsymbol{x}) > 0\}\mathbb{E}_y[\{\mathbb{1}\{\widetilde{f}(\boldsymbol{x}) \neq y\} - \mathbb{1}\{f^*(\boldsymbol{x}) \neq y\}\}|\boldsymbol{x}]] \\
=&\mathbb{E}_{\boldsymbol{x}}[\mathbb{1}\{g^*(\boldsymbol{x}) > 0\}\{\omega(\boldsymbol{x})\mathbb{1}\{\widetilde{f}(\boldsymbol{x}) \neq f^*(\boldsymbol{x})\} + (1 - \omega(\boldsymbol{x}))(\mathbb{1}\{\widetilde{f}(\boldsymbol{x}) = f^*(\boldsymbol{x})\} - 1)\}|\boldsymbol{x}]] \\
=&\mathbb{E}_{\boldsymbol{x}}[\mathbb{1}\{g^*(\boldsymbol{x}) > 0\}\{(2\omega(\boldsymbol{x}) - 1)\mathbb{1}\{\widetilde{f}(\boldsymbol{x}) \neq f^*(\boldsymbol{x})\}\}] \\
\geq&\frac{1}{4}\mathbb{E}_{\boldsymbol{x}}[\mathbb{1}\{g^*(\boldsymbol{x}) > 0\}\{\mathbb{1}\{\widetilde{f}(\boldsymbol{x}) \neq f^*(\boldsymbol{x})\}\}]
\end{aligned}
\tag{46}
$$

Thus using equation 45 and inequality 46 we have

$$
\boldsymbol{Var}_{\boldsymbol{x},y}[\mathbb{1}\{g^*(\boldsymbol{x}) > 0\}\Delta(f^*, \widetilde{f}; \boldsymbol{x}, y)] \leq 4\mathbb{E}_{\boldsymbol{x},y}[\mathbb{1}\{g^*(\boldsymbol{x}) > 0\}\{\mathbb{1}\{\widetilde{f}(\boldsymbol{x}) \neq y\} - \mathbb{1}\{f^*(\boldsymbol{x}) \neq y\}\}]. \tag{47}
$$

Using an approach similar to the proof of Theorem 1 one can show that $n \geq \frac{16 \log(\frac{|\mathcal{F}|}{\delta})}{\varepsilon}$ suffices to guarantee following inequality holds with probability at least $1 - 2\delta$.

$$
\mathbb{E}_{\boldsymbol{x}}[\mathbb{1}\{g^*(\boldsymbol{x}) > 0\}\{\mathbb{1}\{\widetilde{f}(\boldsymbol{x}) \neq f^*(\boldsymbol{x})\}\}] \leq \varepsilon \tag{48}
$$

which implies that

$$
\mathbb{E}_{\boldsymbol{x}}[\{\mathbb{1}\{\widetilde{f}(\boldsymbol{x}) \neq f^*(\boldsymbol{x})\}|\mathbb{1}\{g^*(\boldsymbol{x}) > 0\}] \leq \frac{\varepsilon}{\alpha} \tag{49}
$$

$\square$

## A.6   Proof for Theorem 2

Define $\omega(\boldsymbol{x}) \equiv \mathbb{P}[f^*(\boldsymbol{x}) \neq y | \boldsymbol{x}] = \frac{3}{4} + \frac{\lambda(\boldsymbol{x})(2g^*(\boldsymbol{x})-1)}{2}$. Since $\lambda(\boldsymbol{x}) \leq \frac{1}{2} - h$, one can show $\mathbb{P}[f^*(\boldsymbol{x}) \neq y | \boldsymbol{x}]$ satisfies Massart condition with margin $\frac{h}{2}$. Since $n \geq \frac{64 \log(\frac{|\mathcal{F}|}{\delta})}{h^2\varepsilon}$, the conclusion that $\mathbb{E}_{\boldsymbol{x}}[\widehat{f}(\boldsymbol{x}) \neq f^*(\boldsymbol{x})] \leq \frac{\varepsilon}{\alpha}$ follows from 5.2-section in (Bousquet, 2004). The risk bounds on $\widehat{g}$ and $\widetilde{f}$ follows from Theorem 1 and Theorem 4. $\square$

## A.7   Proof for controlling conditional risk $\mathbb{E}_{\boldsymbol{x}}[\widehat{f}(\boldsymbol{x}) \neq f^*(\boldsymbol{x})|\boldsymbol{x} \in \Omega_I]$

**Definition 4** (Growth Function(Vapnik & Chervonenkis, 2015)). *Let $\mathcal{G}$ be the hypothesis class of function $f$ and $\mathcal{F}_{\boldsymbol{x}_1,...,\boldsymbol{x}_n} = \{(f(\boldsymbol{x}_1),...,f(\boldsymbol{x}_n)) : f \in \mathcal{F}\} \subseteq \{+1, -1\}^n$. The growth function is defined to be the maximum number of ways in which $n$ points can be classified by the function class: $\mathcal{B}_{\mathcal{F}}(n) = \sup_{\boldsymbol{x}_1,...,\boldsymbol{x}_n} |\mathcal{F}_{\boldsymbol{x}_1,...,\boldsymbol{x}_n}|$.*

**Lemma 1** (Sauer–Shelah Lemma(See (Blum et al., 2016; Mohri et al., 2018; Sauer, 1972))). *Let $d_{vc}(\mathcal{G})$ be the VC-dimension of hypothesis class $\mathcal{G}$ and let $\mathcal{B}_{\mathcal{G}}(n)$ be the growth function, for all $n \in \mathbb{N}$,*

$$
\mathcal{B}_{\mathcal{G}}(n) \leq \sum_{i=0}^{d_{vc}} \binom{n}{i} \leq \left(\frac{en}{d_{vc}(\mathcal{G})}\right)^{d_{vc}(\mathcal{G})}
$$

**Theorem 5.** *For every $\varepsilon > 0$, $\delta > 0$, under Assumption 1, given a set of samples $S_n = \{(\boldsymbol{x}_1, y_1), ..., (\boldsymbol{x}_n, y_n)\}$ drawn i.i.d. from the Noisy Generative Process and*

$$
\widehat{f} = \arg\min_{f \in \mathcal{F}} \sum_{i=1}^{n} \mathbb{1}\{f(\boldsymbol{x}_i) \neq y_i\},
$$

*if $n$ is chosen such that*

$$
n \geq \frac{32\left[d_{VC}(\mathcal{F})\log(\frac{1}{\varepsilon}) + \log(\frac{1}{\delta})\right]}{\epsilon^2 \alpha^2},
$$

*then with probability at least $1 - 2\delta$ we have:*

$$
\mathbb{E}_{\boldsymbol{x},y}[\widehat{f}(\boldsymbol{x}) \neq y] \leq \frac{1}{2}(1 - \alpha) + 2\epsilon\alpha.
$$

*Furthermore the following holds:*

$$
\mathbb{P}_{\boldsymbol{x}}[f^*(\boldsymbol{x}) \neq \widehat{f}(\boldsymbol{x})|\boldsymbol{x} \in \Omega_I] \leq 2\epsilon
$$

*Proof.* We first bound the probability of the event that $\mathbb{E}_{\boldsymbol{x},y}[\widehat{f}(\boldsymbol{x}) \neq y] \leq \frac{1}{2}(1-\alpha) + 2\epsilon\alpha$.

By Lemma 3 we have

$$\mathbb{P}_{S_n}[\sup_{f \in \mathcal{F}} |\frac{1}{n}\sum_{i=1}^{n} \mathbb{1}\{f(\boldsymbol{x}_i) \neq y_i\} - \mathbb{E}_{\boldsymbol{x},y}[\mathbb{1}\{f(\boldsymbol{x}) \neq y\}]| \geq t] \leq 4\mathcal{B}_{\mathcal{F}}(2n)e^{-\frac{nt^2}{32}} \tag{50}$$

By setting $t = \alpha\epsilon$ and $n \geq \frac{32(\log(4\mathcal{B}_{\mathcal{F}}(2n))+\log(\frac{1}{\delta}))}{\alpha^2\epsilon^2}$ we have $4\mathcal{B}_{\mathcal{F}}(2n)e^{-\frac{nt^2}{32}} \leq \delta$ and so using inequality 50 we have with probability of at least $1 - \delta$:

$$|\frac{1}{n}\sum_{i=1}^{n} \mathbb{1}\{\widehat{f}(\boldsymbol{x}_i) \neq y_i\} - \mathbb{E}_{\boldsymbol{x},y}[\mathbb{1}\{\widehat{f}(\boldsymbol{x}) \neq y\}]| \leq \alpha\epsilon$$

The term $\mathcal{B}_{\mathcal{F}}(2n)$ could be bounded by Lemma 1 as $\mathcal{B}_{\mathcal{F}}(2n) \leq \left(\frac{2en}{d_{VC}(\mathcal{F})}\right)^{d_{VC}(\mathcal{F})}$. Next we apply the fact that $\widehat{f} = \arg\min_{f \in \mathcal{F}} \frac{1}{n}\sum_{i=1}^{n} \mathbb{1}\{\widehat{f}(\boldsymbol{x}_i) \neq y_i\}$. We have:

$$\mathbb{E}_{\boldsymbol{x},y}[\mathbb{1}\{\widehat{f}(\boldsymbol{x}) \neq y\}] \leq \alpha\epsilon + \frac{1}{n}\sum_{i=1}^{n} \mathbb{1}\{\widehat{f}(\boldsymbol{x}_i) \neq y_i\} \leq \alpha\epsilon + \frac{1}{n}\sum_{i=1}^{n} \mathbb{1}\{f^*(\boldsymbol{x}_i) \neq y_i\} \tag{51}$$

By Lemma 5 we have $\frac{1}{n}\sum_{i=1}^{n} \mathbb{1}\{f^*(\boldsymbol{x}_i) \neq y_i\} \leq \frac{1}{2}(1-\alpha) + \epsilon\alpha$ with failure probability at most $\delta$. Combining this with inequality 51 we have with probability at least $1 - 2\delta$:

$$\mathbb{E}_{\boldsymbol{x},y}[\mathbb{1}\{\widehat{f}(\boldsymbol{x}) \neq y\}] \leq \frac{1}{2}(1-\alpha) + 2\epsilon\alpha.$$

And this finishes the proof of the first claim. Next we prove the claim that:

$$\mathbb{P}_{\boldsymbol{x}\sim\mathcal{D}_I}[f^*(\boldsymbol{x}) \neq \widehat{f}(\boldsymbol{x})] \leq 2\epsilon.$$

First note that $\mathbb{P}_{\boldsymbol{x}\sim\mathcal{D}_I}[f^*(\boldsymbol{x}) \neq \widehat{f}(\boldsymbol{x})] = \mathbb{P}_{\boldsymbol{x}\sim\mathcal{D}_\alpha}[\mathbb{1}\{\widehat{f}(\boldsymbol{x}) \neq f^*(\boldsymbol{x})\}|\boldsymbol{x} \in \Omega_I]$. Using $\mathbb{E}_{\boldsymbol{x},y}[\mathbb{1}\{\widehat{f}(\boldsymbol{x}) \neq y\}] \leq \frac{1}{2}(1-\alpha) + 2\epsilon\alpha$ below we have:

$$\begin{aligned}
&\mathbb{E}_{\boldsymbol{x},y}[\mathbb{1}\{\widehat{f}(\boldsymbol{x}) \neq y\}] \\
=&\mathbb{E}_{(\boldsymbol{x},y)\sim\mathcal{D}_\alpha}[\mathbb{1}\{\widehat{f}(\boldsymbol{x}) \neq y\}] \\
=&\underbrace{\mathbb{E}_{(\boldsymbol{x},y)\sim\mathcal{D}_\alpha}[\mathbb{1}\{\widehat{f}(\boldsymbol{x}) \neq y\}|\boldsymbol{x} \in \Omega_U]}_{\frac{1}{2}}\underbrace{\mathbb{P}_{(\boldsymbol{x},y)\sim\mathcal{D}_\alpha}[\boldsymbol{x} \in \Omega_U]}_{1-\alpha} \\
&+\underbrace{\mathbb{E}_{(\boldsymbol{x},y)\sim\mathcal{D}_\alpha}[\mathbb{1}\{\widehat{f}(\boldsymbol{x}) \neq y\}|\boldsymbol{x} \in \Omega_I]}_{\mathbb{P}_{(\boldsymbol{x},y)\sim\mathcal{D}_\alpha}[\mathbb{1}\{\widehat{f}(\boldsymbol{x})\neq f^*(\boldsymbol{x})\}|\boldsymbol{x}\in\Omega_I]}\underbrace{\mathbb{P}_{(\boldsymbol{x},y)\sim\mathcal{D}_\alpha}[\boldsymbol{x} \in \Omega_I]}_{\alpha} \\
=&\frac{1}{2}(1-\alpha) + \alpha\mathbb{P}_{\boldsymbol{x}\sim\mathcal{D}_\alpha}[\widehat{f}(\boldsymbol{x}) \neq f^*(\boldsymbol{x})|\boldsymbol{x} \in \Omega_I] \\
\leq&\frac{1}{2}(1-\alpha) + 2\epsilon\alpha \\
\Longrightarrow&\mathbb{P}_{\boldsymbol{x}\sim\mathcal{D}_\alpha}[\mathbb{1}\{\widehat{f}(\boldsymbol{x}) \neq f^*(\boldsymbol{x})\}|\boldsymbol{x} \in \Omega_I] \leq 2\epsilon
\end{aligned} \tag{52}$$

This finishes the proof of the second claim. $\qquad\square$

## B Extention to VC-Class

In order to leverage the margin condition of distribution of $z$ to obtain a minimax-optimal generalization rate, we leverage on the Local Rademacher Average tool. Our analysis tool largely follows from (Bousquet et al., 2003; Bartlett et al., 2005). Throughout this section, $\lesssim$ and $\gtrsim$ represent as shorthand for the $\leq$ and $\geq$ that ignores universal constants.

**Definition 5** ($L_2$-Covering Number). *(Wellner et al., 2013) Let $\boldsymbol{x}_{1:n}$ be set of points. A set of $U \subseteq \mathbb{R}^n$ is an $\varepsilon$-cover w.r.t $L_2$-norm of $\mathcal{F}$ on $x_{1:n}$, if $\forall f \in \mathcal{F}$, $\exists u \in U$, s.t. $\sqrt{\frac{1}{n} \sum_{i=1}^n |[u]_i - f(x_i)|^2} \leq \varepsilon$, where $[u]_i$ is the i-th coordinate of u. We define the covering number $\mathcal{N}_2(\varepsilon, \mathcal{F}, \boldsymbol{x}_{1:n})$ :*

$$\mathcal{N}_2(\varepsilon, \mathcal{F}, \boldsymbol{x}_{1:n}) := \min\{|U| : U \text{ is an } \varepsilon\text{-cover of } \mathcal{F} \text{ on } x_{1:n}\}$$

*Let $\mathcal{N}_2(\varepsilon, \mathcal{F}, n)$ be the maximum cardinality of $\mathcal{N}_2(\varepsilon, \mathcal{F}, \boldsymbol{x}_{1:n})$ over all $\boldsymbol{x}_{1:n}$. Formally $\mathcal{N}_2(\varepsilon, \mathcal{F}, n)$ is defined as:*

$$\mathcal{N}_2(\varepsilon, \mathcal{F}, n) := \sup_{\boldsymbol{x}_{1:n} \in \mathcal{X}^n} \min\{|U| : U \text{ is an } \varepsilon\text{-cover of } \mathcal{F} \text{ on } x_{1:n}\}$$

**Definition 6** (Local Rademacher Average (Bartlett et al., 2005; Bousquet et al., 2003)). *Let $\sigma_{1:n}$ be Rademacher sequence of length n, given samples $\{\boldsymbol{x}_i, y_i\}_{i=1}^n$ the Empirical Local Rademacher Complexity at distributional and empirical radius $r \geq 0$ for the class $\mathcal{F}$ are defined as*

$$\mathcal{R}_n(\mathcal{F}, Pf^2 \leq r) \equiv \mathbb{E}_{\sigma_{1:n}}\left[\sup_{f \in \mathcal{F}, \mathbb{E}_{\boldsymbol{x}}[f(\boldsymbol{x})^2] \leq r} \frac{1}{n} \sum_{i=1}^n \sigma_i f(\boldsymbol{x}_i)\right]$$

$$\mathcal{R}_n(\mathcal{F}, P_n f^2 \leq r) \equiv \mathbb{E}_{\sigma_{1:n}}\left[\sup_{f \in \mathcal{F}, \frac{1}{n}\sum_{i=1}^n f(\boldsymbol{x}_i)^2 \leq r} \frac{1}{n} \sum_{i=1}^n \sigma_i f(\boldsymbol{x}_i)\right]$$

*and their distributional Average as: $\mathcal{R}(\mathcal{F}, Pf^2 \leq r) \equiv \mathbb{E}_{S_n}[\mathcal{R}_n(\mathcal{F}, Pf^2 \leq r)]$ and $\mathcal{R}(\mathcal{F}, P_n f^2 \leq r) \equiv \mathbb{E}_{S_n}[\mathcal{R}_n(\mathcal{F}, P_n f^2 \leq r)]$.*

**Definition 7** (Star Hull). *(Bartlett et al., 2005; Bousquet et al., 2003) The star hull of set of functions $\mathcal{F}$ is defined as*

$$*\mathcal{F} \equiv \{\alpha f : f \in \mathcal{F}, \alpha \in [0,1]\}$$

**Definition 8** (Sub-Root Function). *(Bartlett et al., 2005; Massart & Nédélec, 2006; Bousquet et al., 2003) A function $\psi : \mathbb{R} \to \mathbb{R}$ is sub-root if*

- *$\psi$ is non-decreasing*

- *$\psi$ is non-negative*

- *$\psi(r)/\sqrt{r}$ is non-increasing for any $r > 0$*

*And we say $r^*$ is a fixed point of $\psi$ if $\psi(r^*) = r^*$.*

**Theorem 6.** *[Risk Bound VC-Class] Let $S_n = \{(\boldsymbol{x}_i, y_i)\}_{i=1}^n$ be i.i.d samples from Data Generative Process described in Definition 1 under Assumption 1, with $f^*(\cdot) \in \mathcal{F}$ and $g^*(\cdot) \in \mathcal{G}$ with VC-dimension $d_{VC}(\mathcal{F}) < \infty$ $d_{VC}(\mathcal{G}) < \infty$. Given $\bar{\lambda}$, let $\beta \in \left[\frac{3-2\bar{\lambda}}{1+2\bar{\lambda}} + \bar{\lambda}, min(\frac{3+2\bar{\lambda}}{1-2\bar{\lambda}} - \frac{\bar{\lambda}}{1-4\bar{\lambda}^2}, 10)\right]$. For any $\widehat{f}(\cdot) \in \mathcal{F}$, let $\widehat{g} = \arg\min_{g \in \mathcal{G}} R_{S_n}(g; \widehat{f}, \beta)$. Then for any $\varepsilon > 0$, $\delta > 0$ such that the following holds: For*

$$n \gtrsim max\left\{\frac{\beta^4 d_{VC}(\mathcal{G})\log(\frac{1}{\varepsilon}) + \beta^4 \log(\frac{1}{\delta})}{\bar{\lambda}\varepsilon}, \frac{\beta d_{VC}(\mathcal{F})\log(\frac{d_{VC}(\mathcal{F})}{\varepsilon}) + \beta \log(\frac{1}{\delta}))}{\varepsilon}\right\}.$$

*and for $\widehat{f}$ that satisfies one of the following condition:*

- *For any $\widehat{f}(\cdot) \in \mathcal{F}$ that satisfies $\mathbb{E}_{\boldsymbol{x}}[\widehat{f}(\boldsymbol{x}) \neq f^*(\boldsymbol{x})] \lesssim \frac{\varepsilon}{\beta}$ with probability at least $1 - \delta$,*

- *For any $\widehat{f}(\cdot) \in \mathcal{F}$ that satisfies $\mathbb{E}_{\boldsymbol{x}}[\widehat{f}(\boldsymbol{x}) \neq f^*(\boldsymbol{x})|\boldsymbol{x} \in \Omega_I] \lesssim \frac{\varepsilon}{\beta\alpha}$ with probability at least $1 - \delta$,*

*The following holds with probability at least $1 - 3\delta$:*

$$R(\widehat{g}; f^*, \beta) - R(g^*; f^*, \beta) \lesssim \varepsilon$$

*Proof.* The major difference from the proof for Theorem 1 is the fact that $\mathcal{G}$ and $\mathcal{F}$ are not finite hypothesis classes. To achieve fast generalization rate, we leverage the Local Rademacher Complexity Tool from (Bartlett et al., 2005).

**CASE I:** $\widehat{f}(\cdot) \in \mathcal{F}$ and $\mathbb{E}_{\boldsymbol{x}}[\widehat{f}(\boldsymbol{x}) \neq f^*(\boldsymbol{x})] \lesssim \frac{\varepsilon}{\beta}$ with probability at least $1 - \delta$.

We achieve equation 30 which is restated below exactly the same as in Theorem 1.

$$R_{S_n}(\widehat{g}; \widehat{f}, \beta) \geq R_{S_n}(\widehat{g}; f^*, \beta) - \frac{\beta - 1}{n} \sum_{i=1}^n \mathbb{1}\left\{\widehat{f}(\boldsymbol{x}_i) \neq f^*(\boldsymbol{x}_i)\right\}$$

Then to achieve inequality 31, we invoke Lemma 8 instead of Lemma 2 as $\mathcal{F}$ is a VC-class. In particular, since $n \gtrsim \frac{\beta(d_{VC}(\mathcal{F}) \log(\frac{1}{\varepsilon}) + \log(\frac{1}{\delta}))}{\varepsilon}$, we have that $\frac{1}{n} \sum_{i=1}^n \mathbb{1}\{f(\boldsymbol{x}_i) \neq f^*(\boldsymbol{x}_i)\} \leq \mathbb{E}_{\boldsymbol{x}} \mathbb{1}\{f(\boldsymbol{x}) \neq f^*(\boldsymbol{x})\} + \varepsilon$. Moreover, from the case assumption we have $\mathbb{E}_{\boldsymbol{x}}[\widehat{f}(\boldsymbol{x}) \neq f^*(\boldsymbol{x})] \lesssim \frac{\varepsilon}{\beta}$ with probability at least $1 - \delta$. So with probability at least $1 - \delta$

$$R_{S_n}(\widehat{g}; \widehat{f}, \beta) \geq R_{S_n}(\widehat{g}; f^*, \beta) - \varepsilon/4.$$

Similarly we get that with probability at least $1 - \delta$

$$R_{S_n}(g^*; \widehat{f}, \beta) \leq R_{S_n}(g^*; f^*, \beta) + \varepsilon/4.$$

Thus following hold with probability at least $1 - \delta$:

$$R_{S_n}(\widehat{g}; f^*, \beta) \leq R_{S_n}(g^*; f^*, \beta) + \frac{\varepsilon}{2}. \tag{53}$$

Next we turn to bound the risk gap using $R(\widehat{g}; f^*, \beta) - R(g^*; f^*, \beta)$ using the concentration property of inequality 53. Similar to the proof in Theorem 1, for any $f \in \mathcal{F}, g \in \mathcal{G}$ we define $\ell(g; f, \boldsymbol{x}, y) = \beta \mathbb{1}\{g(\boldsymbol{x}) = 1\} \mathbb{1}\{f(\boldsymbol{x}) \neq y\} + \mathbb{1}\{g(\boldsymbol{x}) \neq 1\} \mathbb{1}\{f(\boldsymbol{x}) = y\}$. Based on $\ell$, we define following hypothesis class:

$$\Delta \circ \ell \circ \mathcal{G} \equiv \left\{ \Delta \ell(g; g^*, \boldsymbol{x}, y) = \ell(g; f^*, \boldsymbol{x}, y) - \ell(g^*; f^*, \boldsymbol{x}, y) : g \in \mathcal{G} \right\}. \tag{54}$$

To invoke Lemma 6, we need to establish some hypothesis class $\mathcal{H}$ that satisfies condition $\boldsymbol{Var}[h] \leq B\mathbb{E}[h]$. Next we show $\Delta \circ \ell \circ \mathcal{G}$ satisfies the condition that $\boldsymbol{Var}[h] \leq B\mathbb{E}[h]$ and thus we can apply Lemma 6 with $\mathcal{H} = \Delta \circ \ell \circ \mathcal{G}$. To begin with, using the same approach as in Theorem 1 we can obtain Equation 36 which is restated below.

$$\Delta^2 \ell(g^*, g, \boldsymbol{x}, y) \leq \beta^2 \mathbb{1}\{g^*(\boldsymbol{x}) \neq g(\boldsymbol{x})\}$$

By Equation 36 we have that $\boldsymbol{Var}_{\boldsymbol{x}, y}[\Delta \ell(g^*, g, \boldsymbol{x}, y)] \leq \mathbb{E}_{\boldsymbol{x}, y} \Delta^2 \ell(g^*, g, \boldsymbol{x}, y) \leq \beta^2 \mathbb{E}_{\boldsymbol{x}}[\mathbb{1}\{g^*(\boldsymbol{x}) \neq g(\boldsymbol{x})\}]$.

On the other hand, Equation 16 implies that

$$R(g; f^*, \beta) - R(g^*; f^*, \beta) \geq \frac{\bar{\lambda}}{1 + 2\bar{\lambda}} \mathbb{E}_{\boldsymbol{x}}[\mathbb{1}\{g^*(\boldsymbol{x}) \neq g(\boldsymbol{x})\}].$$

Thus we have following holds:

$$\boldsymbol{Var}_{\boldsymbol{x}, y}[\Delta \ell(g; g^*, \boldsymbol{x}, y)] \leq \frac{\beta^2(1 + 2\bar{\lambda})}{\bar{\lambda}} \mathbb{E}_{\boldsymbol{x}, y}\{\Delta \ell(g; g^*, \boldsymbol{x}, y)\} \tag{55}$$

Above variance bound allows invoking Lemma 6 with $\mathcal{H} = \Delta \circ \ell \circ \mathcal{G}$, $T(h) = \mathbb{E}[h^2]$ and $B = \frac{\beta^2(1 + 2\bar{\lambda})}{\bar{\lambda}}$. To satisfy the rest of the assumptions of Lemma 6, we find a subroot function $\psi(r)$ that

$$\psi(r) \geq \frac{\beta^2(1 + 2\bar{\lambda})}{\bar{\lambda}} \mathbb{E}_{\mathcal{S}_n}[\mathcal{R}_n\{\Delta \ell(g; g^*) \in \mathcal{H} : \mathbb{E}[h^2] \leq r\}].$$

Where we shorten $\Delta \ell(g; g^*, \boldsymbol{x}, y)$ to $\Delta \ell(g; g^*)$. To find $\psi(r)$, we show some analysis on the Local Rademacher Average $\mathbb{E}_{\mathcal{S}_n}[\mathcal{R}_n\{\Delta \ell(g; g^*) \in \mathcal{H} : \mathbb{E}[h^2] \leq r\}]$ as follows.

$$\mathbb{E}_{\mathcal{S}_n}[\mathcal{R}_n(\Delta \circ \ell \circ \mathcal{G}, r)] = \mathbb{E}_{S_n, \sigma_{1:n}}\Big[\sup_{g \in \mathcal{G}, \mathbb{E}_{\boldsymbol{x}, y}[\Delta^2 \ell(g; g^*)] \leq r} \frac{1}{n} \sum_{i=1}^n \sigma_i \Delta \ell(g; g^*, \boldsymbol{x}_i, y_i)\Big]$$

$$\leq \underbrace{\mathbb{E}_{S_n, \sigma_{1:n}}\Big[\sup_{g \in \mathcal{G}, \mathbb{E}_{\boldsymbol{x}}[\mathbb{1}(g \neq g^*)] \leq r} \frac{1}{n} \sum_{i=1}^n \sigma_i \Delta \ell(g; g^*, \boldsymbol{x}_i, y_i)\Big]}_{\mathbb{1}(g \neq g^*) \leq \Delta^2 \ell(g; g^*)}$$

(56)

$$\leq \beta \underbrace{\mathbb{E}_{S_n, \sigma_{1:n}}\Big[\sup_{g \in \mathcal{G}, \mathbb{E}_{\boldsymbol{x}}[\mathbb{1}(g \neq g^*)] \leq r} \frac{1}{n} \sum_{i=1}^n \sigma_i \mathbb{1}\{g(\boldsymbol{x}_i) \neq g^*(\boldsymbol{x}_i)\}\Big]}_{\substack{|\Delta \ell(g_1; g^*) - \Delta \ell(g_2; g^*)| \leq \beta |\mathbb{1}(g_1 \neq g^*) - \mathbb{1}(g_2 \neq g^*)| \\ \text{Talagrand Contraction Inequality (Ledoux \& Talagrand, 1991)}}}$$

In the last inequality we use Talagrand Contraction Inequality (Ledoux & Talagrand, 1991) for which we need to show that (1) $\Delta \ell(g_1; g^*)$ is a function of $\mathbb{1}\{g_1 \neq g^*\}$ and (2) $\Delta \ell(g_1; g^*)$ is $\beta$ Lipschitz in $\mathbb{1}\{g_1 \neq g^*\}$. These conditions are satisfied as follows:

$$\Delta \ell(g; g^*) = \mathbb{1}\{g^* \neq g\} \Delta \ell(g; g^*)$$

and

$$|\Delta \ell(g_1; g^*) - \Delta \ell(g_2; g^*)| \leq |\ell(g_1, f^*) - \ell(g_2, f^*)| \leq \beta |\mathbb{1}(g_1 \neq g_2)| = \beta |\mathbb{1}(g_1 \neq g^*) - \mathbb{1}(g_2 \neq g^*)|.$$

This finishes the proof of Inequality 56. Now define the class $\mathbb{1} \circ \mathcal{G} \equiv \mathbb{1}\{g(\boldsymbol{x}) \neq g^*(\boldsymbol{x}), g \in \mathcal{G}\}$. The indicator function $\mathbb{1}\{g(\boldsymbol{x}) \neq g^*(\boldsymbol{x})\}$ is a Boolean function taking $g$ as input, thus $d_{VC}(\mathbb{1} \circ \mathcal{G}) \leq d_{VC}(\mathcal{G})$. Thus we have

$$\frac{\beta^2(1 + 2\bar{\lambda})}{\bar{\lambda}} \mathbb{E}_{\mathcal{S}_n}[\mathcal{R}_n\{\Delta \ell(g; g^*) \in \mathcal{H} : \mathbb{E}[h^2] \leq r\}]$$

$$\leq \frac{\beta^3(1 + 2\bar{\lambda})}{\bar{\lambda}} \mathbb{E}_{\mathcal{S}_n}[\mathcal{R}_n\{\mathbb{1}\{g(\boldsymbol{x}) \neq g^*(\boldsymbol{x})\} \in \mathbb{1} \circ \mathcal{G} : \mathbb{E}[\mathbb{1}\{g(\boldsymbol{x}) \neq g^*(\boldsymbol{x})\}] \leq r\}]$$

(57)

Above implies that we can pick $\psi(r)$ to be

$$\psi(r) = \frac{\beta^3(1 + 2\bar{\lambda})}{\bar{\lambda}} \mathbb{E}_{\mathcal{S}_n}[\mathcal{R}_n\{* \mathbb{1} \circ \mathcal{G} : \mathbb{E}[\mathbb{1}\{g(\boldsymbol{x}) \neq g^*(\boldsymbol{x})\}] \leq r\}] + \frac{11\beta^2 \log n}{n}$$

(58)

By Equation 80 in Lemma 6, we have:

$$\mathbb{E}_{\boldsymbol{x}, y}[\Delta \ell(\widehat{g}; g^*, \boldsymbol{x}, y)] \leq \frac{2}{n} \sum_{i=1}^n \Delta \ell(\widehat{g}; g^*; \boldsymbol{x}_i, y_i) + \frac{1500\bar{\lambda}}{\beta^2} r^* + \frac{\log(1/\delta)(11\beta + \frac{52}{\bar{\lambda}})}{n}$$

(59)

Where $r^*$ is the fixed point of $\psi(r)$. By inequality 53, we have $\frac{1}{n} \sum_{i=1}^n \Delta \ell(\widehat{g}; g^*; \boldsymbol{x}_i, y_i) = R_{S_n}(\widehat{g}; f^*, \beta) - R_{S_n}(g^*; f^*, \beta) \leq \varepsilon/2$ holds with probability $1 - \delta$. By Lemma 7 we have $r^* \lesssim \frac{\beta^6}{\bar{\lambda}^2} \frac{d_{VC}(\mathcal{G}) \log n}{n}$. Plugging in Equation 59 we have that $n \gtrsim \frac{\beta^4(d_{VC}(\mathcal{G}) \log(\frac{1}{\varepsilon}) + \log(1/\delta))}{\bar{\lambda}\varepsilon}$ suffices to achieve $\mathbb{E}_{\boldsymbol{x}, y}[\Delta \ell(\widehat{g}; g^*, \boldsymbol{x}, y)] \lesssim \varepsilon$. Recall that $\mathbb{E}_{\boldsymbol{x}, y}[\Delta \ell(\widehat{g}; g^*, \boldsymbol{x}, y)] = R(\widehat{g}; f^*, \beta) - R(g^*; f^*, \beta)$ so this finishes the proof.

**CASE II:** $\widehat{f}(\cdot) \in \mathcal{F}$ that satisfies $\mathbb{E}_{\boldsymbol{x}}[\widehat{f}(\boldsymbol{x}) \neq f^*(\boldsymbol{x})|\boldsymbol{x} \in \Omega_I] \leq \frac{\varepsilon}{8\alpha\beta}$ with probability at least $1 - \delta$.
The proof is similar to the one in Theorem 1 except for that we need to bound the VC-dimension of pseudo hypothesis class $\widetilde{\mathcal{F}}$. Since $\widetilde{f}$ can be viewed as Boolean function given $f_1(\boldsymbol{x}), f_2(\boldsymbol{x})$ as input, with two hypothesis $f_1, f_2 \in \mathcal{F}$, by Lemma 3.2.3 in (Blumer et al., 1989) we know $d_{VC}(\widetilde{\mathcal{F}}) \leq 2d_{VC}(\mathcal{F}) \log(d_{VC}(\mathcal{F}))$. The rest of the proof follows from the one in Theorem 1. □

*Next we present our extension of information theoretic lower bound to VC-class. The lower bounds suggest that the risk bound in Theorem 6 is tight up to some logarithmic factor.*

**Theorem 7.** *Consider the noisy generative process defined in Definition 1 with $\Omega_{\mathcal{D}}$ being $\mathcal{G}$-realizable. Then for any $\varepsilon \leq \bar{\lambda}$, to achieve*

$$\mathbb{E}_{S_n}[R(\mathcal{A}(S_n), f^*, \beta) - R(g^*, f^*, \beta)] \leq \frac{\varepsilon}{8(1 + 2\bar{\lambda})}$$

*with $\beta \in \left[\frac{3-2\bar{\lambda}}{1+2\bar{\lambda}} + \bar{\lambda}, \frac{3+2\bar{\lambda}}{1-2\bar{\lambda}} - \frac{\bar{\lambda}}{1-4\bar{\lambda}^2}\right]$ , any algorithm $\mathcal{A}$ will take at least $\frac{d_{VC}(\mathcal{G})}{\log(d_{VC}(\mathcal{G}))\bar{\lambda}\varepsilon}$ many samples.*

*Proof.* The proof follows from the proof of Theorem 3 except for the fact that we need to have an upper bound on the VC-dimension of our hypothesis construction $\mathcal{G}$. Since $\mathcal{G}$ consists of composition of interval hypothesis and each individual interval has VC-dimension at most 3. By Lemma 3.2.3 in (Blumer et al., 1989) we know $d_{VC}(\mathcal{G}) \leq 6d\log(d)$ which implies a $\frac{d_{VC}(\mathcal{G})}{\log(d_{VC}(\mathcal{G}))\bar{\lambda}\varepsilon}$ lower bound. $\qquad\square$

**Theorem 8** (Improving Predictor Selective Risk for VC-Class)**.** *Let $S_n\{(\boldsymbol{x}_i, y_i)\}_{i=1}^n$ be i.i.d samples from the Data Generative Process described in Definition 1 under Assumption 1, with $f^*(\cdot) \in \mathcal{F}$ and $g^*(\cdot) \in \mathcal{G}$, $d_{VC}(\mathcal{F}) < \infty, d_{VC}(\mathcal{G}) < \infty$. For any $\widehat{g}(\cdot) \in \mathcal{G}$ s.t., $\mathbb{E}_{\boldsymbol{x}}[g^*(\boldsymbol{x}) \neq \widehat{g}(\boldsymbol{x})] \leq \varepsilon$, let $\widetilde{f} = \arg\min_{f \in \mathcal{F}} \frac{1}{n}\sum_{i=1}^n \mathbb{1}\{\widehat{g}(\boldsymbol{x}_i) > 0\}\mathbb{1}\{f(\boldsymbol{x}_i) \neq y_i\}$. Then for any $\varepsilon > 0$, $\delta > 0$ such that the following holds: if*

$$n \gtrsim max\left\{\frac{d_{VC}(\mathcal{F})\log(\frac{d_{VC}(\mathcal{F})}{\varepsilon}) + \log(\frac{1}{\delta})}{\varepsilon}, \frac{d_{VC}(\mathcal{G})\log(\frac{d_{VC}(\mathcal{G})}{\varepsilon}) + \log(\frac{1}{\delta}))}{\varepsilon}\right\},$$

*we have*

$$\mathbb{E}_{\boldsymbol{x}}[\widetilde{f}(\boldsymbol{x}) \neq f^*(\boldsymbol{x})|g^*(\boldsymbol{x}) \neq 1] \leq \frac{\varepsilon}{\alpha} \tag{60}$$

*Proof.* We use a proof similar to the one in Theorem 4 up to Equation (42), restated below: For any $f$:

$$\left|\frac{1}{n}\sum_{i=1}^n \mathbb{1}\{\widehat{g}(\boldsymbol{x}_i) > 0\}\mathbb{1}\{f(\boldsymbol{x}_i) \neq y_i\} - \frac{1}{n}\sum_{i=1}^n \mathbb{1}\{g^*(\boldsymbol{x}_i) > 0\}\mathbb{1}\{f(\boldsymbol{x}_i) \neq y_i\}\right|$$

$$\leq \frac{1}{n}\sum_{i=1}^n \mathbb{1}\{\widehat{g}(\boldsymbol{x}_i) \neq g^*(\boldsymbol{x}_i)\} \tag{61}$$

Since $\mathcal{G}$ is a VC-class, we will invoke Lemma 8 instead of Lemma 2. Since $n \gtrsim \frac{d_{VC}(\mathcal{G})\log(\frac{d_{VC}(\mathcal{G})}{\varepsilon}) + \log(\frac{1}{\delta})}{\varepsilon}$, by Lemma 8 we have $\frac{1}{n}\sum_{i=1}^n \mathbb{1}\{\widehat{g}(\boldsymbol{x}_i) \neq g^*(\boldsymbol{x}_i)\} \leq \mathbb{E}_{\boldsymbol{x}}[\mathbb{1}\{\widehat{g}(\boldsymbol{x}) \neq g^*(\boldsymbol{x})\}] + \epsilon$ with probability at least $1 - \delta$. Since $\mathbb{E}_{\boldsymbol{x}}[\mathbb{1}\{\widehat{g}(\boldsymbol{x}) \neq g^*(\boldsymbol{x})\}] \leq \epsilon$, by inequality 61 we have the following with probability at least $1 - \delta$.

$$\frac{1}{n}\sum_{i=1}^n \mathbb{1}\{g^*(\boldsymbol{x}_i) > 0\}\mathbb{1}\{\widetilde{f}(\boldsymbol{x}_i) \neq y_i\} \leq \frac{1}{n}\sum_{i=1}^n \mathbb{1}\{g^*(\boldsymbol{x}_i) > 0\}\mathbb{1}\{f^*(\boldsymbol{x}_i) \neq y_i\} + 2\varepsilon \tag{62}$$

Similar to the proof of Theorem 6, we next establish some hypothesis class $\mathcal{H}$ that satisfies condition $\boldsymbol{Var}[h] \leq B\mathbb{E}[h]$. We define:

$$\mathcal{G} \cdot \Delta \circ \mathcal{F} \equiv \left\{\mathbb{1}\{g^*(\boldsymbol{x}_i) > 0\}\Delta(f; f^*, \boldsymbol{x}, y) : f \in \mathcal{F}\right\}. \tag{63}$$

To invoke Lemma 6, we need to establish some hypothesis class $\mathcal{H}$ that satisfies condition $\boldsymbol{Var}[h] \leq B\mathbb{E}[h]$. Next we show $\mathcal{G} \cdot \Delta \circ \mathcal{F}$ satisfies this condition and thus we can apply Lemma 6 with $\mathcal{H} = \mathcal{G} \cdot \Delta \circ \mathcal{F}$. Recall Equation (47) in Theorem 4 shows that,

$$\boldsymbol{Var}_{\boldsymbol{x},y}[\mathbb{1}\{g^*(\boldsymbol{x}) > 0\}\Delta(\widetilde{f}, f^*; \boldsymbol{x}, y)] \leq 4\mathbb{E}_{\boldsymbol{x},y}[\mathbb{1}\{g^*(\boldsymbol{x}) > 0\}\{\mathbb{1}\{\widetilde{f}(\boldsymbol{x}) \neq y\} - \mathbb{1}\{f^*(\boldsymbol{x}) \neq y\}\}].$$

where $\Delta(f_1, f_2, \boldsymbol{x}_i, y_i) = \{\mathbb{1}\{f_1(\boldsymbol{x}_i) \neq y_i\} - \mathbb{1}\{f_2(\boldsymbol{x}_i) \neq y_i\}\}$. So $\mathcal{H}$ satisfies the condition that $\boldsymbol{Var}[h] \leq B\mathbb{E}[h]$ for $B = 4$. To invoke Lemma 6 we let $\mathcal{H} = \mathcal{G} \cdot \Delta \circ \mathcal{F}$, $T[h] = \mathbb{E}[h^2]$ and $B = 4$. Similar to the proof in Theorem 6 we next find a subroot function $\psi(r)$ that

$$\psi(r) \geq 4\mathbb{E}_{S_n}[\mathcal{R}_n\{\mathbb{1}\{g^*(\boldsymbol{x}_i) > 0\}\Delta(f; f^*) \in \mathcal{H} : \mathbb{E}[h^2] \leq r\}].$$

To find $\psi(r)$, we show some analysis on the Local Rademacher Average.

$$
\begin{aligned}
\mathbb{E}_{S_n}[\mathcal{R}_n(\mathcal{G} \cdot \Delta \circ \mathcal{F}, r)] =& \mathbb{E}_{S_n, \sigma_{1:n}}\Big[\sup_{f \in \mathcal{F}, \mathbb{E}_{\boldsymbol{x},y}[\mathbb{1}\{g^*(\boldsymbol{x})>0\}\Delta^2(f;f^*)] \leq r} \frac{1}{n}\sum_{i=1}^n \sigma_i \mathbb{1}\{g^*(\boldsymbol{x}_i) > 0\}\Delta(f;f^*;x_i,y_i)\Big] \\
\leq & \underbrace{\mathbb{E}_{S_n, \sigma_{1:n}}\Big[\sup_{f \in \mathcal{F}, \mathbb{E}_{\boldsymbol{x}}[\mathbb{1}\{g^*(\boldsymbol{x})>0\}\mathbb{1}(f \neq f^*)] \leq r} \frac{1}{n}\sum_{i=1}^n \sigma_i \mathbb{1}\{g^*(\boldsymbol{x}_i) > 0\}\Delta(f;f^*;x_i,y_i)\Big]}_{\mathbb{1}\{g^*(\boldsymbol{x})>0\}\mathbb{1}(f \neq f^*) = \mathbb{1}\{g^*(\boldsymbol{x})>0\}\Delta^2(f;f^*)} \\
\leq & \underbrace{\mathbb{E}_{S_n, \sigma_{1:n}}\Big[\sup_{f \in \mathcal{F}, \mathbb{E}_{\boldsymbol{x}}[\mathbb{1}\{g^*(\boldsymbol{x})>0\}\mathbb{1}(f \neq f^*)] \leq r} \frac{1}{n}\sum_{i=1}^n \sigma_i \mathbb{1}\{g^*(\boldsymbol{x}) > 0\}\mathbb{1}\{f(\boldsymbol{x}_i) \neq f^*(\boldsymbol{x}_i)\}\Big]}_{\substack{|\mathbb{1}\{g^*(\boldsymbol{x}_i)>0\}\Delta(f_1;f^*) - \mathbb{1}\{g^*(\boldsymbol{x}_i)>0\}\Delta(f_2;f^*)| \leq |\mathbb{1}\{g^*(\boldsymbol{x}_i)>0\}\mathbb{1}(f_1 \neq f^*) - \mathbb{1}\{g^*(\boldsymbol{x}_i)>0\}\mathbb{1}(f_2 \neq f^*)| \\ \text{Talagrand Contraction Inequality (Ledoux \& Talagrand, 1991)}}}
\end{aligned}
$$
$$(64)$$

In the last inequality, we use the fact that $\mathbb{1}\{g^*(\boldsymbol{x}_i) > 0\}\Delta(f;f^*)$ is a 1-Lipschitz function of $\mathbb{1}\{g^*(\boldsymbol{x}_i) > 0\}\mathbb{1}\{f^* \neq f\}$. In particular, $\Delta(f;f^*) = \mathbb{1}\{f^* \neq f\}\Delta(f;f^*)$ and

$$
\begin{aligned}
&|\mathbb{1}\{g^*(\boldsymbol{x}_i) > 0\}\Delta(f_1;g^*) - \mathbb{1}\{g^*(\boldsymbol{x}_i) > 0\}\Delta(f_2;g^*)| && (65) \\
\leq & \mathbb{1}\{g^*(\boldsymbol{x}_i) > 0\}|\Delta(f_1;g^*) - \Delta(f_2;g^*)| \\
\leq & \mathbb{1}\{g^*(\boldsymbol{x}_i) > 0\}|\mathbb{1}(f_1 \neq y) - \mathbb{1}(f_2 \neq y)| \\
= & \mathbb{1}\{g^*(\boldsymbol{x}_i) > 0\}|\mathbb{1}(f_1 \neq f_2)| \\
= & |\mathbb{1}\{g^*(\boldsymbol{x}_i) > 0\}\mathbb{1}(f_1 \neq f^*) - \mathbb{1}\{g^*(\boldsymbol{x}_i) > 0\}\mathbb{1}(f_2 \neq f^*)| && (66)
\end{aligned}
$$

And so we can use Talagrand Contraction Inequality and this finishes the proof of inequality 64.

Now define the class $\mathcal{G} \cdot \mathbb{1} \circ \mathcal{F} \equiv \mathbb{1}\{g^*(\boldsymbol{x}) > 0\}\mathbb{1}\{f(\boldsymbol{x}) \neq f^*(\boldsymbol{x}), f \in \mathcal{F}\}$. The indicator function $\mathbb{1}\{g^*(\boldsymbol{x}) > 0\}\mathbb{1}\{f(\boldsymbol{x}) \neq f^*(\boldsymbol{x})\}$ is a Boolean function taking $f$ as input, thus $d_{VC}(\mathcal{G} \cdot \mathbb{1} \circ \mathcal{F}) \leq d_{VC}(\mathcal{F})$ (Vidyasagar, 2013). Thus we have

$$
\begin{aligned}
&\mathbb{E}_{S_n}[\mathcal{R}_n\{\mathbb{1}\{g^*(\boldsymbol{x}) > 0\}\Delta(f;f^*) \in \mathcal{H} : \mathbb{E}[h^2] \leq r\}] \\
\leq & \mathbb{E}_{S_n}[\mathcal{R}_n\{\mathbb{1}\{g^*(\boldsymbol{x}) > 0\}\mathbb{1}\{f(\boldsymbol{x}) \neq f^*(\boldsymbol{x})\} \in \mathbb{1} \circ \mathcal{F} : \mathbb{E}[\mathbb{1}\{g^*(\boldsymbol{x}) > 0\}\mathbb{1}\{f(\boldsymbol{x}) \neq f^*(\boldsymbol{x})\}] \leq r\}]
\end{aligned}
$$
$$(67)$$

Above implies that we can pick $\psi(r)$ to be

$$
\psi(r) = 4\mathbb{E}_{S_n}[\mathcal{R}_n\{*\mathcal{G} \cdot \mathbb{1} \circ \mathcal{F} : \mathbb{E}[\mathbb{1}\{g^*(\boldsymbol{x}) > 0\}\mathbb{1}\{f(\boldsymbol{x}) \neq f^*(\boldsymbol{x})\}] \leq r\}] + \frac{176 \log n}{n} \tag{68}
$$

By Equation (80), we have:

$$
\mathbb{E}_{\boldsymbol{x},y}[\mathbb{1}\{g^*(\boldsymbol{x}) > 0\}\Delta(\widetilde{f};f^*)] \leq \frac{2}{n}\sum_{i=1}^n \mathbb{1}\{g^*(\boldsymbol{x}_i) > 0\}\Delta(\widetilde{f};f^*;\boldsymbol{x}_i,y_i) + 1500r^* + \frac{176\log(1/\delta)}{n} \tag{69}
$$

By inequality (62), we have $\frac{1}{n}\sum_{i=1}^n \Delta(\widetilde{f};f^*;\boldsymbol{x}_i,y_i) \leq 2\varepsilon$ holds with probability $1 - \delta$. By Lemma 7 we have $r^* \lesssim \frac{d_{VC}(\mathcal{F})\log n}{n}$. Plugging in Equation 80 we have that $n \gtrsim \frac{(d_{VC}(\mathcal{F})\log(\frac{d_{VC}(\mathcal{F})}{\varepsilon}) + \log(1/\delta))}{\varepsilon}$ suffices to achieve $\mathbb{E}_{\boldsymbol{x},y}[\mathbb{1}\{g^*(\boldsymbol{x}) > 0\}\Delta(\widetilde{f};f^*,\boldsymbol{x},y)] \lesssim \varepsilon$. Similar to the proof in Theorem 4, we have:

$$
\mathbb{E}_{\boldsymbol{x}}[\{\mathbb{1}\{\widetilde{f}(\boldsymbol{x}) \neq f^*(\boldsymbol{x})\}|\mathbb{1}\{g^*(\boldsymbol{x}) > 0\}] \leq \frac{\varepsilon}{\alpha} \tag{70}
$$

$\square$

## C  Technical Lemmas

**Lemma 2.** *Let $S_n = \{(\boldsymbol{x}_i, y_i)\}$ be i.i.d sample from Data Generative Process described in Definition 1. For every $\varepsilon > 0$, there exist a $\delta > 0$ such that if $n \geq \frac{3 \log(\frac{|\mathcal{F}|}{\delta})}{\varepsilon}$, the following inequality holds simultaneously for all $f \in \mathcal{F}$ with $|\mathcal{F}| < \infty$ with probability at least $1 - \delta$*

$$\frac{1}{n} \sum_{i=1}^{n} \mathbb{1}\{f(\boldsymbol{x}_i) \neq f^*(\boldsymbol{x}_i)\} < \mathbb{E}_{\boldsymbol{x}}[\mathbb{1}\{f(\boldsymbol{x}) \neq f^*(\boldsymbol{x})\}] + \varepsilon \tag{71}$$

*Proof.* By taking union bound one can ensure that

$$\mathbb{P}_{S_n}\left[\sup_{f \in \mathcal{F}}\left\{\left|\sum_{i=1}^{n} \mathbb{1}\{f(\boldsymbol{x}_i) \neq f^*(\boldsymbol{x}_i)\} - n\mathbb{E}_{\boldsymbol{x}}[\mathbb{1}\{f(\boldsymbol{x}) \neq f^*(\boldsymbol{x})\}]\right|\right\} \geq n\varepsilon\right]$$

$$\leq \mathbb{P}_{S_n}\left[\forall f \in \mathcal{F} : \left\{\left|\sum_{i=1}^{n} \mathbb{1}\{f(\boldsymbol{x}_i) \neq f^*(\boldsymbol{x}_i)\} - n\mathbb{E}_{\boldsymbol{x}}[\mathbb{1}\{f(\boldsymbol{x}) \neq f^*(\boldsymbol{x})\}]\right| \geq n\varepsilon\right\}\right]$$

$$\leq \sum_{f \in \mathcal{F}} \mathbb{P}_{S_n}\left[\left|\sum_{i=1}^{n} \mathbb{1}\{f(\boldsymbol{x}_i) \neq f^*(\boldsymbol{x}_i)\} - n\mathbb{E}_{\boldsymbol{x}}[\mathbb{1}\{f(\boldsymbol{x}) \neq f^*(\boldsymbol{x})\}]\right| \geq n\varepsilon\right]$$

We next apply the following version of Chernoff inequality with $a \geq 1$: Let $X = \sum_{i=1}^{n} X_i$ where $X_i \in \{0, 1\}$. Then

$$\mathbb{P}[X \geq (1 + a)\mathbb{E}(X)] \leq \exp\left(-\frac{a^2}{2 + a}\mathbb{E}(X)\right) \leq \exp\left(-\frac{a}{3}\mathbb{E}(X)\right)$$

$$\mathbb{P}[X \leq (1 - a)\mathbb{E}(X)] \leq \exp\left(-\frac{a^2}{2}\mathbb{E}(X)\right) \leq \exp\left(-\frac{a}{3}\mathbb{E}(X)\right)$$

So we have

$$\mathbb{P}[|X - \mathbb{E}(X)| \geq a\mathbb{E}(X)] \leq \exp\left(-\frac{a}{3}\mathbb{E}(X)\right) \tag{72}$$

For any fixed $f \in \mathcal{F}$, let $X_i = \mathbb{1}\{f(\boldsymbol{x}_i) \neq f^*(\boldsymbol{x}_i)\}$, and let $a = \varepsilon/\mathbb{E}_{\boldsymbol{x}}[\mathbb{1}\{f(\boldsymbol{x}) \neq f^*(\boldsymbol{x})\}]$. Then by inequality 72 we have

$$\mathbb{P}_{S_n}\left[\left|\sum_{i=1}^{n} \mathbb{1}\{f(\boldsymbol{x}_i) \neq f^*(\boldsymbol{x}_i)\} - n\mathbb{E}_{\boldsymbol{x}}\mathbb{1}\{f(\boldsymbol{x}) \neq f^*(\boldsymbol{x})\}\right| \geq n\varepsilon\right]$$

$$\leq \exp\left(-\frac{n\mathbb{E}_{\boldsymbol{x}}[\mathbb{1}\{f(\boldsymbol{x}) \neq f^*(\boldsymbol{x})\}]a}{3}\right) = \exp\left(-\frac{n\varepsilon}{3}\right) \tag{73}$$

Using (72) and setting $\delta = |\mathcal{F}| \exp(-n\epsilon/3)$ finishes the proof. $\qquad\square$

**Lemma 3.** *Suppose $S_n = \{(\boldsymbol{x}_1, y_1), ..., (\boldsymbol{x}_n, y_n)\}$ are i.i.d sampled , and let $L(f, \boldsymbol{x}, y) \in \{+1, -1\}$ be a function. Let $L_{S_n}(f) = \frac{1}{n} \sum_{i=1}^{n} L(f, \boldsymbol{x}_i, y_i)$ and $L(f) = \mathbb{E}_{\boldsymbol{x}, y}[L(f, \boldsymbol{x}, y)]$. Given parameter $t$ such that*

$$nt^2 \geq 2b^2$$

*then we have:*

$$\mathbb{P}_{S_n \sim \mathcal{D}}[\sup_{f \in \mathcal{F}} |L_{S_n}(f) - L(f)| \geq t] \leq 4\mathcal{B}_{\mathcal{F}}(2n)e^{-\frac{nt^2}{4b^2}}$$

**Proof**: For two sample sets $S_n$ and $S'_n$, if we have $|L_{S_n}(f) - L(f)| \geq t$ and $|L_{S'_n}(f) - L(f)| \leq \frac{t}{2}$ then we get that $|L_{S_n} - L_{S'_n}| \geq \frac{t}{2}$. Let $\widehat{f}$ be $f$ that attains $\sup_{f \in \mathcal{F}} |L_{S_n}(f) - L(f)|$, one can verify that :

$$\mathbb{1}\{|L_{S_n}(\widehat{f}) - L(\widehat{f})| \geq t\} \cdot \mathbb{1}\{|L_{S'_n}(\widehat{f}) - L(\widehat{f})| \leq \frac{t}{2}\}$$

$$\leq \mathbb{1}\{\sup_{f \in \mathcal{F}} |L_{S_n}(f) - L_{S'_n}(f)| \geq \frac{t}{2}\} \tag{74}$$

Taking expectation w.r.t $S_n \sim \mathcal{D}$ and $S'_n \sim \mathcal{D}$ we have

$$
\mathbb{P}_{S_n \sim \mathcal{D}}\Big[\sup_{f \in \mathcal{F}} |L_{S_n}(f) - L(f)| \geq t\Big] \cdot \mathbb{P}_{S'_n \sim \mathcal{D}}\Big[|L_{S'_n}(\widehat{f}) - L(\widehat{f})| \leq \frac{t}{2}|S_n, \widehat{f} \triangleq \sup_{f \in \mathcal{F}} |L_{S_n}(f) - L(f)|\Big]
$$
$$
\leq \mathbb{P}_{S_n, S'_n \sim \mathcal{D}}\Big[\sup_{f \in \mathcal{F}} |L_{S_n}(f) - L_{S'_n}(f)| \geq \frac{t}{2}\Big] \tag{75}
$$

Next we lower bound $\mathbb{P}\big[|L_{S_n}(f) - L(f)| \leq \frac{t}{2}\big]$ for any fixed $f$. Note that the choice of $\widehat{f}$ is free of $S'_n$ since $S'_n$ and $S_n$ are iid samples. Since $L(f, x, y) \in [-1, 1]$ and so $Var(L(f, x, y)) \leq b^2/4$, using $nt^2 \geq 2b^2$ we have that:

$$
\mathbb{P}_{S_n \sim \mathcal{D}}\big[|L_{S_n}(f) - L(f)| \geq \frac{t}{2}\big] \leq \frac{4 Var(L_{S_n})}{nt^2} \leq \frac{1}{2}
$$

So we have $\mathbb{P}_{S'_n \sim \mathcal{D}}\big[|L_{S'_n}(\widehat{f}) - L(\widehat{f})| \leq \frac{t}{2}|S_n, \widehat{f} \triangleq \sup_{f \in \mathcal{F}} |L_{S_n}(f) - L(f)|\big] \geq \frac{1}{2}$. Let $\mathcal{F}_{S_{2n}} \subseteq \{+1, -1\}^{2n}$ be projection of $\mathcal{F}$ on $S_n \bigcup S'_n$, combining this inequality with (75) we have

$$
\mathbb{P}_{S_n \sim \mathcal{D}}[\sup_{f \in \mathcal{F}} |L_{S_n}(f) - L(f)| \geq t]
$$
$$
\leq 2\mathbb{P}_{S_n, S'_n \sim \mathcal{D}}[\sup_{f \in \mathcal{F}} |L_{S_n}(f) - L_{S'_n}(f)| \geq \frac{t}{2}]
$$
$$
= 2\mathbb{P}_{S_n, S'_n \sim \mathcal{D}}[\sup_{f(\boldsymbol{x}) \in \mathcal{F}_{S_{2n}}} |L_{S_n}(f) - L_{S'_n}(f)| \geq \frac{t}{2}]
$$
$$
\leq 2\mathbb{P}_{S_{2n}}\big[\mathbb{P}_{S_n = S_{2n} - S'_n}[\sup_{f(\boldsymbol{x}) \in \mathcal{F}_{S_{2n}}} |L_{S_n}(f) - L_{S'_n}(f)| \geq \frac{t}{2}|S_{2n}]\big]
$$
$$
\leq 2\mathbb{P}_{S_{2n}}\big[\bigcup_{f(\boldsymbol{x}) \in \mathcal{F}_{S_{2n}}} \mathbb{P}_{S_n = S_{2n} - S'_n}[|L_{S_n}(f) - L_{S'_n}(f)| \geq \frac{t}{2}|S_{2n}]\big] \tag{76}
$$
$$
\leq 2\mathbb{P}_{S_{2n}}\big[2|\mathcal{F}_{S_{2n}}|e^{-\frac{nt^2}{4b^2}}|S_{2n}]\big]
$$
$$
\leq 2\mathbb{P}_{S_{2n}}\big[\sup_{S_{2n}} |\mathcal{F}_{S_{2n}}|e^{-\frac{nt^2}{4b^2}}|S_{2n}]\big]
$$
$$
\leq 2\sup_{S_{2n}} |\mathcal{F}_{S_{2n}}|\mathbb{P}_{S_{2n}}\big[e^{-\frac{nt^2}{4b^2}}\big]\big]
$$
$$
\leq 2\mathcal{B}_{\mathcal{F}}(2n)e^{-\frac{nt^2}{4b^2}}
$$

**Lemma 4** (Hoeffding's Inequality). *Let $Z_1, ..., Z_n$ be independent bounded random variables with $Z_i \in [a, b]$ for all $i$, where $-\infty < a < b < \infty$. Then for all $t > 0$:*

$$
\mathbb{P}(\frac{1}{n}|\sum_{i=1}^n Z_i - \mathbb{E}[Z_i]| \geq t) \leq 2e^{-\frac{2nt^2}{(b-a)^2}} \tag{77}
$$

**Lemma 5.** *Consider a set of samples $S = \{(\boldsymbol{x}_1, y_1), ..., (\boldsymbol{x}_n, y_n)\}$ drawn i.i.d. from the Noisy Generative Process and $f^*$ in the hypothesis class $\mathcal{F}$ satisfying $f(\boldsymbol{x}) \in \{-1, +1\}$. If:*

$$
n \geq \frac{3\log(\frac{1}{\delta})}{\epsilon^2 \alpha^2}
$$

*Then we have with probability at least $1 - \delta$ :*

$$
\frac{1}{n}\sum_{i=1}^n \mathbb{1}\{f^*(\boldsymbol{x}_i) \neq y_i\} \leq \frac{1}{2}(1 - \alpha) + \alpha\varepsilon \tag{78}
$$

**Proof**:

The terms $\mathbb{1}\{f(\boldsymbol{x}_i) \neq y_i\} \in \{0,1\}$ for $i \in [n]$ are a set of $n$ independent random variables since $(\boldsymbol{x}_i, y_i)$ are independent. So we can use Hoeffding's inequality (Lemma 4). By setting $b - a = 1$, $t = \alpha\epsilon$ in Equation 77, the choice of $n$ ensures that $\frac{-2nt^2}{(b-a)^2} \leq 6\log(\delta)$. Thus

$$\mathbb{P}_{S_n \sim \mathcal{D}_\alpha}[|\frac{1}{n}\sum_{i=1}^n \mathbb{1}\{f^*(\boldsymbol{x}_i) \neq y_i\} - \mathbb{E}_{\boldsymbol{x},y}[\mathbb{1}\{f^*(\boldsymbol{x}) \neq y\}]| \geq \epsilon\alpha] \leq \delta.$$

where we have

$$
\begin{aligned}
&\mathbb{E}_{(\boldsymbol{x},y)\sim\mathcal{D}_\alpha}[\mathbb{1}\{f^*(\boldsymbol{x}) \neq y\}] \\
=&\underbrace{\mathbb{E}_{(\boldsymbol{x},y)\sim\mathcal{D}_\alpha}[\mathbb{1}\{f^*(\boldsymbol{x}) \neq y\}|\boldsymbol{x} \in \Omega_U]\mathbb{P}[\boldsymbol{x} \in \Omega_U]}_{\frac{1}{2}\mathbb{P}[\boldsymbol{x}\in\Omega_U]\text{: Since } y \text{ is labeled by coin flipping in } \Omega_U} + \underbrace{\mathbb{E}_{(\boldsymbol{x},y)\sim\mathcal{D}_\alpha}[\mathbb{1}\{f^*(\boldsymbol{x}) \neq y\}|\boldsymbol{x} \in \Omega_I]\mathbb{P}[\boldsymbol{x} \in \Omega_I]}_{0\text{: Since } y \text{ is labeled by } f^* \text{ with 0 Bayes Risk in } \Omega_I} \\
=&\frac{1}{2}(1 - \alpha)
\end{aligned}
\tag{79}
$$

This way we have:

$$\mathbb{P}_{S_n \sim \mathcal{D}_\alpha}[|\frac{1}{n}\sum_{i=1}^n \mathbb{1}\{f^*(\boldsymbol{x}_i) \neq y_i\} - \frac{1}{2}(1 - \alpha)| \geq \epsilon\alpha] \leq \delta.$$

which implies that Equation 78 holds with probability at least $1 - \delta$. $\qquad\square$

**Lemma 6** (Theorem 3.3 in (Bartlett et al., 2005)). *Let $\mathcal{F}$ be a class of functions with range in $[a, b]$ and assume that there are some functional $T : \mathcal{H} \to \mathbb{R}^+$ and some constant $B$ such that for every $h \in \mathcal{H}$, $\boldsymbol{Var}(h) \leq T(h) \leq B\mathbb{E}[h]$. Let $\psi$ be a subroot function and $r^*$ be the fixed point of $\psi$. Assume the $\psi$ satisfies, for any $r \geq r^*$,*

$$\psi(r) \geq B\mathbb{E}_{S_n}\mathcal{R}_n\{h \in \mathcal{H} : T(h) \leq r\}$$

*Then with $c_1 = 704$ and $c_2 = 26$, for any $K > 1$ and every $t > 1$ with probability at least $1 - e^{-t}$,*

$$\forall h \in \mathcal{H}, P[h] \leq \frac{K}{K-1}P_n h + \frac{c_1 K}{B}r^* + \frac{t(11(b-a) + c_2 BK)}{n}\tag{80}$$

*Also with probability at least $1 - e^{-t}$,*

$$\forall h \in \mathcal{H}, P_n[h] \leq \frac{K+1}{K}Ph + \frac{c_1 K}{B}r^* + \frac{t(11(b-a) + c_2 BK)}{n}\tag{81}$$

*where $Pf = \mathbb{E}_{\boldsymbol{x}}[h(\boldsymbol{x})]$ and $P_n = \frac{1}{n}\sum_{i=1}^n h(\boldsymbol{x}_i)$.*

**Lemma 7.** *Given hypothesis class $\mathcal{F} : \mathcal{X} \to [-b, b]$ with some universal constant $b$ and its VC-dimension $d_{VC}(\mathcal{F}) < \infty$. Define following sub-root function with $B \geq 1$:*

$$\psi(r) = 100B\mathbb{E}_{S_n}\mathcal{R}_n\{*\mathcal{F}, r\} + \frac{11b^2 \log n}{n}.$$

*Let $r^*$ be fixed point of $\psi(r)$ so that $r^* = \psi(r^*)$, suppose $n \geq d_{VC}(\mathcal{F})$, we have*

$$r^* \lesssim \frac{B^2 d_{VC}(\mathcal{F})\log(\frac{n}{d_{VC}(\mathcal{F})})}{n}$$

*Proof.* The proof largely follows from the proof in Corollary 3.7 in (Bartlett et al., 2005). We include it here for completeness. Since $f$ is uniformly bounded by $b$, for any $r \geq \psi(r)$, Corollary 2.2 in (Bartlett et al., 2005) implies that with probability at least $1 - \frac{1}{n}$, $\{f \in *\mathcal{F} : Pf^2 \leq r\} \subseteq \{f \in *\mathcal{F} : P_n f^2 \leq 2r\}$. Let $\mathcal{E}$ be event that $\{f \in *\mathcal{F} : Pf^2 \leq r\} \subseteq \{f \in *\mathcal{F} : P_n f^2 \leq 2r\}$ holds, above implies

$$
\begin{aligned}
&\mathbb{E}_{S_n}\mathcal{R}_n\{*\mathcal{F}, Pf^2 \leq r\} \\
\leq&\mathbb{P}[\mathcal{E}]\mathbb{E}_{S_n}[\mathcal{R}_n\{*\mathcal{F}, Pf^2 \leq r\}|\mathcal{E}] + \mathbb{P}[\mathcal{E}^c]\mathbb{E}_{S_n}[\mathcal{R}_n\{*\mathcal{F}, Pf^2 \leq r\}|\mathcal{E}^c] \\
\leq&\mathbb{E}_{S_n}[\mathcal{R}_n\{*\mathcal{F}, P_n f^2 \leq 2r\}] + \frac{b}{n}
\end{aligned}
\tag{82}
$$

Since $r^* = \psi(r^*)$, $r^*$ satisfies

$$r^* \leq 100B\mathbb{E}_{S_n}\mathcal{R}_n\{*\mathcal{F}, P_n f^2 \leq 2r^*\} + \frac{b + 11b^2 \log n}{n}. \tag{83}$$

Next we leverage Dudley's chaining bound (Dudley, 2014) to upper bound $\mathbb{E}\mathcal{R}_n\{*\mathcal{F}, P_n f^2 \leq 2r^*\}$ using integral of covering number. We first bound the covering number of a star hull of $\mathcal{F}$. It follows from (Bartlett et al., 2005) Corollary 3.7 that

$$\log \mathcal{N}_2(\varepsilon, \mathcal{F}, \boldsymbol{x}_{1:n}) \leq \log \left\{ \mathcal{N}_2\left(\frac{\varepsilon}{2}, \mathcal{F}, \boldsymbol{x}_{1:n}\right)\left(\lceil \frac{2}{\varepsilon} \rceil + 1\right)\right\}$$

And covering number $\log \mathcal{N}_2(\varepsilon, \mathcal{F}, n)$ can be bounded using VC-dimension of $\mathcal{F}$ using Haussler's bound on the covering number (Haussler, 1995; Wellner et al., 2013):

$$\log \mathcal{N}_2\left(\frac{\varepsilon}{2}, \mathcal{F}, n\right) \leq c_1 d_{VC} \log\left(\frac{1}{\varepsilon}\right)$$

where $c_1$ is some universal constant. Now we are ready to apply the chaining bound, it follows from Theorem B.7 (Bartlett et al., 2005) that

$$\begin{aligned}
&\mathbb{E}_{S_n}[\mathcal{R}_n(*\mathcal{F}, P_n f^2 \leq 2r^*)]\\
\leq& \frac{c_2}{\sqrt{n}}\mathbb{E}_{S_n}\int_0^{\sqrt{2r^*}}\sqrt{\log \mathcal{N}_2(\varepsilon, *\mathcal{F}, \boldsymbol{x}_{1:n})}d\varepsilon\\
\leq& \frac{c_2}{\sqrt{n}}\mathbb{E}_{S_n}\int_0^{\sqrt{2r^*}}\sqrt{\log \mathcal{N}_2\left(\frac{\varepsilon}{2}, \mathcal{F}, \boldsymbol{x}_{1:n}\right)\left(\lceil \frac{2}{\varepsilon}\rceil + 1\right)}d\varepsilon\\
\leq& c_3\sqrt{\frac{d_{VC}(\mathcal{F})r^*\log(1/r^*)}{n}}\\
\leq& c_3\sqrt{\frac{d_{VC}^2(\mathcal{F})}{n^2} + \frac{d_{VC}(\mathcal{F})r^*\log(n/ed_{VC}(\mathcal{F}))}{n}}
\end{aligned} \tag{84}$$

Where $c_2$ and $c_3$ are some universal constants. Together with Equation 83 one can solve for $r^* \lesssim \frac{B^2 d_{VC}(\mathcal{F})\log(\frac{n}{d_{VC}(\mathcal{F})})}{n}$ □

**Lemma 8.** *Let $S_n = \{(\boldsymbol{x}_i, y_i)\}$ be i.i.d sample from Data Generative Process described in Definition 1. For every $\varepsilon > 0$, there exist a $\delta > 0$ such that if $n \gtrsim \frac{d_{VC}(\mathcal{F})\log(\frac{1}{\varepsilon}) + \log(\frac{1}{\delta})}{\varepsilon}$, following inequality holds simultaneously for all $f \in \mathcal{F}$ with $d_{VC}(\mathcal{F}) < \infty$, with probability at least $1 - \delta$*

$$\frac{1}{n}\sum_{i=1}^n \mathbb{1}\{f(\boldsymbol{x}_i) \neq f^*(\boldsymbol{x}_i)\} \lesssim \mathbb{E}_{\boldsymbol{x}}\mathbb{1}\{f(\boldsymbol{x}) \neq f^*(\boldsymbol{x})\} + \varepsilon \tag{85}$$

*Proof.* The proof invokes Lemma 6, in particular, the Equation 81. Let $\mathbb{1} \circ \mathcal{F} : \mathbb{1}\{f(\boldsymbol{x}) \neq f^*(\boldsymbol{x}), f \in \mathcal{F}\}$ be the hypothesis class $\mathcal{H}$ in Lemma 6. Since $f^*$ is a deterministic boolean function, it does not increase the number of points that can be shattered by $\mathcal{F}$. We have $d_{VC}(\mathbb{1} \circ \mathcal{F}) \leq d_{VC}(\mathcal{F})$. In particular, we choose the functional $T(\cdot) = \mathbb{E}[\cdot]$ and it is easy to verify that

$$\boldsymbol{Var}(\mathbb{1}\{f(\boldsymbol{x}) \neq f^*(\boldsymbol{x})\}) \leq \mathbb{E}_{\boldsymbol{x}}[\mathbb{1}\{f(\boldsymbol{x}) \neq f^*(\boldsymbol{x})\}] = \mathbb{E}_{\boldsymbol{x}}[\mathbb{1}^2\{f(\boldsymbol{x}) \neq f^*(\boldsymbol{x})\}].$$

Let $\psi(r) = 100\mathbb{E}\mathcal{R}_n\{*\mathcal{F}, \mathbb{E}f \leq r\} + \frac{11 \log n}{n}$. We have

$$\mathbb{E}\mathcal{R}_n\{\mathcal{F}, \mathbb{E}f^2 \leq r\} \leq \mathbb{E}\mathcal{R}_n\{*\mathcal{F}, \mathbb{E}f^2 \leq r\} \leq 100\mathbb{E}\mathcal{R}_n\{*\mathcal{F}, \mathbb{E}f^2 \leq r\} + \frac{11 \log n}{n} = \psi(r)$$

Since local Rademacher averages of the star-hull is sub-root function, we know for all $r \geq r^*$, $\psi(r) \geq \psi(r^*) = r^*$. By Equation 81 in Lemma 6 we have

$$\frac{1}{n} \sum_{i=1}^{n} \mathbb{1}\{f(\boldsymbol{x}_i) \neq f^*(\boldsymbol{x}_i)\} \leq 2\mathbb{E}_{\boldsymbol{x}} \mathbb{1}\{f(\boldsymbol{x}) \neq f^*(\boldsymbol{x})\} + 15r^* + \frac{\log(1/\delta) + 5200}{n}\varepsilon \tag{86}$$

Next we bound $r^*$. A direct application of Lemma 7 show that

$$r^* \lesssim \frac{d_{VC}(\mathbb{1} \circ \mathcal{F}) \log(\frac{n}{d_{VC}(\mathbb{1}\circ\mathcal{F})})}{n} \lesssim \frac{d_{VC}(\mathcal{F}) \log(\frac{n}{d_{VC}(\mathcal{F})})}{n}.$$

The rest of the proof follows from plugging $r^*$ in Equation 81 and removing absolute constants. $\qquad\square$

## D  More Experimental Results and Details

### D.1  Experiment Setting and Implementation Details

**Extension to multi-class.** Our method extends to the multi-class setting naturally. In the case of $K$-class classification, our selector loss remains the same while the predictor becomes $f(\boldsymbol{x}) = f(\boldsymbol{x})_{1:K} : \mathcal{X} \to \Delta^K$ where $\Delta^K$ is the $K$-simplex. Meanwhile, we use multi-class cross entropy loss to train the classifier. The pseudo-informative label becomes $\widehat{z}_i = \mathbb{1}\{\arg\max_{k \in [K]} f(\boldsymbol{x}_i)_k = y_i\}$.

**Hyper-parameters and Neural Network Architectures** We list the hyper-parameters and neural network architectures in Table 4,5 and 6.

Table 4: Hyper-parameters used in real-world experiments. Notation follows original paper.

| Baseline | Hyper-params |
|---|---|
| SLNet | $\alpha = 0.5$, $\lambda = 32$ |
| DeepGambler | $o = 2$ in Volatility and $o = 1.5$ in LC and BUS, Pretrain Epoch=10 |
| Adaptive | $\alpha = 0.9$, Pretrain Epoch=10 |
| Oneside | $\mu = 0.5$, Pretrain Epoch=10 |
| ISA | $\beta = 30$, $\Delta T = 2$, Pretrain Epoch=10 |

Table 5: CNN Architecture.

| Layer Name | Filter Size | Output Size |
|---|---|---|
| 2d Convolution | 3×3 | 32×28×28 |
| ReLU | - | 1×28×28 |
| 2d MaxPool | 2×2 | 32×14×14 |
| 2d Convolution | 3×3 | 64×14×14 |
| ReLU | - | 64×14×14 |
| 2d MaxPool | 2×2 | 64×7×7 |
| Linear | - | 600 |
| Drop-out | - | - |
| Linear | - | 120 |
| Linear | - | 10 |

### D.2  More Synthetic Experiments Results.

We present the exact numbers corresponding to Figure 2-5:

We also present experiments results that only use 25% of the dataset below to show the sample efficiency of our algorithm. We can see all conclusions still hold when we sharply reduce the sample size.

Table 6: Hyper-parameters used in semi-synthetic experiments. Notation follows original paper.

| Dataset | Baseline | Hyper-params |
|---|---|---|
| MNIST+Fashion | SLNet | $\alpha_{slnet} = 0.5$, $\lambda_{slnet} = 32$ |
| | DeepGambler | $o = 1.5$, Pretrain Epoch=5 |
| | Adaptive | $\alpha_{adaptive} = 0.9$, Pretrain Epoch=5 |
| | Oneside | $\mu = 0.5$, Pretrain Epoch=5 |
| | ISA | $\beta = 3$, $\Delta T = 10$, Pretrain Epoch=5 |
| SVHN | SLNet | $\alpha_{slnet} = 0.5$, $\lambda_{slnet} = 32$ |
| | DeepGambler | $o = 2.6$, Pretrain Epoch=10 |
| | Adaptive | $\alpha_{adaptive} = 0.9$, Pretrain Epoch=10 |
| | Oneside | $\mu = 0.5$, Pretrain Epoch=10 |
| | ISA | $\beta = 10$, $\Delta T = 1$, Pretrain Epoch=10 |

Table 7: $g^*$ recovering: AP ↑ v.s $\alpha$ with 100% Sample Size

| Dataset | $\alpha$ | Confidence | SLNet | DeepGambler | Adaptive | Oneside | ISA | ISA-V2 |
|---|---|---|---|---|---|---|---|---|
| MNIST + Fashion | 0.20 | **1.000 ± 0.000** | 0.982 ± 0.011 | **1.000 ± 0.000** | 0.807 ± 0.027 | **1.000 ± 0.000** | **1.000 ± 0.000** | **1.000 ± 0.000** |
| | 0.33 | **1.000 ± 0.000** | 0.998 ± 0.002 | **1.000 ± 0.000** | 0.825 ± 0.027 | **1.000 ± 0.000** | **1.000 ± 0.000** | **1.000 ± 0.000** |
| | 0.43 | **1.000 ± 0.000** | 0.943 ± 0.002 | **1.000 ± 0.000** | 0.855 ± 0.003 | **1.000 ± 0.000** | **1.000 ± 0.000** | **1.000 ± 0.000** |
| | 0.50 | **1.000 ± 0.000** | 0.998 ± 0.003 | **1.000 ± 0.000** | 0.829 ± 0.011 | **1.000 ± 0.000** | **1.000 ± 0.000** | **1.000 ± 0.000** |
| SVHN | 0.20 | 0.984 ± 0.001 | 0.621 ± 0.256 | 0.831 ± 0.270 | 0.984 ± 0.001 | 0.957 ± 0.007 | **0.989 ± 0.001** | 0.982 ± 0.001 |
| | 0.33 | 0.989 ± 0.001 | 0.868 ± 0.039 | 0.990 ± 0.001 | 0.990 ± 0.001 | 0.978 ± 0.002 | **0.992 ± 0.000** | 0.987 ± 0.001 |
| | 0.43 | 0.990 ± 0.001 | 0.925 ± 0.002 | 0.990 ± 0.000 | 0.990 ± 0.000 | 0.982 ± 0.001 | **0.993 ± 0.001** | 0.990 ± 0.001 |
| | 0.50 | 0.991 ± 0.000 | 0.914 ± 0.053 | 0.991 ± 0.000 | 0.989 ± 0.001 | 0.983 ± 0.002 | **0.993 ± 0.001** | 0.991 ± 0.001 |

Table 8: $g^*$ Recovering: AP ↑ v.s. $\alpha$ with 25% Sample Size

| Dataset | $\alpha$ | Confidence | SLNet | DeepGambler | Adaptive | Oneside | ISA | ISA-V2 |
|---|---|---|---|---|---|---|---|---|
| MNIST+Fashion | 0.20 | **1.000 ± 0.000** | 0.412 ± 0.052 | 0.985 ± 0.019 | 0.413 ± 0.024 | 0.969 ± 0.002 | **1.000 ± 0.000** | **1.000 ± 0.000** |
| | 0.33 | **1.000 ± 0.000** | 0.704 ± 0.208 | 0.990 ± 0.001 | 0.362 ± 0.011 | 0.977 ± 0.004 | **1.000 ± 0.000** | **1.000 ± 0.000** |
| | 0.43 | **1.000 ± 0.000** | 0.881 ± 0.179 | 0.987 ± 0.010 | 0.333 ± 0.006 | 0.976 ± 0.007 | **1.000 ± 0.000** | **1.000 ± 0.000** |
| | 0.50 | **1.000 ± 0.000** | 0.876 ± 0.186 | 0.991 ± 0.009 | 0.336 ± 0.006 | 0.980 ± 0.004 | **1.000 ± 0.000** | **1.000 ± 0.000** |
| SVHN | 0.20 | 0.952 ± 0.005 | 0.711 ± 0.121 | 0.540 ± 0.024 | 0.909 ± 0.013 | 0.607 ± 0.125 | **0.972 ± 0.003** | 0.856 ± 0.115 |
| | 0.33 | 0.967 ± 0.001 | 0.914 ± 0.033 | 0.979 ± 0.002 | 0.950 ± 0.009 | 0.926 ± 0.019 | **0.981 ± 0.001** | 0.970 ± 0.003 |
| | 0.43 | 0.973 ± 0.001 | 0.931 ± 0.021 | 0.983 ± 0.002 | 0.967 ± 0.003 | 0.962 ± 0.001 | **0.984 ± 0.001** | 0.977 ± 0.001 |
| | 0.50 | 0.978 ± 0.002 | 0.947 ± 0.006 | **0.986 ± 0.000** | 0.966 ± 0.005 | 0.965 ± 0.004 | 0.985 ± 0.001 | 0.981 ± 0.001 |

Table 9: $g^*$ recovering: AP ↑ v.s $\lambda$ with 100% Sample Size

| Dataset | $\lambda$ | Confidence | SLNet | DeepGambler | Adaptive | Oneside | ISA | ISA-V2 |
|---|---|---|---|---|---|---|---|---|
| MNIST + Fashion | 0.1 | 0.915 ± 0.010 | 0.599 ± 0.168 | 0.985 ± 0.010 | 0.550 ± 0.033 | 0.862 ± 0.033 | **1.000 ± 0.000** | 0.999 ± 0.001 |
| | 0.2 | 0.986 ± 0.001 | 0.712 ± 0.140 | 0.992 ± 0.013 | 0.566 ± 0.018 | 0.973 ± 0.004 | **1.000 ± 0.000** | **1.000 ± 0.000** |
| | 0.3 | **1.000 ± 0.000** | 0.886 ± 0.148 | **1.000 ± 0.000** | 0.571 ± 0.008 | 0.998 ± 0.000 | **1.000 ± 0.000** | **1.000 ± 0.000** |
| SVHN | 0.1 | **0.955 ± 0.004** | 0.641 ± 0.164 | 0.736 ± 0.045 | 0.952 ± 0.002 | 0.944 ± 0.001 | 0.931 ± 0.021 | **0.956 ± 0.003** |
| | 0.2 | **0.978 ± 0.002** | 0.874 ± 0.029 | 0.974 ± 0.003 | 0.953 ± 0.001 | 0.972 ± 0.001 | 0.977 ± 0.003 | **0.978 ± 0.002** |
| | 0.3 | **0.989 ± 0.002** | 0.949 ± 0.013 | 0.981 ± 0.002 | 0.970 ± 0.008 | 0.981 ± 0.002 | 0.986 ± 0.002 | 0.987 ± 0.000 |

Table 10: $g^*$ recovering: AP ↑ v.s $\lambda$ with 25% Sample Size

| Dataset | $\lambda$ | Confidence | SLNet | DeepGambler | Adaptive | Oneside | ISA | ISA-V2 |
|---|---|---|---|---|---|---|---|---|
| MNIST+Fashion | 0.1 | 0.866 ± 0.008 | 0.507 ± 0.011 | 0.799 ± 0.155 | 0.935 ± 0.027 | 0.584 ± 0.017 | **1.000 ± 0.000** | 0.968 ± 0.038 |
| | 0.2 | 0.974 ± 0.005 | 0.625 ± 0.100 | 0.917 ± 0.015 | 0.795 ± 0.013 | 0.718 ± 0.031 | **1.000 ± 0.000** | **1.000 ± 0.000** |
| | 0.3 | 0.999 ± 0.000 | 0.686 ± 0.105 | 0.972 ± 0.010 | 0.521 ± 0.055 | 0.870 ± 0.004 | **1.000 ± 0.000** | **1.000 ± 0.000** |
| SVHN | 0.1 | **0.883 ± 0.009** | 0.784 ± 0.086 | 0.567 ± 0.036 | 0.795 ± 0.015 | 0.820 ± 0.014 | 0.848 ± 0.084 | **0.892 ± 0.008** |
| | 0.2 | 0.938 ± 0.013 | 0.916 ± 0.039 | 0.811 ± 0.108 | 0.857 ± 0.012 | 0.913 ± 0.011 | 0.944 ± 0.006 | **0.959 ± 0.006** |
| | 0.3 | 0.965 ± 0.002 | 0.938 ± 0.015 | 0.938 ± 0.013 | 0.901 ± 0.029 | 0.952 ± 0.006 | 0.967 ± 0.000 | **0.969 ± 0.004** |

## D.3 Real-World Dataset Description

Oxford realized volatility (Volatility) (Heber et al., 2009) data set contains 5-min realized volatility of 31 stock indices from 2000 to 2022 and 155107 records in total. We use past volatility and returns as features, and the task here is to predict whether the next day volatility will be higher than current one, making it a

Figure 5: Average Precision (AP) ↑ v.s Different λ - 25% Samples Size. Numerical results in Table 10.

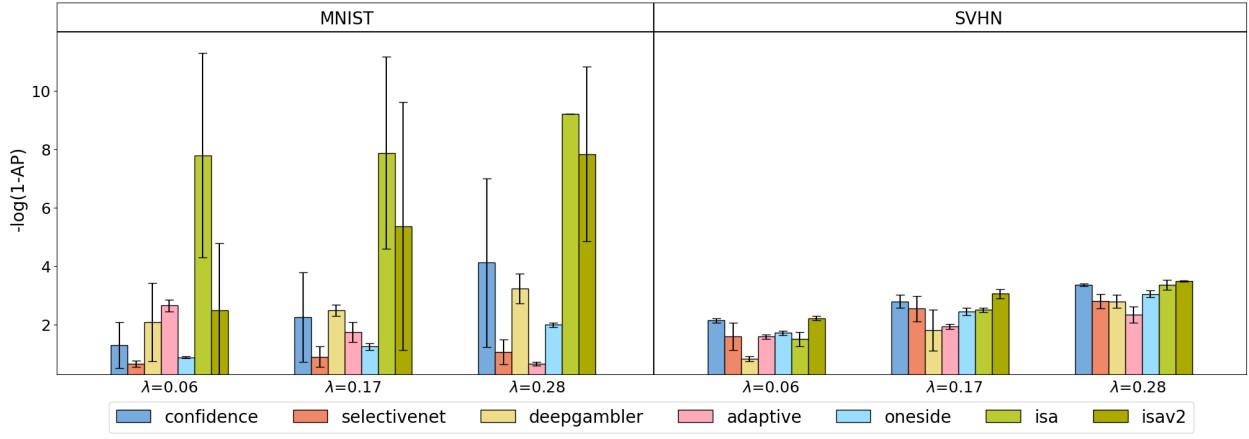

binary classification task. We choose data from 2000 to 2020 as our training set and the rest for the testing (2020 Jan. to 2021 Oct.). This data set is used as an example to show our algorithm's possible application in selectively forecasting financial time series.

Breast ultrasound images (BUS) (Al-Dhabyani et al., 2020). BUS contains 780 gray-scale breast ultrasound images among women in ages between 25 and 75 years old. These images have average size $500 \times 500$ pixels and can be categorized into 3 classes (487 benign, 210 malign and 133 healthy). We randomly choose 80% of data as training dataset and the rest 20% for testing. We are going to use this dataset as an example to show a possible application of our algorithm in automatic diagnosis. The machine can generate diagnosis result only on selected cases and deliver unsure cases to human expert for further investigation.

Lending club(Lending Club, 2007) is a peer-to-peer lending company that matches borrowers with investors through an online platform. The lending club dataset (LC) contains loan data of its customers from 2007 to 2018. We compare different version of existing dataset of LC and remove all inconsistent and incomplete records. There are different status of loans record in this dataset, we keep 3 types of these record that consist the major part of the dataset (261442 charged off cases, 1035418 fully paid cases and 25757 late cases). We use 20% of the dataset as the testing set. This example shows our algorithm can be use to grant loan given on different risk preference.

Table 12 presents the original accuracy given by neural network on each of these 3 real-world data set. For all dataset, the neural network without using selection mechanism cannot give reliable inference. In mortgage granting, high risk like this can cause significant loss. In medical diagnosis which is healthy issue critical, a diagnosis with miss-diagnose rate as high as 15% is not acceptable. However, if we apply our selective algorithm, we can see that the risk on all dataset sharply reduced. In BUS dataset, we can even almost perfectly guarantee the diagnosis result empirically for our most confident cases. These evidence are of practical interest.

Table 11: Real-world Dataset Description

| Dataset | Category | Input Feature | Num. Class | Train | Test |
|---|---|---|---|---|---|
| Volatility | Time Series | 2 | 2 | 143784 | 9525 |
| BUS | Image | $3 \times 324 \times 324$ | 3 | 624 | 156 |
| LC | Tabular | 1805 | 3 | 1058093 | 264524 |

## D.4 Ablation Study Results

## E Illustrative Example for Algorithm 1

Table 12: DNN Original Risk on Each Dataset

|      | Volatility        | BUS               | LC                |
| ---- | ----------------- | ----------------- | ----------------- |
| Risk | $0.340\pm0.002$   | $0.152\pm0.008$   | $0.392\pm0.001$   |

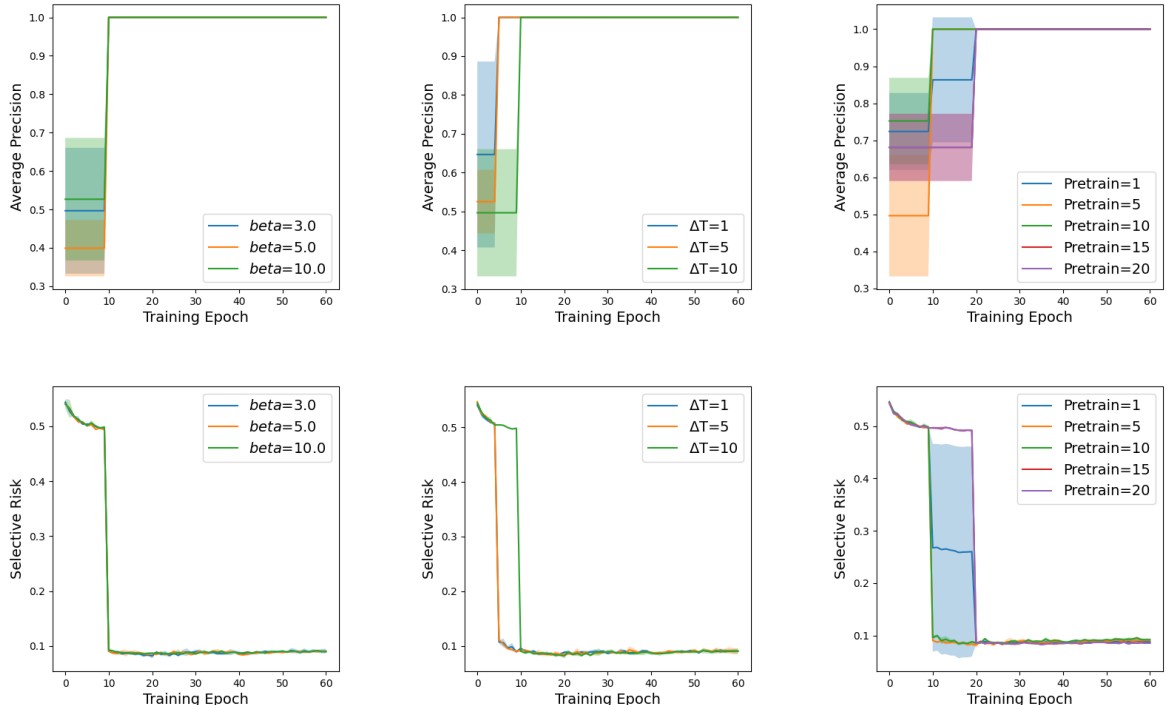

Figure 6: Ablation Study. First column shows the result of different choices of $\beta$. Second column shows different choices of $\Delta T$. Third column shows different choices of pre-training epochs which gives different initial model $\hat{f}^{(0)}$
. Upper panel presents results of average precision and bottom panel presents results of selective risk.

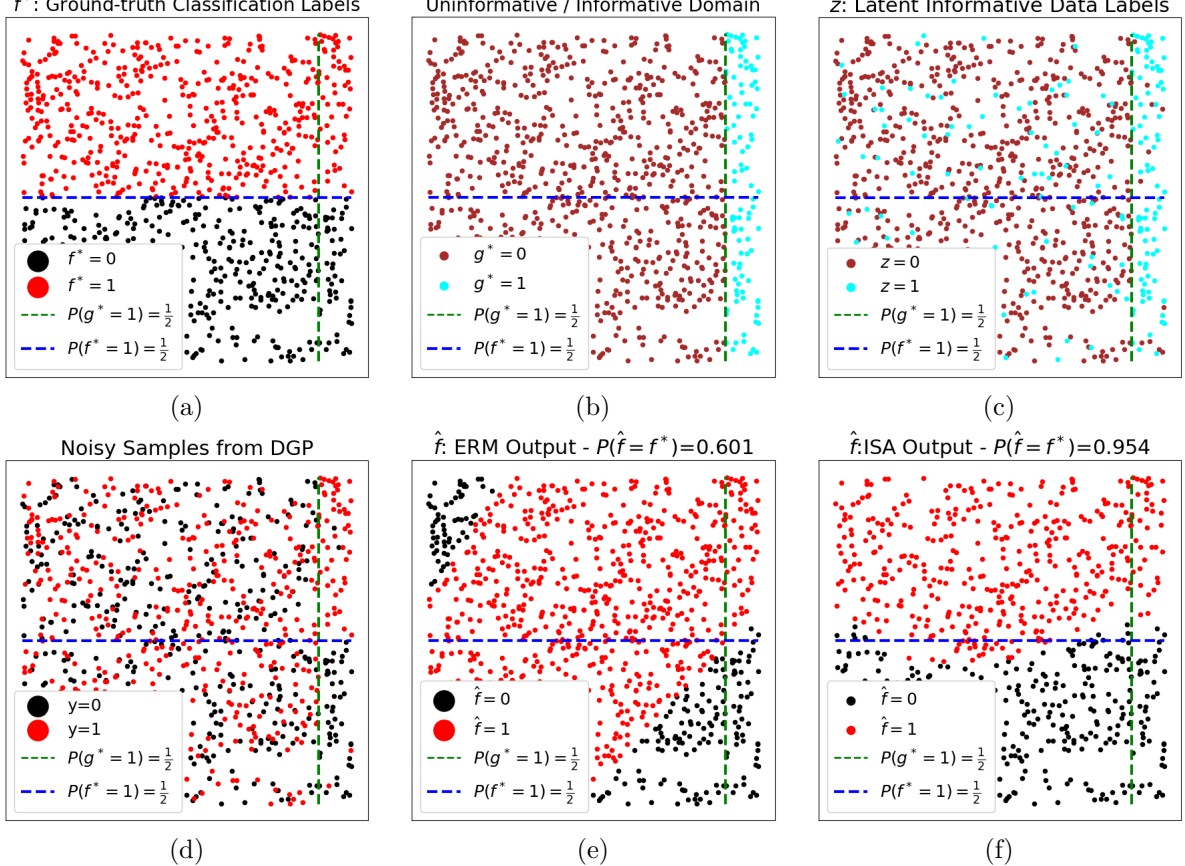

Figure 7: Illustration of Algorithm 1 when $\bar{\lambda} = 0.4$ and $\alpha = 0.1$. The middle blue horizontal line is the ground truth classifier $f^*$. The vertical green line is the boundary separates $\Omega_I$ and $\Omega_U$. The data generation process follows Definition 1. We use SVM with degree-2 polynomial kernel for both $\hat{f}$ and $\hat{g}$. a) shows the ground truth classification labels. b) shows the region of $D_I$ and $D_U$. c) shows the latent informative label $z$. d) shows the realized samples from the noisy data generation process. e) shows the classification result given by ERM. f) shows the classification result given by ISA.

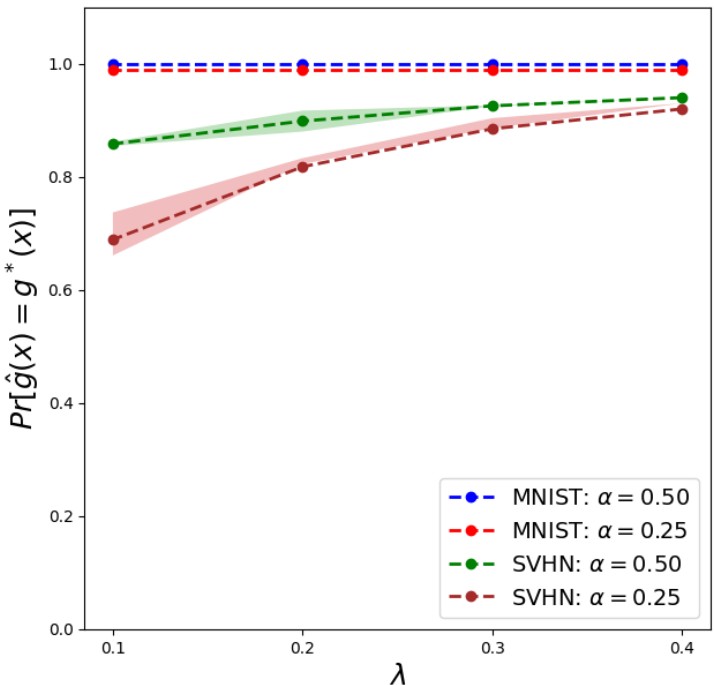

Figure 8: Accuracy on recovering $g^*(x)$.

