# OpenReview forum: "Learning to Abstain From Uninformative Data"
_TMLR — Accepted by TMLR_

### Review · Reviewer_UPfk · 2023-09-25

**Summary Of Contributions:**

The paper study a high noise-to-signal setting where the data is partitioned into two groups: informative data and the uninformative data. They propose a new selective training loss for this setting using two models: selector and predictor. They provide minimax and joint risk bounds for their framework. An iterative algorithm is proposed that solves the problem practically. Experimental results compare their algorithms to the benchmarks on synthetic and real-world data.

**Audience:**

Yes

**Broader Impact Concerns:**

Not needed.

**Claims And Evidence:**

Yes

**Requested Changes:**

1. Please add the corrections required to answer / clarify the "weaknesses" items above.

2. Few writing mistakes could be fixed: the missing space in line 3 of Section 7.1, wrong style for the citation in the first line of page 9, the missing space in the "baselines" part of Section 7, between "OnesideGangrade et al. ...".

**Strengths And Weaknesses:**

Strengths:
1. The paper has a comprehensive presentation of the problem at hand; it starts off with a reasonable motivation intro, then formulates it and proves theoretical guarantees for their framework; finally using several empirical analyses, they complement their theoretical assessments.
2. The paper proposes a practical algorithm for their minimax framework; this is usually lacking in minimax optimal guarantees.

Weakness:
1. The presentation could improve:
1.1. The psudeo-informative loss is defined as $\mathbb{1}\{f(x)\neq y\}$ which lies in $\{0,1\}$, as a surrogate for $z$ which is $\in\{-1, 1\}$. That causes confusion. Also, later on similar expressions are defined with $=$ instead of  $\neq$, like line 4 of Algorithm 1.

1.2. $\beta$ is not clearly defined at Eq. (5) and is later discussed.

2. The claims are not well-supported: the main proposed and analyzed algorithm, Algorithm 1, is missing from the experiments in Sections 7.1-7.3 and is represented with its relaxations, ISA and ISA-V2. This while Section 7.4 shows Algorithm 1 performs drastically worse than its relaxations.

---

> ### Author Response · Authors · 2023-11-15
> **Reply to Reviewer UPfk**
>
> We genuinely value all the feedback provided by the reviewer. We have add more description and discussion accordingly and give a summarized reply below:
>
> ---
>
> __Q1__: On the notation inconsistency.
>
> __A1__: We have unified the notation: now $z\in\{0,1\}$ and $g^*(\boldsymbol{x}) \in \{0,1\}$ which is consistent with pseudo-informative label.
>
> ___
>
> __Q2__: Add description of $\beta$ before Equation (5).
>
> __A2__:  We have added description of $\beta$ for Equation (5)(page 5, section 4.1).
>
> ___
>
> __Q3__:The claims are not well-supported: the main proposed and analyzed algorithm, Algorithm 1, is missing from the experiments in Sections 7.1-7.3 and is represented with its relaxations, ISA and ISA-V2. This while Section 7.4 shows Algorithm 1 performs drastically worse than its relaxations.:
>
> __A3__: We thank the reviewer for pointing out such gap. We believe the performance drop in Section 7.4 is mainly due to the relaxation of using SGD to replace the ERM oracle from the Theorem. We acknowledge that the practical algorithms, including Algo1, ISA, and ISA-V2 have incorporated design choices that are not covered by Theorem 2.
>
> But this does not necessarily contradict with the claims of the paper. We use the theory to justify the proposed practical algorithms, as it establishes the foundation of these practical solutions. Like most existing works, to establish a theoretical result, we inevitably must simplify the real world setting with assumptions. In practical algorithms, we introduce relaxations to adapt the theory to the complex real-world data. For example, the moving average mechanism is introduced to mitigate the variation of Algo 1 using SGD approximation; the Focal loss is introduced to solve the class-imbalance between informative and uninformative data. These additions help stabilize the algorithm and make it robust against variance from stochastic approximation. In the future, we plan to further investigating the theory supporting these design choices using tools from [III, IV].
>
> [III] Awasthi, Pranjal, et al. "H-consistency bounds for surrogate loss minimizers." International Conference on Machine Learning. PMLR, 2022.
>
> [IV] Chen, Tianyi, Yuejiao Sun, and Wotao Yin. "Closing the gap: Tighter analysis of alternating stochastic gradient methods for bilevel problems." Advances in Neural Information Processing Systems 34 (2021)

---

### Review · Reviewer_VdEU · 2023-10-20

**Summary Of Contributions:**

This paper studies the problem of classification where a fraction of the data is uninformative, i.e. likely to have randomly assigned labels. The authors prove theoretical bounds on the performance of a data selection model $g$ that depends on the quality of the predictor $f$. This leads to a natural alternating minimization approach to jointly optimize $f$ and $g$. Empirical results show that a relaxed version of the proposed algorithm outperforms baselines on MNIST + Fashion and SVHN datasets.

**Audience:**

Yes

**Broader Impact Concerns:**

No concerns about broader impact

**Claims And Evidence:**

Yes

**Requested Changes:**

### Critical
- Compare results to those stated in [1-2] if applicable, i.e. expand on statements such as Remarks 4-5 and move them to the main paper
- Add informal statement about the range of $\beta$ to Section 4
- Motivate the use of average precision as a performance metric rather than recall/F1-score/AUC. Is $\beta$ tuned to maximize precision?


### Non-critical
- Show how performance varies with different quality of the initial $\hat{f}^0$
- Describe the main technical challenges to proving convergence guarantees for ISA and ISA-V2 (applying Theorem 2 to Algorithm 1 and relaxations 1-4)
- Compare to semi-supervised learning methods, i.e. discard only the labels of uninformative data

[1] Cortes et al. Learning with rejection. In ICALT, pp. 67–82, 2016.

[2] Yonatan Geifman and Ran El-Yaniv. Selectivenet: A deep neural network with an integrated reject option. In ICML, pp. 2151–2159, 2019.

**Strengths And Weaknesses:**

### Strengths
- Relevant problem area with good motivation
- Mostly well written and organized
- Through ablation studies

### Weaknesses
Related Work
- Brief comparison to [1-2] which should be discussed in more detail. The paper describes differences in motivation, but are any bounds directly comparable to Theorems 1-2?

Experiments
- The experiments would be improved if the choice of average precision as a main metric was better motivated

Clarity
- The term "heuristic" often implies a simple rule that is adopted without proof (e.g. Baseline 5 on page 8). Since ISA and ISA-V2 originate from the guarantees of Theorem 2, in this case the term "practical algorithm" is more appropriate for these algorithms.
- "subset-risk minimization" is mentioned without definition
- Notation alternates between $\Omega_D$ and $\Omega$
- Learning rate $\eta$ is mentioned in Algorithm 1 but not used
- "remains an opened problem" should be "remains an open problem"
- Some important details are only mentioned in the Appendix (see below)

[1] Cortes et al. Learning with rejection. In ICALT, pp. 67–82, 2016.

[2] Yonatan Geifman and Ran El-Yaniv. Selectivenet: A deep neural network with an integrated reject option. In ICML, pp. 2151–2159, 2019.

---

> ### Author Response · Authors · 2023-11-15
> **Reply to reviewer VdEU**
>
> We sincerely appreciate the reviewer for recognizing our paper as well-written and commending its good motivation. We have fixed typos and modified our paper accordingly. We also thank for the revision suggestions given by the reviewer. We reply to each of these concerns below:
>
> __Q1__: Compare results to those stated in [1-2] if applicable, i.e. expand on statements such as Remarks 4-5 and move them to the main paper
>
> __A1__: We added more discussion accordingly in the revised paper (Remark3, Page-7)
>
> ---
>
> __Q2__: Add informal statement about the range of $\beta$ to section 4
>
> __A2__: We added more discussion over the range of $\beta$ to ( section 4, page-5 )
>
> ---
>
> __Q3__: Motivate the use of average precision as a performance metric rather than recall/F1-score/AUC.
>
>  __A3__: In our study, we focus on small $\alpha$ case. In this scenario, there is class imbalance in latent informative/uninformative labeling. Average precision is preferred under this scenario (see [V]).
>
> ---
>
>  __Q4__: Is $\beta$ tuned to maximize precision?
>
>  __A4__: $\beta$ is a fixed hyper-param during the training. As we can see from the ablation study (Figure 6, Page 42), the performance is robust w.r.t choice of $\beta$.
>
> ---
>
> __Q5__: Show how performance varies with different quality of the initial $\hat{f}^0$
>
> __A5__: we have included an ablation study of the quality of $\hat{f}^9$ in Figure 6, Section D.4 in the Appendix. Overall, the performance is stable w.r.t different $\hat{f}^9$.
>
> ---
>
> __Q6__:Describe the main technical challenges to proving convergence guarantees for ISA and ISA-V2 (applying Theorem 2 to Algorithm 1 and relaxations 1-4)
>
> __A6__:The primary challenge arises from two key factors:
>
> 1) Analyzing the gap between minimizing cross-entropy as a surrogate for binary classification. We are optimistic that H-consistency bound framework presented in [III] could be utilized for this purpose.
>
> 2) Analyzing the utilization of Stochastic Gradient Descent (SGD) as opposed to the Empirical Risk Minimization (ERM) oracle in the Theorem 2. In this regard, we believe that tools for analyzing SGD in bi-level optimization problems [IV] could be applied.
>
> ---
>
> __Q7__:Compare to semi-supervised learning methods, i.e. discard only the labels of uninformative data
>
> __A7__:Thank you for the suggestion. We believe this could be a promising direction for future work. In the current framework, the fundamental principle of our proposed method is distinguishing the uninformative data points and to discard them during the training process. While this approach may enhance the statistical efficiency and risk associated with informative data, it may under-utilize the potential of data from a representation learning perspective - for instance, while the uninformative labels are useless by definition, the discarded samples themselves may still be helpful for representation learning.
>
> [III] Awasthi, Pranjal, et al. "H-consistency bounds for surrogate loss minimizers." International Conference on Machine Learning. PMLR, 2022.
>
> [IV] Chen, Tianyi, Yuejiao Sun, and Wotao Yin. "Closing the gap: Tighter analysis of alternating stochastic gradient methods for bilevel problems." Advances in Neural Information Processing Systems 34 (2021)
>
> [V] Saito, Takaya, and Marc Rehmsmeier. "The precision-recall plot is more informative than the ROC plot when evaluating binary classifiers on imbalanced datasets." PloS one 10.3 (2015): e0118432.

---

> > ### Comment · Reviewer_VdEU · 2023-11-21
> > **Future Work**
> >
> > Thank you for thoroughly addressing all of my questions/concerns and for revising the submission. Additionally, I think that adding the answers **A6** and **A7** above would strengthen the future work section.

---

### Review · Reviewer_rVgM · 2023-11-02

**Summary Of Contributions:**

This paper studies the problem of learning from both informative and uniofrmative data. In particular, the setting supposes there are distributions of informative (potentially low noise) and uninformative (potentially high noise) data. The authors propose a selective learning method that attempts to distinguish the informative data points from the uninformative ones. The resulting learner can then ‘condition’ on the informative data points and potentially benefit from an enhanced sample complexity bound. The main contributions are as follows:

The theoretical selective sampling algorithm is proposed, which is iterative where each iteration consists of two stages.

Theoretical results are derived for a theoretical algorithm, which justifies the methodology.

Two practical versions of the algorithms are proposed along with a number of heuristics. Experiments with the practical algorithms show that they are effective in learning to distinguish and use the informative data.

**Audience:**

Yes

**Broader Impact Concerns:**

No concerns.

**Claims And Evidence:**

Yes

**Requested Changes:**

The paper states that the experiments are run with varying $\alpha \in \{ 1, .75, .5, .25 \}$. However, I don’t see where this is represented in the tables. I would advise making this more clear.

The titles like ‘purely uninformative data’ do not match how the datasets are described in the text which makes it difficult to quickly understand the results.

I do not think the results are best represented as a series of long tables. Once again, it is difficult to get a grasp for how the different methods compare without spending a great deal of time scrutinizing the numbers. Many of the tables could be replaced with bar graphs.

For Table 5, is there some way to understand what would be the naive performance of simply doing ERM? From my understanding, Confidence is also doing some selection, but just in a very naive way.

**Strengths And Weaknesses:**

Overall this is a good paper. The theoretical results provide ample support for the methodology of using this selective learning approach. One can see from the Theorem 2 that a good selector will yield a stronger guarantees for the informative points by leveraging the selector.


The loss function it self used to derive the theoretical algorithm is also intuitive and easy to understand, which I think is a major bonus.


The practical algorithms also appear to yield strong results on both synthetic and real datasets, compared to the baselines. The experiments are also thorough and provide ablations for a better understanding of the choices made for the practical algorithms.


One potential downside is that it is not clear how well the method fairs at extreme values of $\alpha$.


Other than this I don’t have really any major weaknesses discuss, but I have pointed out some changes that would better help the presentation of the paper listed below.

---

> ### Author Response · Authors · 2023-11-15
> **Reply to Reviewer rVgM**
>
> We are grateful to the reviewer for regarding our paper as overall good.  We have improved our presentation accordingly and fixed typos. We reply to the reviewer's concern in below:
>
> __Q1__:One potential downside is that it is not clear how well the method fairs at extreme values of $\alpha$.
>
> __A1__: Thanks for pointing out this. The paper exclusively focuses on scenarios  where $\alpha$ is non-vanishing, e.g., $\alpha \geq 0.01$. We believe that the current framework may not be directly applicable when $\alpha \rightarrow o(1)$ for the following reasons.
>
> 1) quality of $\widehat f$: in the case where diminishing fraction of data is informative, applying the current framework alone may not guarantee quality of $\widehat f$ since sample complexity required for learning a good $\widehat f$ is of order $\frac{1}{\alpha}$.
>
>  2) Imbalances of informative/uninformative data: when learning the selector, whether the data is informative or uninformative serves as the binary target that one aims to approximate. The value of alpha characterizes the balance between two classes, e.g., $\alpha=0.5$ represents the simplest balanced binary classification case.
> When $\alpha$ approaches a vanishing value, an additional challenge arises due to the  class imbalance, which requires specialized recipes [I,II].
> We are thinking of exploring this direction in future work.
>
> ---
>
> __Q2__: On the connection between ERM and confidence.
>
> __A2__: Thank you for pointing out such confusion. We have clarified in the paper. The confidence method could be viewed as a proxy for ERM method with following adjustments: 1)  replacing cross-entropy as surrogate loss for binary classification loss; 2) replacing selector $\mathbf{1}\{\widehat g(\boldsymbol x) \geq q\}$ by using estimated margin based selection rule $\mathbf{1}\{ |\widehat f_{ERM}(\boldsymbol{x})-1/2| \geq q\}$.
>
> ---
>
> __Q3__: Presentation related issues
>
> __A3__:
> 1) We modified the title and x-axis ticks' labels of Table 1 and Table 2 to match the description of $\alpha$.
>
> 2) We only kept results from Table 1 and Table 3 in the main text and replaced the table with a bar chart.  We also moved Table 2 and Table 4 to the Appendix and added corresponding charts.
>
> 3) We have added clarification for Table 5 about the connection between confidence and ERM. (section 7.2, page11)
>
> [I] Cao, Kaidi, et al. "Learning imbalanced datasets with label-distribution-aware margin loss." Advances in neural information processing systems 32 (2019).
>
> [II] Khan, Salman, et al. "Striking the right balance with uncertainty." Proceedings of the IEEE/CVF Conference on Computer Vision and Pattern Recognition. 2019.

---

### Author Response · Authors · 2023-11-15
**Overall Response**

Dear Reviewers,

We thank you for the acknowledgement of this work and constructive feedback. We have just uploaded our revised article. The revision are marked as blue in the draft.

Furthermore, we have provided responses to individual questions below. We hope that all concerns have been addressed and are open to further discussion if needed.

---

### Decision · Action_Editor_VbhZ · 2024-01-01

**Recommendation:** Accept with minor revision

**Comment:**

The paper considers learning (mostly classification) in a setting in which a fraction of the data is uninformative, i.e. on that data the observed label is random noise unrelated to the true target. The method proposes simultaneously learning a "selector" which tries to identify which data is informative and the "classifier" which does the classification. The proposed algorithm is an alternate optimization method for learning these two functions. There is an algorithm for which there are guarantees proved under some conditions. There is also an easier to implement algorithm presented, which is more of a  heuristic, and relies on estimating whether or not a data point is informative by evaluating whether or not its predicted label is correct, and then these pseudo-informative labels are used to update the selector. The classifier is trained by weighting the data using the selector.  There are experiments on MNIST and SVHN which demostrate the efficacy of the proposed methods.

**Audience:**

The paper will be of interest to researchers in ML theory and those interested in selective classification.

**Claims And Evidence:**

The claims made in the paper are adequately supported. There is some theory which suggests a particular methodology to be used in practice. There are some experiments that suggest that the algorithms (which use some heuristics not proved by theory) perform well on some standard datasets.